# Change point detection and inference in multivariate non-parametric models under mixing conditions

**Carlos Misael Madrid Padilla**
Department of Mathematics
University of Notre Dame
cmadridp@nd.edu

**Haotian Xu**
Department of Statistics
University of Warwick
haotian.xu.1@warwick.ac.uk

**Daren Wang**
Department of Statistics
University of Notre Dame
dwang24@nd.edu

**Oscar Hernan Madrid Padilla**
Department of Statistics
University of California, Los Angeles
oscar.madrid@stat.ucla.edu

**Yi Yu**
Department of Statistics
University of Warwick
yi.yu.2@warwick.ac.uk

## Abstract

This paper addresses the problem of localizing and inferring multiple change points, in non-parametric multivariate time series settings. Specifically, we consider a multivariate time series with potentially short-range dependence, whose underlying distributions have Hölder smooth densities and can change over time in a piecewise-constant manner. The change points, which correspond to the times when the distribution changes, are unknown. We present the limiting distributions of the change point estimators under the scenarios where the minimal jump size vanishes or remains constant. Such results have not been revealed in the literature in non-parametric change point settings. As byproducts, we develop a sharp estimator that can accurately localize the change points in multivariate non-parametric time series, and a consistent block-type long-run variance estimator. Numerical studies are provided to complement our theoretical findings.

## 1 Introduction

Given a time series $\{X_t\}_{t=1}^T \subset \mathbb{R}^p$, which is assumed to be an $\alpha$-mixing sequence of random vectors with unknown marginal distributions $\{P_t\}_{t=1}^T$. To incorporate the nonstationarity of $\{X_t\}_{t=1}^T$, we assume that there exists $K \in \mathbb{N}$ change points, namely $\{\eta_k\}_{k=1}^K \subset \{2, ..., T\}$ with $1 = \eta_0 < \eta_1 < \ldots < \eta_k \leq T < \eta_{K+1} = T + 1$, such that

$$P_t \neq P_{t-1} \text{ if and only if } t \in \{\eta_1, \ldots, \eta_K\}. \tag{1}$$

Our primary interest is to develop accurate estimators of $\{\eta_k\}_{k=1}^K$ and study their limiting properties. We refer to Assumption 1 for detailed technical conditions of our non-parametric change point model.

Nonstationary multivariate data are frequently encountered in real-world applications, including biology (e.g. Molenaar et al. 2009, Wolkovich & Donahue 2021), epidemiology (e.g. Azhar et al. 2021, Nguyen et al. 2021), social science (e.g. Kunitomo & Sato 2021, Cai et al. 2022), climatology (e.g. Corbella & Stretch 2012, Heo & Manuel 2022), finance (e.g. Herzel et al. 2002, Schmitt et al. 2013), neuroscience (e.g. Gorrostieta et al. 2019, Frolov et al. 2020), among others.

37th Conference on Neural Information Processing Systems (NeurIPS 2023).

Due to the importance of modeling nonstationary data in various scientific fields, we have witnessed a soaring growth of statistical change point literature, (e.g. Aue et al. 2009, Fryzlewicz 2014, Cho & Fryzlewicz 2015, Cho 2016, Wang et al. 2020, Padilla et al. 2022). However, there are a few limitations in the existing works on multivariate non-parametric settings. Firstly, to the best of our knowledge, temporal dependence, which commonly appears in time series, has not been considered. Secondly, there is no localization consistency result for data with the underlying densities being Hölder smooth with arbitrary degree of smoothness. Lastly and most importantly, the limiting distributions of change point estimators and the asymptotic inference for change points have not been well studied.

Taking into account the aforestated limitations, this paper examines change point problems in a fully non-parametric time series framework, wherein the underlying distributions are only assumed to have Hölder smooth continuous densities and can change over time in a piecewise constant manner. The rest of the paper is organized as follows. In Section 2, we explain the model assumptions for multivariate time series with change points in a non-parametric setting. Section 3 details the two-step change point estimation procedure, as well as the estimators at each step. Theoretical results, including the consistency of the preliminary estimator and the limiting distribution of the final estimator, are presented in Section 4. Section 5 evaluates the practical performance of the proposed procedure via various simulations and a real data analysis. Finally, Section 6 concludes with a discussion.

**Notation.** For any function $f : \mathbb{R}^p \to \mathbb{R}$ and for $1 \leq q < \infty$, define $\|f\|_{L_q} = (\int_{\mathbb{R}^p} |f(x)|^q dx)^{1/q}$ and for $q = \infty$, define $\|f\|_{L_\infty} = \sup_{x \in \mathbb{R}^p} |f(x)|$. Define $L_q = \{f : \mathbb{R}^p \to \mathbb{R}, \|f\|_q < \infty\}$. Moreover, for $q = 2$, define $\langle f, g \rangle_{L_2} = \int_{\mathbb{R}^p} f(x)g(x)dx$ where $f, g : \mathbb{R}^p \to \mathbb{R}$. For any vector $s = (s_1, \ldots, s_p)^\top \in \mathbb{N}^p$, define $|s| = \sum_{i=1}^p s_i$, $s! = s_1! \cdots s_p!$ and the associated partial differential operator $D^s = \frac{\partial^{|s|}}{\partial x_1^{s_1} \cdots \partial x_p^{s_p}}$. For $\alpha > 0$, denote $\lfloor \alpha \rfloor$ to be the largest integer smaller than $\alpha$. For any function $f : \mathbb{R}^p \to \mathbb{R}$ that is $\lfloor \alpha \rfloor$-times continuously differentiable at point $x_0$, denote by $f_{x_0}^\alpha$ its Taylor polynomial of degree $\lfloor \alpha \rfloor$ at $x_0$, which is defined as $f_{x_0}^\alpha(x) = \sum_{|s| \leq \lfloor \alpha \rfloor} (x - x_0)^s / s! D^s f(x_0)$. For a constant $L > 0$, let $\mathcal{H}^\alpha(L, \mathbb{R}^p)$ be the set of functions $f : \mathbb{R}^p \to \mathbb{R}$ such that $f$ is $\lfloor \alpha \rfloor$-times differentiable for all $x \in \mathbb{R}^p$ and satisfy $|f(x) - f_{x_0}^\alpha(x)| \leq L|x - x_0|^\alpha$, for all $x, x_0 \in \mathbb{R}^p$. Here $|x - x_0|$ is the Euclidean distance between $x, x_0 \in \mathbb{R}^p$. In non-parametric statistics literature, $\mathcal{H}^\alpha(L, \mathbb{R}^p)$ is often referred to as the class of Hölder functions. For two positive sequences $\{a_n\}_{n \in \mathbb{N}^+}$ and $\{b_n\}_{n \in \mathbb{N}^+}$, we write $a_n = O(b_n)$ or $a_n \lesssim b_n$, if $a_n \leq Cb_n$ with some constant $C > 0$ that does not depend on $n$, and $a_n = \Theta(b_n)$ or $a_n \asymp b_n$, if $a_n = O(b_n)$ and $b_n = O(a_n)$. For a deterministic or random $\mathbb{R}$-valued sequence $a_n$, write that a sequence of random variable $X_n = O_p(a_n)$, if $\lim_{M \to \infty} \lim \sup_{n \to \infty} \mathbb{P}(|X_n| \geq Ma_n) = 0$. Write $X_n = o_p(a_n)$ if $\lim \sup_{n \to \infty} \mathbb{P}(|X_n| \geq Ma_n) = 0$ for all $M > 0$. The convergences in distribution and probability are respectively denoted by $\xrightarrow{\mathcal{D}}$ and $\xrightarrow{P}$.

## 2 Model setup

Detailed assumptions imposed on the model (1) are collected in Assumption 1.

**Assumption 1.** *The data $\{X_t\}_{t=1}^T \subset \mathbb{R}^p$ are generated based on model* (1) *and satisfy the following.*
**a.** *For each $t = \{1, \ldots, T\}$, the distribution $P_t$ has a Lebesgue density function $f_t : \mathbb{R}^p \to \mathbb{R}$, such that $f_t \in \mathcal{H}^r(L, \mathcal{X})$ with $r, L > 0$, where $\mathcal{X}$ is the union of the supports of all $f_t$, and $\mathcal{X}$ has bounded Lebesgue measure.*
**b.** *Let $g_t$ be the joint density of $X_1$ and $X_{t+1}$. It satisfies that $\|g_t\|_{L_\infty} < \infty$.*
**c.** *The minimal spacing between two consecutive change points $\Delta = \min_{k=1}^{K+1}(\eta_k - \eta_{k-1}) > 0$.*
**d.** *The minimal jump size between two consecutive change points $\kappa = \min_{k=1,\ldots,K} \kappa_k > 0$, where $\kappa_k = \|f_{\eta_k} - f_{\eta_{k+1}}\|_{L_2}$ denotes the jump size at the kth change point.*
**e.** *The process $\{X_t\}_{t \in \mathbb{Z}}$ is $\alpha$-mixing with mixing coefficients*

$$\alpha_k = \sup_{t \in \mathbb{Z}} \alpha(\sigma(X_s, s \leq t), \sigma(X_s, s \geq t + k)) \leq e^{-2ck} \quad \text{for all } k \in \mathbb{Z}. \tag{2}$$

The minimal spacing $\Delta$ and the minimal jump size $\kappa$ are two key parameters characterizing the change point phenomenon. Assumption 1**d.** characterizes the changes in density functions through the function's $L_2$-norm, enabling us to detect local and global changes in non-parametric settings.

The decay rate of $\alpha_k$ in Assumption 1**e.** imposes an upper bound on the temporal dependence. This is a standard requirement in the literature (e.g Abadi 2004, Merlevède et al. 2009).

Revolving the change point estimators, we are to conduct the estimation and inference tasks. For a sequence of estimators $\widehat{\eta}_1 < \ldots < \widehat{\eta}_{\widehat{K}} \subset \{1, \ldots, T\}$, our first task is to show the localization consistency, i.e. with probability tending to one as the sample size $T$ grows unbounded, it holds that

$$\widehat{K} = K \text{ and } \max_{k=1,\ldots,\widehat{K}} |\widehat{\eta}_k - \eta_k| \le \epsilon, \text{ with } \lim_{T\to\infty} \frac{\epsilon}{\Delta} = 0. \tag{3}$$

We refer to $\epsilon$ as the localization error in the rest of this paper.

With a consistent estimation result, we further refine $\{\widehat{\eta}_k\}_{k=1}^{\widehat{K}}$ and obtain $\{\widetilde{\eta}_k\}_{k=1}^{\widehat{K}}$ with error bounds $|\widetilde{\eta}_k - \eta_k| = O_p(1)$ and derive the limiting distribution of $(\widetilde{\eta}_k - \eta_k)\kappa_k^{\frac{p}{r}+2}$.

We briefly summarize the contributions of our paper as follows.

- We develop a multivariate non-parametric seeded change point detection algorithm detailed Algorithm 1, which is based on the seeded binary segmentation method (SBS), proposed in Kovács et al. (2020) in the univariate setting. To the best of our knowledge, we are the first to innovatively adapt SBS to a multivariate non-parametric change point model.
- Under the signal-to-noise ratio condition in Assumption 3 that $\kappa^2\Delta \gtrsim \log(T)T^{p/(2r+p)}$, we demonstrate that the output of Algorithm 1 is consistent, with localization errors $\epsilon \asymp \kappa_k^{-2}T^{p/(2r+p)}\log(T)$, for $k \in \{1, \ldots, K\}$. This localization error is first obtained for $\alpha$-mixing time series with a generic smoothness assumption, while the state-of-the-art method from Padilla et al. (2021) only focuses on Lipschitz smooth densities and under temporal independence.
- Based on the consistent estimators $\{\widehat{\eta}\}_{k=1}^{\widehat{K}}$, we construct the refined estimators $\{\widetilde{\eta}_k\}_{k=1}^{\widehat{K}}$ and derive their limiting distributions in different regimes, as detailed in Theorem 2. These results are novel in the literature of change point and time series analysis.
- Extensive numerical results are presented in Section 5 to corroborate the theoretical findings. The code used for numerical experiments is available upon request prior to publication.

## 3 A two-step multivariate non-parametric change point estimators

In this section, we present the initial and refined change point estimators, both of which share the same building block, namely the non-parametric CUSUM statistic.

**Definition 1** (Non-parametric CUSUM statistic)**.** *For any integer triplet $0 \le s < t < e \le T$, let the CUSUM statistic be*

$$\widetilde{F}_{t,h}^{(s,e)}(x) = \sqrt{\frac{e-t}{(e-s)(t-s)}} \sum_{i=s+1}^{t} F_{i,h}(x) - \sqrt{\frac{t-s}{(e-s)(e-t)}} \sum_{i=t+1}^{e} F_{i,h}(x), \ x \in \mathbb{R}^p,$$

*where $F_{t,h}$ is a kernel estimator of $f_t$, i.e. $F_{t,h}(x) = \mathcal{K}_h(x - X_t)$ with the kernel function*

$$\mathcal{K}_h(x) = \frac{1}{h^p}\mathcal{K}\left(\frac{x}{h}\right), \quad x \in \mathbb{R}^p,$$

*accompanied with the bandwidth $h > 0$.*

The CUSUM statistic is a key ingredient of our algorithm and is based on the kernel estimator $F_{t,h}(\cdot)$. We highlight that kernel-based change-point estimation techniques have been employed in detecting change points in non-parametric models in existing literature, as demonstrated in, for instance, Arlot et al. (2019), Li et al. (2019), Padilla et al. (2021).

Our preliminary estimator is obtained by combining the CUSUM statistic in Definition 1 with a modified version of SBS based on a collection of deterministic seeded intervals defined in Definition 2.

**Definition 2** (Seeded intervals)**.** *Let $\mathfrak{K} = \left\lceil C_{\mathfrak{K}} \log_2\left(\frac{T}{\Delta}\right)\right\rceil$, with some sufficiently large absolute constant $C_{\mathfrak{K}} > 0$. For $k \in \{1, \ldots, \mathfrak{K}\}$, let $\mathcal{J}_k$ be the collection of $2^k - 1$ intervals of length $l_k = T2^{-k+1}$ that are evenly shifted by $l_k/2 = T2^{-k}$, i.e.*

$$\mathcal{J}_k = \{(\lfloor (i-1)T2^{-k}\rfloor, \lceil (i-1)T2^{-k} + T2^{-k+1}\rceil], \quad i = 1, \ldots, 2^k - 1\}.$$

*The overall collection of seeded intervals is denoted as $\mathcal{J} = \cup_{k=1}^{\mathfrak{K}}\mathcal{J}_k$.*

With the CUSUM statistics and the seeded intervals as building blocks, we are now ready to present our multivariate non-parametric seeded change point detection algorithm.

---

**Algorithm 1** Multivariate non-parametric Seeded Binary Segmentation. MNSBS $((s, e), \mathcal{J}, \tau, h)$

---

**INPUT:** Sample $\{X_t\}_{t=s}^e \subset \mathbb{R}^p$, collection of seeded intervals $\mathcal{J}$, tuning parameter $\tau > 0$ and bandwidth $h > 0$.

    **initialization**: If $(s, e] = (0, n]$, set $\mathbf{S} \to \varnothing$ and set $\rho \to \log(T)h^{-p}$.

    **for** $\mathcal{I} = (\alpha, \beta] \in \mathcal{J}$ **do**

        **if** $\mathcal{I} = (\alpha, \beta] \subseteq (s, e]$ and $\beta - \alpha > 2\rho$ **then**

            $b_{\mathcal{I}} \leftarrow \arg\max_{\alpha + \rho \leq t \leq \beta - \rho} \|\widetilde{F}_{t,h}^{(\alpha, \beta]}\|_{L_2}$

            $a_{\mathcal{I}} \leftarrow \|\widetilde{F}_{b_{\mathcal{I}},h}^{(\alpha, \beta]}\|_{L_2}$

        **else**

            $a_{\mathcal{I}} \leftarrow -1$

        **end if**

    **end for**

    $\mathcal{M}^{s,e} = \{\mathcal{I} : a_{\mathcal{I}} > \tau\}$

    **if** $\mathcal{M}^{s,e} \neq \emptyset$ **then**

        $\mathcal{I}^* \leftarrow \arg\min_{\mathcal{I} \in \mathcal{M}^{s,e}} |\mathcal{I}|$

        $\mathbf{S} \leftarrow \mathbf{S} \cup \{b_{\mathcal{I}^*}\}$

        MNSBS$((s, b_{\mathcal{I}^*}), \mathcal{J}, \tau, h)$

        MNSBS$((b_{\mathcal{I}^*} + 1, e), \mathcal{J}, \tau, h)$

    **end if**

**OUTPUT:** The set of estimated change points $\mathbf{S}$.

---

Algorithm 1 presents a methodological approach to addressing the problem of estimating multiple change points in multivariate time series data. At its core, the algorithm leverages the strength of seeded intervals, forming a multi-scale search mechanism. To identify potential change points, the method recursively employs the CUSUM statistics. For the functionality of the algorithm, specific inputs are required. These include the observed data set, represented as $X_t{}_{t=1}^T$, the seeded intervals denoted by $\mathcal{J}$, the bandwidth $h$ that is crucial for constructing the CUSUM statistics, and a threshold, $\tau$, which is instrumental in change point detection. We provide theoretical and numerical guidance for tuning parameters in Sections 4 and 5, respectively.

Delving deeper into the architecture of Algorithm 1, it becomes evident that the SBS functions as its foundational framework, while the nonparametric version of the CUSUM statistics acts as its functional units. The design of this algorithm is particularly tailored given its inclination toward nonparametric detection and its ability to identify multiple change points. The SBS is, in essence, an advanced version of the moving-window scanning technique. Its distinctive characteristic is its adaptability in handling the challenges posed by multiple change points that exhibit unpredictable spacing. Instead of being confined to a fixed window width, the SBS introduces versatility by incorporating a range of window width options. Each of these widths is methodically applied during a moving-window scan.

Based on the preliminary estimators $\{\widehat{\eta}_k\}_{k=1}^{\widehat{K}}$ provided by Algorithm 1, we further develop a refinement procedure to enhance the localization accuracy. To be more specific, let

$$s_k = \frac{9}{10}\widehat{\eta}_{k-1} + \frac{1}{10}\widehat{\eta}_k \quad \text{and} \quad e_k = \frac{9}{10}\widehat{\eta}_{k+1} + \frac{1}{10}\widehat{\eta}_k. \tag{4}$$

Then, the preliminary estimators $\{\widehat{\eta}_k\}_{k=1}^{\widehat{K}}$ and $\widetilde{h} \asymp h$ produce an estimator of $\kappa_k$ as:

$$\widehat{\kappa}_k = \left\| \sqrt{\frac{\widehat{\eta}_{k+1} - \widehat{\eta}_k}{(\widehat{\eta}_{k+1} - \widehat{\eta}_{k-1})(\widehat{\eta}_k - \widehat{\eta}_{k-1})}} \sum_{i=\widehat{\eta}_{k-1}+1}^{\widehat{\eta}_k} F_{i,\widetilde{h}} - \sqrt{\frac{(\widehat{\eta}_k - \widehat{\eta}_{k-1})}{(\widehat{\eta}_{k+1} - \widehat{\eta}_{k-1})(\widehat{\eta}_{k+1} - \widehat{\eta}_k)}} \sum_{i=\widehat{\eta}_k+1}^{\widehat{\eta}_{k+1}} F_{i,\widetilde{h}} \right\|_{L_2}$$

$$\times \sqrt{\frac{\widehat{\eta}_{k+1} - \widehat{\eta}_{k-1}}{(\widehat{\eta}_k - \widehat{\eta}_{k-1})(\widehat{\eta}_{k+1} - \widehat{\eta}_k)}}. \tag{5}$$

We then propose the final change points estimators as

$$\widetilde{\eta}_k = \underset{s_k < \eta < e_k}{\arg\min} \widehat{Q}_k(\eta), \tag{6}$$

where

$$\widehat{Q}_k(\eta) = \Big\{ \sum_{t=s_k+1}^{\eta} \|F_{t,h_1} - F_{(s_k,\widehat{\eta}_k],h_1}\|_{L_2}^2 + \sum_{t=\eta+1}^{e_k} \|F_{t,h_1} - F_{(\widehat{\eta}_k,e_k],h_1}\|_{L_2}^2 \Big\},$$

with $h_1 = c_{\widehat{\kappa}_k} \widehat{\kappa}_k^{1/r}$ and $F_{(s,e],h_1} = \frac{1}{e-s} \sum_{i=s+1}^{e} F_{i,h_1}$ for integers $e > s$.

If the initial change point estimators are consistent, i.e. (3) holds with probability tending to 1, then the interval $(\widehat{\eta}_{k-1}, \widehat{\eta}_{k+1})$ is anticipated to contain merely one undetected change point. By conservatively trimming this interval to $(s_k, e_k)$, we can safely any change points previously detected within $(\widehat{\eta}_{k-1}, \widehat{\eta}_{k+1})$. Consequently, the trimmed interval $(s_k, e_k)$ contains only true change point $\eta_k$ with high probability. Due to the same reason, our choice of weight in Equation 4,1/10, is a convenient choice. In general, any constant weight between 0 and 1/2 would suffice. Inspired by Padilla et al. (2021), who proposed to use $O(\kappa_k)$ as an optimal bandwidth in the context of Lipschitz densities, we adopt $h_1 = O\left(\widehat{\kappa}_k^{1/r}\right)$ as the bandwidth for our kernel density estimator. This choice incorporates the broader scope of our work, which studies a more general degree of smoothness. Notably, if the underlying density functions strictly adhere to the Lipschitz criterion and $r = 1$, our bandwidth selection aligns with that recommended by Padilla et al. (2021). We would like to emphasize that while the procedure proposed by Padilla et al. (2021) required knowledge of the population quantities $\kappa_k$, our approach is adaptive as we provide data-driven methods to estimate $\kappa_k$ accurately.

With our newly proposed estimators, in Theorem 2, we derive an improved error bound for the refined estimators $\{\widetilde{\eta}_k\}_{k=1}^{\widehat{K}}$ over the preliminary estimators $\{\widehat{\eta}_k\}_{k=1}^{\widehat{K}}$. We also study the limiting distributions of the refined estimators. Section 4.4 and Section 5 will discuss the theoretically justified rates and practical choices of tuning parameters, respectively.

The computational complexity of Algorithm 1 is $O(T^2 \log(T)\Delta^{-1} \cdot \text{Kernel})$, where $O(T^2 \log(T)\Delta^{-1})$ is due to the computational cost of the SBS, and "Kernel" stands for the computational cost of numerical computation of the $L_2$-norm of the CUSUM statistics based on the kernel function evaluated at each time point. The dependence on the dimension $p$ is only through the evaluation of the kernel function. The computational complexity of the final estimators (including estimating $\widehat{\kappa}_k$'s) is $O(T \cdot \text{Kernel})$. Therefore, the overall cost for deriving $\{\widetilde{\eta}_k\}_{k=1}^{\widehat{K}}$ is $O(T^2 \log T \Delta^{-1} \cdot \text{Kernel})$.

## 4 Consistent estimation and limiting distributions

To establish the theoretical guarantees of our estimators, we first state conditions needed for the kernel function $\mathcal{K}(\cdot)$.

**Assumption 2** (The kernel function). *Assume that the kernel function $\mathcal{K}(\cdot)$ has compact support and satisfies the following additional conditions.*

**a.** *For the Hölder smooth parameter $r$ in Assumption 1a, assume that $\mathcal{K}(\cdot)$ is adaptive to $\mathcal{H}^r(L, \mathbb{R}^p)$, i.e. for any $f \in \mathcal{H}^r(L, \mathbb{R}^p)$, it holds that*

$$\sup_{x \in \mathbb{R}^p} \Big| \int_{\mathbb{R}^p} h^{-p} \mathcal{K}\Big(\frac{x-z}{h}\Big) f(z) \, \mathrm{d}z - f(x) \Big| \le \widetilde{C} h^r,$$

*for some absolute constant $\widetilde{C} > 0$ and tuning parameter $h > 0$.*

**b.** *The class of functions $\mathcal{F}_\mathcal{K} = \{\mathcal{K}(x - \cdot)/h : \mathbb{R}^p \to \mathbb{R}^+, h > 0\}$ is separable in $L_\infty(\mathbb{R}^p)$ and is a uniformly bounded VC-class; i.e. there exist constants $A, \nu > 0$ such that for any probability measure $Q$ on $\mathbb{R}^p$ and any $u \in (0, \|\mathcal{K}\|_{L_\infty})$, it holds that $\mathcal{N}(\mathcal{F}_\mathcal{K}, L_2(Q), u) \le (A\|\mathcal{K}\|_{L_\infty}/u)^\nu$, where $\mathcal{N}(\mathcal{F}_\mathcal{K}, L_2(Q), u)$ denotes the $u$-covering number of the metric space $(\mathcal{F}_\mathcal{K}, L_2(Q))$.*

**c.** *Let $m = \lfloor r \rfloor$ and it holds that $\int_0^\infty t^{m-1} \sup_{\|x\| \ge t} |\mathcal{K}(x)|^m \, \mathrm{d}t < \infty$, $\int_{\mathbb{R}^p} \mathcal{K}(z)\|z\| \, \mathrm{d}z \le C_\mathcal{K}$, where $C_\mathcal{K} > 0$ is an absolute constant.*

Assumption 2 is a standard condition in the non-parametric literature (e.g. Giné & Guillou 1999, 2001, Sriperumbudur & Steinwart 2012, Kim et al. 2019, Padilla et al. 2021) and holds for various kernels, such as the Triweight, Epanechnikov and Gaussian kernels, which are considered in Section 5.

## 4.1 Consistency of preliminary estimators

To establish the consistency of the preliminary estimators outputted by Algorithm 1, we impose the following signal-to-noise ratio condition.

**Assumption 3** (Signal-to-noise ratio). *Assume there exists an arbitrarily slow diverging sequence $\gamma_T > 0$ such that*

$$\kappa^2 \Delta > \gamma_T \log(T) T^{\frac{p}{2r+p}}.$$

We note that Assumption 3 is a mild condition, as it allows both the jump size $\kappa$ to vanish asymptotically and/or the spacing $\Delta$ between change points to be much smaller than $T$. The consistency of Algorithm 1 is established in the following theorem.

**Theorem 1.** *Suppose Assumptions 1, 2 and 3 hold. Let $\{\widehat{\eta}_k\}_{k=1}^{\widehat{K}}$ be the estimated change points returned by Algorithm 1 with tuning parameters $\tau = c_\tau T^{p/(4r+2p)} \log^{1/2}(T)$ and $h = c_h T^{-1/(2r+p)}$ for sufficiently large constants $c_h, c_\tau > 0$. Then*

$$\mathbb{P}\Big\{\widehat{K} = K,\ |\widehat{\eta}_k - \eta_k| \leq C_\epsilon \kappa_k^{-2} T^{\frac{p}{2r+p}} \log(T), \forall k = 1, \ldots, K\Big\} \geq 1 - 3C_{p,\mathcal{K}} T^{-1},$$

*where $C_\epsilon$ and $C_{p,\mathcal{K}}$ are positive constants only depending on the kernel and the dimension $p$.*

## 4.2 Refined estimators and their limiting distributions

To develop refined estimators based on the preliminary estimators and study their limiting distributions, we would need to require a slightly stronger signal-to-noise ratio condition below.

**Assumption 4** (Signal-to-noise ratio for inference). *Assume that there exists an arbitrarily slow diverging sequence $\gamma_T > 0$ such that*

$$\kappa^{\frac{2p}{r}+3} \Delta > \gamma_T \log(T) T^{\frac{p}{2r+p}}.$$

Assumption 4 is slightly stronger than Assumption 3. This is because our refined estimators are based on a sequence of random endpoints, i.e. the preliminary estimators. This brings theoretical challenges in deriving limiting distributions and estimating the long-run variances. It is worth noting that a similar phenomenon has been observed in the study on conducted by Xu, Wang, Zhao & Yu (2022).

**Theorem 2.** *Suppose that Assumptions 1, 2 and 3 hold. Let $\{\widetilde{\eta}_k\}_{k=1}^{\widehat{K}}$ be the refined change point estimators defined in Section 3, with the preliminary estimators $\{\widehat{\eta}_k\}_{k=1}^{\widehat{K}}$ returned by Algorithm 1, the intervals $\{(s_k, e_k)\}_{k=1}^{\widehat{K}}$ defined in (4), and $\widehat{\kappa}_k$ defined as in (5). The following holds:*

**a.** *(Non-vanishing regime) Suppose the jump size at the change point $\eta_k$ satisfies $\lim_{T\to\infty} \kappa_k \to \varrho_k$ for some absolute constant $\varrho_k > 0$. Then, as $T \to \infty$, it holds that $|\widetilde{\eta}_k - \eta_k| = O_p(1)$ and that*

$$(\widetilde{\eta}_k - \eta_k)\kappa_k^{\frac{p}{r}+2} \xrightarrow{\mathcal{D}} \underset{\widetilde{r}\in\mathbb{Z}}{\arg\min}\ P_k(\widetilde{r})$$

*where*

$$P_k(\widetilde{r})$$
$$= \begin{cases} \sum_{t=\widetilde{r}+1}^{0} 2\big\langle F_{\eta_k+t,h_2} - f_{\eta_k+t} * \mathcal{K}_{h_2}, (f_{\eta_k} - f_{\eta_{k+1}}) * \mathcal{K}_{h_2}\big\rangle_{L_2} + \widetilde{r}\|(f_{\eta_{k+1}} - f_{\eta_k}) * \mathcal{K}_{h_2}\|_{L_2}^2, & \widetilde{r} < 0; \\ 0, & \widetilde{r} = 0; \\ \sum_{t=1}^{\widetilde{r}} 2\big\langle F_{\eta_k+t,h_2} - f_{\eta_k+t} * \mathcal{K}_{h_2}, (f_{\eta_{k+1}} - f_{\eta_k}) * \mathcal{K}_{h_2}\big\rangle_{L_2} + \widetilde{r}\|(f_{\eta_{k+1}} - f_{\eta_k}) * \mathcal{K}_{h_2}\|_{L_2}^2, & \widetilde{r} > 0. \end{cases}$$

*Here $*$ denotes convolution and $h_2 = c_{\kappa_k}\kappa_k^{1/r}$ for some absolute constant $c_{\kappa_k} > 0$.*
**b.** *(Vanishing regime) Suppose the jump size at the change point $\eta_k$ satisfies $\lim_{T\to\infty} \kappa_k = 0$. Then, as $T \to \infty$, it holds that $|\widetilde{\eta}_k - \eta_k| = O_p(\kappa_k^{-2-p/r})$ and that*

$$(\widetilde{\eta}_k - \eta_k)\kappa_k^{\frac{p}{r}+2} \xrightarrow{\mathcal{D}} \underset{\widetilde{r}\in\mathbb{Z}}{\arg\min}\ \{\widetilde{\sigma}_\infty(k)B(\widetilde{r}) + |\widetilde{r}|\}, \tag{7}$$

*where $h_2 = c_{\kappa_k} \kappa_k^{1/r}$ for some absolute constant $c_{\kappa_k} > 0$. Here*

$$B(\widetilde{r}) = \begin{cases} B_1(-\widetilde{r}), & \widetilde{r} < 0, \\ 0, & \widetilde{r} = 0, \\ B_2(\widetilde{r}), & \widetilde{r} > 0, \end{cases}$$

*with $B_1(r)$ and $B_2(r)$ being two independent standard Brownian motions, and*

$$\widetilde{\sigma}_\infty^2(k) = \lim_{T \to \infty} \frac{\kappa_k^{\frac{p}{r} - 2}}{T} \mathrm{Var}\Big( \sum_{t=1}^{T} \big\langle F_{t,h_2} - f_t * \mathcal{K}_{h_2}, (f_{\eta_k} - f_{\eta_{k+1}}) * \mathcal{K}_{h_2} \big\rangle_{L_2} \Big). \tag{8}$$

Theorem 2 considers vanishing and non-vanishing regimes of the jump sizes. The upper bounds on the localization error in both regimes can be written as

$$\max_{1 \leq k \leq K} |\widetilde{\eta}_k - \eta_k| \kappa_k^{\frac{p}{r} + 2} = O_p(1).$$

Therefore, when the Hölder smoothness parameter $r = 1$, our final estimator $\{\widetilde{\eta}_k\}$ attains the minimax optimal convergence rate developed in Lemma 3 by Padilla et al. (2021). Furthermore, when $r = 1$, our resulting rate is sharper than that in Theorem 1 in Padilla et al. (2021), as we are able to remove the logarithmic factors from the upper bound. Additionally, our method can achieve optimal rates with choices of tuning parameters that do not depend on the unknown jump sizes $\kappa_k$.

Theorem 2 summarizes the limiting distributions of the refined estimators $\{\widetilde{\eta}_k\}_{k=1}^{\widehat{K}}$. In the non-vanishing case, the resulting limiting distribution can be approximated by a two-sided random walk and the change points can be accurately estimated within a constant error rate. In contrast, in the vanishing regime, a central limit theorem under mixing conditions leads to a two-sided Brownian motion distribution in the limit, which quantifies the asymptotic uncertainty of $\{\widetilde{\eta}_k\}_{k=1}^{\widehat{K}}$, enabling inference on change point locations, and allowing for the construction of confidence intervals.

### 4.3 Consistent long-run variance estimation

To obtain valid confidence intervals for change points using the limiting distributions in Theorem 2**b.**, it is crucial to access robust estimators for the long-run (asymptotic) variances $\{\widetilde{\sigma}_\infty^2(k)\}_{k=1}^{\widehat{K}}$ defined in (8). We propose a block-type long-run variance estimator in Algorithm 2 to fulfill this task and demonstrate its consistency in the following theorem.

**Theorem 3.** *Suppose Assumptions 1, 2 and 3 hold. Let $\{\widetilde{\sigma}_\infty^2(k)\}_{k=1}^{\widehat{K}}$ be the population long-run variance defined in (8) and $\widehat{\sigma}_\infty^2(k)$ the output of Algorithm 2 with $R = O(T^{(p+r)/(2r+p)}/\kappa_k^{p/(2r)+3/2})$. Then it holds that*

$$\max_{k=1}^{K} \big| \widehat{\sigma}_\infty^2(k) - \widetilde{\sigma}_\infty^2(k) \big| \xrightarrow{P} 0 \quad as \quad T \to \infty.$$

### 4.4 Discussions on MNSBS

**Tuning parameters.** Our procedure comprises three steps: (1) preliminary estimation, (2) local refinement, and (3) confidence interval construction with three key tuning parameters. For step (1), we use a kernel density estimator with bandwidth $h \asymp T^{-1/(2r+p)}$, which follows from the classical non-parametric literature (e.g. Yu 1993, Tsybakov 2009). The threshold tuning parameter $\tau$ is set to a high-probability upper bound on the CUSUM statistics when there is no change point, which reflects the requirement on the signal-to-noise ratio detailed in Assumption 3. For refined estimation in step (2) and long-run variance estimation in step (3), we set the bandwidth parameter $h_1 \asymp \widehat{\kappa}_k^{1/r}$. This choice of bandwidth is inspired by the minimax rate-optimal bandwidth used in Padilla et al. (2021).

**Comparison with Padilla et al. (2021).** Our main contribution is deriving the limiting distribution of multivariate non-parametric change point estimators. This problem has not been formally studied in the existing literature. Additionally, our Hölder assumption is more general than the Lipschitz assumption used in Padilla et al. (2021). Our Assumption 1**d** specifies changes through the $L_2$-norm of probability density functions, which is a weaker assumption than the $L_\infty$-norm used in Padilla

---

**Algorithm 2** Long-run variance estimators

---

**INPUT:** $\{X_t\}_{t=1}^T, \{\widehat{\eta}_k\}_{k=1}^{\widehat{K}}, \{\widehat{\kappa}_k\}_{k=1}^{\widehat{K}}, \{(s_k, e_k)\}_{k=1}^{\widehat{K}}$ and tuning parameter $R \in \mathbb{N}$

    **for** $k = 1, \ldots, \widehat{K}$ **do**

        Let $h_1 = c_{\widehat{\kappa}} \widehat{\kappa}_k^{\frac{1}{r}}$

        **for** $t \in \{s_k, \ldots, e_k - 1\}$ **do**

            $Y_t = \widehat{\kappa}_k^{\frac{p}{2r}-1} \big\langle F_{t,h_1} - f_t * \mathcal{K}_{h_1}, (f_{\widehat{\eta}_k} - f_{\widehat{\eta}_{k+1}}) * \mathcal{K}_{h_1} \big\rangle_{L_2}$

        **end for**

        $S = \lfloor \frac{e_k - s_k}{R} \rfloor$

        **for** $r \in \{1, \ldots, R\}$ **do**

            $\mathcal{S}_r = \{s_k + (r-1)S, \ldots, s_k + rS - 1\}$

        **end for**

        $\widehat{\sigma}_\infty^2(k) = \frac{1}{R} \sum_{r=1}^R \left( \frac{1}{\sqrt{S}} \sum_{i \in \mathcal{S}_r} Y_i \right)^2$

    **end for**

**OUTPUT:** $\{\widehat{\sigma}_\infty^2(k)\}_{k=1}^{\widehat{K}}$.

---

et al. (2021). Furthermore, our assumptions allow for temporal dependence captured by $\alpha$-mixing coefficients, whereas Padilla et al. (2021) assumed independent observations.

**Comparison with existing literature in nonparametric, online, and inference change point.**
In the nonparametric change point literature, different kernel-based methods are adopted for change point localisation and testing. In the offline setting, the penalized kernel least squares estimator, originally introduced by Harchaoui & Cappé (2007), was explored by Arlot et al. (2012) for multivariate change point problems, and an oracle inequality was derived. An upper bound on the localization rate provided by this method was established by Garreau & Arlot (2018) and was computationally enhanced further in Celisse et al. (2018). With a focus on a so-called running maximum partition strategy, Harchaoui et al. (2008) formulated a kernel-based test statistic to ascertain the existence of a change-point. In a similar vein, Zou et al. (2014) investigated a problem where $s$ out of $n$ sequences are anomalous and devised a test statistic using the kernel maximum mean discrepancy.

In the online setting, Kifer et al. (2004) introduces a meta-algorithm comparing data from a "reference window" to current data using empirical measures. Desobry et al. (2005) detects shifts by comparing two descriptor sets from the signal's immediate past and future, using a dissimilarity metric resembling the Fisher ratio in Gaussian cases via a soft margin single-class SVM. Meanwhile, Liu et al. (2013) adopts density ratio estimation with a non-parametric Gaussian kernel model for change-point detection, updating its parameters online through stochastic gradient descent.

The core methodology is largely shared but with different goals and performance measurements regarding online and offline change point literature comparisons. How to conduct inference in the online change point context is also unclear.

Compared to the existing work, in this paper, we follow the suit of using kernel-based CUSUM statistics but incorporate temporal dependence, which is rarely seen in the literature. Most importantly, we are unaware of existing work on nonparametric change point inference, which is the main selling point of our paper.

Most change point inference work focuses on fixed-dimensional parameters as well as lacks tracking of many model parameters. Xu, Wang, Zhao & Yu (2022), in terms of style, is indeed the most closely related. But tackles high-dimensional linear regression, fundamentally distinct from our nonparametric density estimation.

## 5   Numerical Experiments

We refer to MNSBS as our final estimator, which is used for both localization and inference tasks. To evaluate its localization performance, we compare our proposed method against four competitors – MNP (Padilla et al. 2021), EMNCP (Matteson & James 2014), SBS (Cho & Fryzlewicz 2015) and DCBS (Cho 2016) – across a wide range of simulation settings, using corresponding R functions in `changepoints` (Xu, Padilla, Wang & Li 2022), `ecp` (James et al. 2019) and `hdbinseg` (Cho

& Fryzlewicz 2018) packages. However, to the best of our knowledge, no competitor is currently available for the inference task.

**Tuning parameters.** For MNSBS implementation, we use the Gaussian kernel and the false discovery rate control-based procedure of Padilla et al. (2021) for $\tau$ selection. Preliminary estimators are set as $h = 2 \times (1/T)^{1/(2r+p)}$, while the second stage estimator has bandwidths respectively set as $\widetilde{h} = 0.05$ and $h_1 = 2 \times \widehat{\kappa}_k^{1/r}$. Selection of $R = \big\lfloor \big( \max_{k=1}^{\widehat{K}} \{e_k - s_k\} \big)^{3/5} \big\rfloor$ with $\{(s_k, e_k)\}_{k=1}^{\widehat{K}}$ is guided by Theorem 3 using $\{(s_k, e_k)\}_{k=1}^{\widehat{K}}$ from (4). For the confidence interval construction, we use $\{\widehat{\kappa}_k\}_{k=1}^{\widehat{K}}$ and $\{\widehat{\sigma}_\infty^2(k)\}_{k=1}^{\widehat{K}}$ to estimate the required unknown quantities. We evaluate $L_2$ based statistics in Change point estimation and Long-run variance estimation using the Subregion-Adaptive Vegas Algorithm[1] with a maximum of $10^5$ function evaluations.

**Evaluation measurements** For a given set of true change points $\mathcal{C} = \{\eta_k\}_{k=0}^{K+1}$, to assess the accuracy of the estimator $\widehat{\mathcal{C}} = \{\widehat{\eta}_k\}_{k=0}^{\widehat{K}+1}$ with $\widehat{\eta}_0 = 1$ and $\widehat{\eta}_{T+1} = T + 1f$, we report (1) Misestimation rate: the proportion of misestimating $K$ and (2) Scaled Hausdorff distance: $d_{\mathrm{H}}(\widehat{\mathcal{C}}, \mathcal{C})$, defined by $d_{\mathrm{H}}(\widehat{\mathcal{C}}, \mathcal{C}) = \frac{1}{T} \max\{\max_{x\in\widehat{\mathcal{C}}} \min_{y\in\mathcal{C}}\{|x - y|\}, \max_{y\in\widehat{\mathcal{C}}} \min_{x\in\mathcal{C}}\{|x - y|\}\}$.

The performance of our change point inference is measured by the coverage of $\eta_k$, defined as $cover_k(1 - \alpha)$ for significance level $\alpha \in (0, 1)$. For, $k = 1, \ldots, K$,

$$cover_k(1 - \alpha) = \mathbb{1}\left\{\eta_k \in \left[\widetilde{\eta}_k + \frac{\widehat{q}_u(\alpha/2)}{\widehat{\kappa}_k^{p/r+2}}, \widetilde{\eta}_k + \frac{\widehat{q}_u(1 - \alpha/2)}{\widehat{\kappa}_k^{p/r+2}}\right]\right\},$$

with $\widehat{q}_u(\alpha/2)$ and $\widehat{q}_u(1 - \alpha/2)$ are the $\alpha/2$ and $1 - \alpha/2$ empirical quantiles of the simulated limiting distribution given in (7), $\widehat{\kappa}_k$ is defined in (5), and $k = 1, \ldots, K$.

## 5.1 Localization

We consider three different scenarios with two equally spaced change points. For each scenario, we set $r = 2$, and vary $T \in \{150, 300\}$ and $p \in \{3, 5\}$. Moreover, we consider $\{Y_t = \mathbb{1}\{\lfloor T/3 \rfloor < t \leq \lfloor 2T/3 \rfloor\}Z_t + X_t\}_{t=1}^T \subset \mathbb{R}^p$ with $X_t = 0.3X_{t-1} + \epsilon_t$.

• **Scenario 1 (S1)** Let $Z_t = \mu \in \mathbb{R}^p$, where $\mu_j = 0$ for $j \in \{1, \ldots, \lceil p/2 \rceil\}$ and $\mu_j = 2$ otherwise. Let $\{\epsilon_t\}$ be i.i.d. $\mathcal{N}(0_p, I_p)$.

• **Scenario 2 (S2)** Let $Z_t|\{u_t = 1\} = 1.5 \times 1_p$, $Z_t|\{u_t = 0\} = -1.5 \times 1_p$, where $\{u_t\}$ are i.i.d. Bernoulli(0.5) random variables. Let $\{\epsilon_t\}$ be i.i.d. $\mathcal{N}(0_p, I_p)$.

• **Scenario 3 (S3)** Let $Z_t = 0.3Z_{t-1} + 0.5 \times 1_p + \epsilon_t^*$, where $\{\epsilon_t^*\} \subset \mathbb{R}^p$ and $\{\epsilon_t\} \subset \mathbb{R}^p$ are mutually independent. They are i.i.d. with entries independently follow Unif$(-\sqrt{3}, \sqrt{3})$ and the standardized Pareto$(3, 1)$, respectively.

**S1-S3** encompass a variety of simulation settings, including the same type of distributions, changed mean and constant covariance in **S1**; a mixture of distributions in **S2**; and change between light-tailed and heavy-tailed distributions in **S3**. We conduct 200 repetitions of each experiment and present the results for localization in Figure 1. Our proposed method, MNSBS, generally outperforms all other methods in all scenarios, except for **S2**, where ECP performs better. However, we observe that MNSBS achieves comparable performance to ECP for large $T$ in **S2**.

## 5.2 Inference

In this section, we focus solely on analyzing the limiting distribution obtained in Theorem 2.**b.**, which pertains to the vanishing regime. We explain this from two different perspectives. Firstly, in the non-vanishing regime (Theorem 2.**a.**), the localization error is at the order of $O(1)$. As a result, the construction of confidence intervals, which is a direct application of the limiting distribution, is of little demand with such estimation results. Secondly, since the localization error is only at the order of $O(1)$, the universality cannot come into play to produce a useful limiting distribution.

We consider the process $\{Y_t = \mathbb{1}\{\lfloor T/2 \rfloor < t \leq T\}\mu + X_t\}_{t=1}^T$, with $X_t = 0.3X_{t-1} + \epsilon_t$, Here, $\mu = 1_p$ and $\{\epsilon_t\}$ are i.i.d. $\mathcal{N}(0_p, I_p)$. We vary $T \in \{100, 200, 300\}$ and $p \in \{2, 3\}$, and observe

---

[1]The Subregion-Adaptive Vegas Algorithm is available in R package `cubature` (Narasimhan et al. 2022)

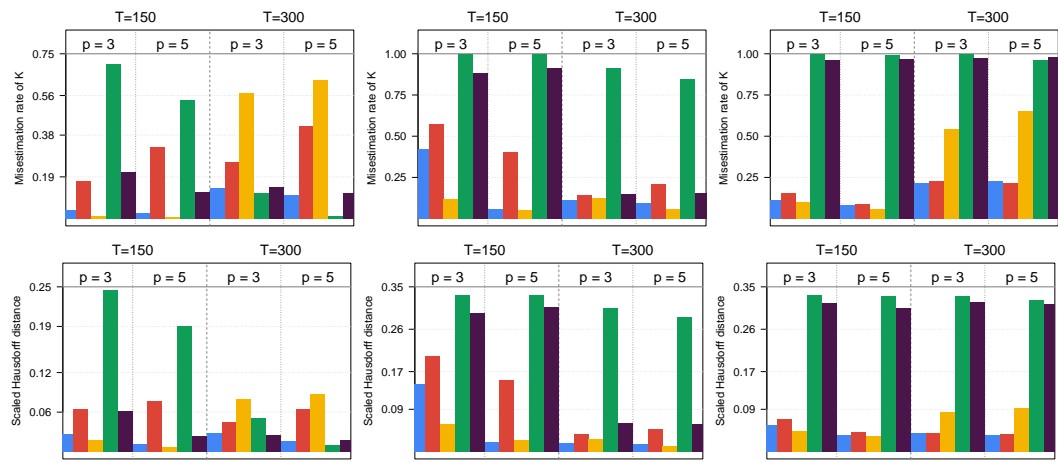

Figure 1: From top to bottom: Misestimation rate of the number of change point $K$; Scaled Hausdorff distances. From left to right: Scenarios **S1-S3**. Different colors represent different methods, ordered as MNSBS, NMP, ECP, SBS, DCBS.

Table 1: Results for change point inference.

| $n$ | $\alpha = 0.01$ | | $\alpha = 0.05$ | |
| | cover$(1-\alpha)$ | width$(1-\alpha)$ | cover$(1-\alpha)$ | width$(1-\alpha)$ |
|---|---|---|---|---|
| | | $p = 2$ | | |
| 100 | 0.864 | 17.613 (6.712) | 0.812 | 14.005 (5.639) |
| 200 | 0.904 | 22.940 (7.740) | 0.838 | 18.407 (6.541) |
| 300 | 0.993 | 26.144 (9.027) | 0.961 | 20.902 (5.936) |
| | | $p = 3$ | | |
| 100 | 0.903 | 15.439 (5.792) | 0.847 | 11.153 (4.361) |
| 200 | 0.966 | 20.108 (7.009) | 0.949 | 13.920 (5.293) |
| 300 | 0.981 | 22.395 (6.904) | 0.955 | 15.376 (4.763) |

that our localization results are robust to the bandwidth parameters, yet sensitive to the smoothness parameter $r$. We thus set $r = 1000$ in our simulations, as the density function of a multivariate normal distribution belongs to the Hölder function class with $r = \infty$. Table 1 shows that our proposed inference procedure produces good coverage in the considered setting.

## 6    Conclusion

We tackle the problem of change point detection for short range dependent multivariate non-parametric data, which has not been studied in the literature. Our two-stage algorithm MNSBS can consistently estimate the change points in stage one, a novelty in the literature. Then, we derived limiting distributions of change point estimators for inference in stage two, a first in the literature.

Our theoretical analysis reveals multiple challenging and interesting directions for future exploration. Relaxing the assumption $\Delta \asymp T$ may be of interest. In addition, in Theorem 2.**a**, we can see the limiting distribution is a function of the data-generating mechanisms, lacking universality, therefore deriving a practical method to derive the limiting distributions in the non-vanishing regime may be interesting.

## Acknowledgments and Disclosure of Funding

HX was supported by Swiss National Science Foundation Postdoc Mobility fellowship. OHMM is partially funded by NSF DMS-2015489. YY's research is partially funded by EPSRC EP/V013432/1.

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

# A  Additional simulation results

## A.1  Robustness to Kernel Selection and Bandwidth Parameters

In this subsection, we provide additional simulation results in Table 2 to show that our proposed localization method, i.e. MNSBS, is robust against both the choice of Kernel functions and the choice of bandwidth parameters.

We consider the setting with $T = 150$ and $p = 3$ of **Scenario 1**. In addition to the Gaussian kernel used in the previous numeric experiments, we consider the Epanechnikov kernel and the Triweight kernel. We also let the bandwidth $h = c_h \times (1/T)^{1/(2r+p)}$ with $r = 2$ and $c_h \in \{1, 2, 5, 10, 20\}$. Out of the 500 iterations for each case, the table below reports the proportion of the number of change points $K$ and the averaged localization errors.

Table 2: Additional localization results of **Scenario 1**.

| Kernel | $c_h = 1$ | $c_h = 2$ | $c_h = 5$ | $c_h = 10$ | $c_h = 20$ |
|---|---|---|---|---|---|
| $T = 150, p = 3$ and $r = 2$ | | | | | |
| Propotion of times $\widehat{K} \neq K$ | | | | | |
| Gaussian | 0.138 | 0.070 | 0.066 | 0.070 | 0.068 |
| Epanechnikov | 0.748 | 0.184 | 0.070 | 0.070 | 0.064 |
| Triweight | 0.280 | 0.082 | 0.068 | 0.066 | 0.060 |
| Average (standard deviation) of $d_{\mathrm{H}}$ | | | | | |
| Gaussian | 0.038(0.050) | 0.029(0.043) | 0.025(0.039) | 0.026(0.040) | 0.026(0.038) |
| Epanechnikov | 0.118(0.045) | 0.037(0.049) | 0.012(0.035) | 0.011(0.035) | 0.010(0.035) |
| Triweight | 0.053(0.057) | 0.017(0.038) | 0.010(0.034) | 0.010(0.035) | 0.010(0.034) |

## A.2  Runtime and localization for Independent Data

**Examination of Scenario 1 with independent data.**
We examined **Scenario 1** where $p = 3$, $n$ is in the set $\{150, 300\}$, and $X_t$ is i.i.d. distributed as $N(0_p, I_p)$. Our results indicate that MNSBS excels in change point localization. The refinement process further enhances its performance. Refer to Table 3 for specifics.

**Runtime Comparison.**
We benchmarked the runtime of our method against others. The tests were conducted on a machine powered by an Apple M2 chip with an 8-core CPU. The parameters were set at $p = 3$ and $n$ in the set $\{150, 300\}$ for the independent setting. Our method performs comparably at $n = 150$. However, it is slower at $n = 300$, attributed to the computational demands of CUSUM. Detailed findings are presented in Table 3.

Table 3: Runtime and localization results of **Scenario 1** with independent data.

| METHOD | $T = 150$ $p = 3$ | $T = 300$ $p = 3$ |
|---|---|---|
| AVERAGE (STANDARD DEVIATION) OF RUNTIME (IN SECONDS) | | |
| MNSBS(INITIAL & REFINED) | 1.130 (0.171) | 13.134 (2.006) |
| NMP | 0.812 (0.142) | 10.899 (1.742) |
| ECP | **0.117** (0.053) | 0.646 (0.140) |
| SBS | *0.619* (0.033) | 0.656 (0.040) |
| DCBS | 0.866 (0.092) | 1.254 (0.134) |
| PROPOTION OF TIMES $\widehat{K} \neq K$ | | |
| MNSBS(INITIAL) | **0.000** | *0.005* |
| MNSBS(REFINED) | **0.000** | *0.005* |
| NMP | **0.000** | **0.000** |
| ECP | **0.000** | 0.040 |
| SBS | 0.445 | 0.015 |
| DCBS | 0.075 | 0.065 |
| AVERAGE (STANDARD DEVIATION) OF $d_{\mathrm{H}}$ | | |
| MNSBS(INITIAL) | 0.010 (0.013) | 0.009 (0.012) |
| MNSBS(REFINED) | **0.006** (0.011) | **0.005** (0.011) |
| NMP | 0.016 (0.019) | *0.008* (0.010) |
| ECP | *0.007* (0.011) | 0.009 (0.028) |
| SBS | 0.158 (0.157) | 0.012 (0.041) |
| DCBS | 0.021 (0.037) | 0.011 (0.022) |

# B Real data application

We applied our proposed change point inference procedure to analyze stock price data[2], which consisted of daily adjusted close price of the 3 major stock market indices (S&P 500, Dow Jones and NASDAQ) from Jan-01-2021 to Jan-19-2023. After removing missing values and standardizing the raw data, the sample size was $n = 515$ and the dimension $p = 3$.

We localized 6 estimated change points and performed inference based on them; results are summarized in Table 4. We also implemented the NMP and ECP methods on the same dataset, the estimated change points being presented below. Except for the time point Aug-24-2022 estimated by ECP, all other estimated change points were located in the constructed $99\%$ confidence intervals by our proposed method.

The transformed real data is illustrated in the figure below. These data correspond to the daily adjusted close price, from Jan-01-2021 to Jan-19-2023, of the 3 major stock market indices, S&P 500, Dow Jones and NASDAQ. Moreover, in Table 4, we present the estimated change point by our proposed method MNSBS on the data before mentioned, together with their respective inference.

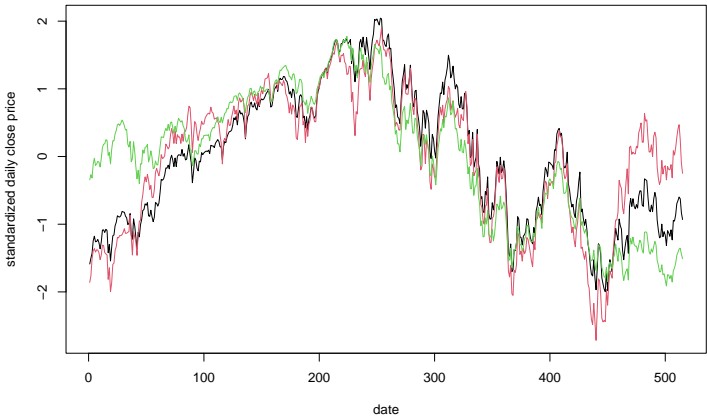

Figure 2: Plot of the standardized daily close price, from Jan-01-2021 to Jan-19-2023, of the 3 major stock market indices.

Table 4: Confidence intervals constructed for change point locations in the Real data example.

| $\widehat{\eta}$ | $\alpha = 0.01$ | | $\alpha = 0.05$ | |
| --- | --- | --- | --- | --- |
| | LOWER BOUND | UPPER BOUND | LOWER BOUND | UPPER BOUND |
| APRIL-07-2021 | APRIL-01-2021 | APRIL-12-2021 | APRIL-05-2021 | APRIL-09-2021 |
| JUNE-30-2021 | JUNE-23-2021 | JULY-09-2021 | JUNE-25-2021 | JULY-07-2021 |
| OCT-19-2021 | OCT-12-2021 | OCT-26-2021 | OCT-14-2021 | OCT-22-2021 |
| JAN-18-2022 | JAN-12-2022 | JAN-21-2022 | JAN-13-2022 | JAN-20-2022 |
| APRIL-25-2022 | APRIL-20-2022 | APRIL-28-2022 | APRIL-21-2022 | APRIL-27-2022 |
| OCT-27-2022 | OCT-24-2022 | NOV-01-2022 | OCT-25-2022 | OCT-31-2022 |

The result of the implementation of NMP and ECP methods on the same dataset are {April-01-2021, July-01-2021, Oct-19-2021, Jan-14-2022, April-21-2022, Oct-26-2022} and {April-08-2021, June-25-2021, Oct-18-2021, Jan-18-2022, April-28-2022, Aug-24-2022, Oct-27-2022} respectively.

---

[2]The stock price data are downloaded from `https://fred.stlouisfed.org/series`.

# C  $\alpha$-mixing coefficients

This paper focuses on multivariate time series that exhibit $\alpha$-mixing behavior with exponential decay coefficients. This condition is denoted as Assumption 1**e**. While the constant $2c$ is present in the exponent of the exponential function, it plays a non-essential role in our theoretical framework. We include it solely for the sake of convenience during verification.

The $\alpha$-mixing condition with exponential decay as specified in Assumption 1**e** is a commonly held assumption in time series analysis. A broad spectrum of multivariate time series satisfies this condition, including linear/nonlinear VAR models [e.g. Liebscher (2005)], a comprehensive class of GARCH models [e.g. Boussama et al. (2011)], and various Markov processes [e.g. Chan & Tong (2001)]. To further elaborate, consider the $p$ dimensional stationary VAR(1) model:

$$X_t = AX_{t-1} + \epsilon_t$$

where $A$ is the $p \times p$ transition matrix whose spectral norm satisfying $\|A\| \in (0, 1)$ and the innovations $\epsilon_t$ are i.i.d. Gaussian vectors. Denote $\Sigma = \text{cov}(X_1)$, and let $\lambda_{\max}$ and $\lambda_{\min}$ be the largest and smallest eigenvalues of $\Sigma$. Then by Theorem 3.1 in Han & Wu (2023), we have that for any $k \geq 0$, the $\alpha$-mixing coefficient of the time series $X_t$ satisfying

$$\alpha_k \leq \sqrt{\frac{\lambda_{\max}}{\lambda_{\min}}} |A|^k \leq e^{-C \log(1/|A|)k}$$

where $C > 0$ is some constant depending only on $\sqrt{\lambda_{\max}/\lambda_{\min}}$. In this example, the constant $C \log(1/|A|)$ corresponds to the constant $2c$ in Assumption 1**e**. Essentially, Assumption 1**e** is useful to unlock several technical tools under temporal dependence, which include a Bernstein's inequality Merlevède et al. (2012), a moment inequality [see Proposition 2.5 in Fan & Yao (2003)], maximal inequalities (see Section G.1) and a central limit theorem (see Section G.2). For instance, we utilize the moment inequality to bound the autocovariances of a dependence process with all lags by $\alpha$-mixing coefficients, thereby demonstrating the existence of the long-run variance, which is the sum of all the autocovariances.

# D  Proof of Theorem 1

In this section, we present the proof of theorem Theorem 1.

*Proof of Theorem 1.* For any $(s, e] \subseteq (0, T]$, let

$$\widetilde{f}_t^{(s,e]}(x) = \sqrt{\frac{e-t}{(e-s)(t-s)}} \sum_{l=s+1}^{t} f_l(x) - \sqrt{\frac{t-s}{(e-s)(e-t)}} \sum_{l=t+1}^{e} f_l(x), \ x \in \mathcal{X}.$$

For any $\widetilde{r} \in (\rho, T - \rho]$, we consider

$$\mathcal{A}((s, e], \rho, \lambda) = \left\{ \max_{t=s+\rho+1}^{e-\rho} \sup_{x \in \mathbb{R}^p} |\widetilde{F}_{t,h}^{s,e}(x) - \widetilde{f}_t^{s,e}(x)| \le \lambda \right\};$$

$$\mathcal{B}(\widetilde{r}, \rho, \lambda) = \left\{ \max_{N=\rho}^{T-\widetilde{r}} \sup_{x \in \mathbb{R}^p} \left| \frac{1}{\sqrt{N}} \sum_{t=\widetilde{r}+1}^{\widetilde{r}+N} F_{t,h}(x) - \frac{1}{\sqrt{N}} \sum_{t=\widetilde{r}+1}^{\widetilde{r}+N} f_t(x) \right| \le \lambda \right\} \bigcup$$

$$\left\{ \max_{N=\rho}^{\widetilde{r}} \sup_{x \in \mathbb{R}^p} \left| \frac{1}{\sqrt{N}} \sum_{t=\widetilde{r}-N+1}^{\widetilde{r}} F_{t,h}(x) - \frac{1}{\sqrt{N}} \sum_{t=\widetilde{r}-N+1}^{\widetilde{r}} f_t(x) \right| \le \lambda \right\}.$$

From Algorithm 1, we have that

$$\rho = \frac{\log(T)}{h^p}.$$

Therefore, Proposition 2 imply that with

$$\lambda = C_\lambda \left( 2C \sqrt{\frac{\log T}{h^p}} + \frac{2C_1 \sqrt{p}}{\sqrt{h^p}} + 2C_2 \sqrt{T} h^r. \right), \tag{9}$$

for some diverging sequence $C_\lambda$, it holds that

$$P\left\{ \mathcal{A}^c((s, e], \rho, \lambda) \right\} \lesssim \frac{1}{T^2},$$

and,

$$P\left\{ \mathcal{B}^c\left(\widetilde{r}, \rho, \frac{\lambda}{2}\right) \right\} \lesssim \frac{1}{T^2}.$$

Now, we notice that,

$$\sum_{k=1}^{\mathfrak{K}} \widetilde{n}_k = \sum_{k=1}^{\mathfrak{K}} (2^k - 1) \le \sum_{k=1}^{\mathfrak{K}} 2^k \le 2(2^{\left\lceil C_\mathfrak{K} \left( \frac{\log\left(\frac{T}{\Delta}\right)}{\log(2)} \right) \right\rceil} - 1) \le 2(2^{\left\lceil C_\mathfrak{K} \left( \frac{\log(T)}{\log(2)} \right) \right\rceil} - 1) = O(T).$$

since $2^{-x} < 1$ for any $x > 0$. In addition, there are $K = O(1)$ number of change points. In consequence, it follows that

$$P\left\{ \mathcal{A}(\mathcal{I}, \rho, \lambda) \text{ for all } \mathcal{I} \in \mathcal{J} \right\} \ge 1 - \frac{1}{T}, \tag{10}$$

$$P\left\{ \mathcal{B}(s, \rho, \lambda) \cup \mathcal{B}(e, \rho, \lambda) \text{ for all } (s, e] = \mathcal{I} \in \mathcal{J} \right\} \ge 1 - \frac{1}{T}, \tag{11}$$

$$P\left\{ \mathcal{B}(\eta_k, \rho, \lambda) \text{ for all } 1 \le k \le K \right\} \ge 1 - \frac{1}{T}. \tag{12}$$

The rest of the argument is made by assuming the events in equations (10), (11) and (12) hold. By Remark 1, we have that on these events, it is satisfied that

$$\max_{t=s+\rho+1}^{e-\rho} ||\widetilde{F}_{t,h}^{s,e}(x) - \widetilde{f}_t^{s,e}(x)||_{L_2} \le \lambda.$$

Denote

$$\Upsilon_k = C \log(T)\left(T^{\frac{p}{2r+p}}\right)\kappa_k^{-2} \quad \text{and} \quad \Upsilon_{\max} = C \log(T)\left(T^{\frac{p}{2r+p}}\right)\kappa^{-2},$$

where $\kappa = \min\{\kappa_1, \ldots, \kappa_K\}$. Since $\Upsilon_k$ is the desired localisation rate, by induction, it suffices to consider any generic interval $(s, e] \subseteq (0, T]$ that satisfies the following three conditions:

$$\eta_{m-1} \leq s \leq \eta_m \leq \ldots \leq \eta_{m+q} \leq e \leq \eta_{m+q+1}, \quad q \geq -1;$$
$$\text{either } \eta_m - s \leq \Upsilon_m \quad \text{or} \quad s - \eta_{m-1} \leq \Upsilon_{m-1};$$
$$\text{either } \eta_{m+q+1} - e \leq \Upsilon_{m+q+1} \quad \text{or} \quad e - \eta_{m+q} \leq \Upsilon_{m+q}.$$

Here $q = -1$ indicates that there is no change point contained in $(s, e]$.

Denote

$$\Delta_k = \eta_{k-1} - \eta_k \text{ for } k = 1, \ldots, K+1 \quad \text{and} \quad \Delta = \min\{\Delta_1, \ldots, \Delta_{K+1}\}.$$

Observe that by assumption 3,

$$\Upsilon_{\max} = C \log(T)\left(T^{\frac{p}{2r+p}}\right)\kappa^{-2} \leq \frac{\Delta}{4}$$

Therefore, it has to be the case that for any true change point $\eta_m \in (0, T]$, either $|\eta_m - s| \leq \Upsilon_m$ or $|\eta_m - s| \geq \Delta - \Upsilon_{\max} \geq \frac{3}{4}\Delta$. This means that $\min\{|\eta_m - e|, |\eta_m - s|\} \leq \Upsilon_m$ indicates that $\eta_m$ is a detected change point in the previous induction step, even if $\eta_m \in (s, e]$. We refer to $\eta_m \in (s, e]$ as an undetected change point if $\min\{\eta_m - s, \eta_m - e\} \geq \frac{3}{4}\Delta$. To complete the induction step, it suffices to show that $\text{MNSBS}((s, e], h, \tau)$
**(i)** will not detect any new change point in $(s, e]$ if all the change points in that interval have been previously detected, and
**(ii)** will find a point $D^{\mathcal{I}^*}$ in $(s, e]$ such that $|\eta_m - D^{\mathcal{I}^*}| \leq \Upsilon_m$ if there exists at least one undetected change point in $(s, e]$.

In order to accomplish this, we need the following series of steps.

**Step 1.** We first observe that if $\eta_k \in \{\eta_k\}_{k=1}^K$ is any change point in the functional time series, by Lemma 5, there exists a seeded interval $\mathcal{I}_k = (s_k, e_k]$ containing exactly one change point $\eta_k$ such that

$$\min\{\eta_k - s_k, e_k - \eta_k\} \geq \frac{1}{16}\Delta, \quad \text{and} \quad \max\{\eta_k - s_k, e_k - \eta_k\} \leq \frac{9}{10}\Delta$$

Even more, we notice that if $\eta_k \in (s, e]$ is any undetected change point in $(s, e]$. Then it must hold that

$$s - \eta_{k-1} \leq \Upsilon_{\max}.$$

Since $\Upsilon_{\max} = O(\log(T)T^{\frac{p}{2r+p}})$ and by assumption 3, we have that $\Upsilon_{\max} < \frac{1}{10}\Delta$. Moreover, $\eta_k - s_k \leq \frac{9}{10}(\eta_k - \eta_{k-1})$, so that it holds that

$$s_k - \eta_{k-1} \geq \frac{1}{10}(\eta_k - \eta_{k-1}) > \Upsilon_{\max} \geq s - \eta_{k-1}$$

and in consequence $s_k \geq s$. Similarly $e_k \leq e$. Therefore

$$\mathcal{I}_k = (s_k, e_k] \subseteq (s, e].$$

**Step 2.** Consider the collection of intervals $\{\mathcal{I}_k = (s_k, e_k]\}_{k=1}^K$ in **Step 1.** In this step, it is shown that for each $k \in \{1, \ldots, K\}$, it holds that

$$\max_{t=s_k+\rho}^{t=e_k-\rho} ||\widetilde{F}_{t,h}^{(s_k,e_k]}||_{L_2} \geq c_1 \sqrt{\Delta}\kappa_k, \tag{13}$$

for some sufficient small constant $c_1$.

Let $k \in \{1, \ldots, K\}$. By **Step 1**, $\mathcal{I}_k$ contains exactly one change point $\eta_k$. Since $f_t$ is a one-dimensional population time series and there is only one change point in $\mathcal{I}_k = (s_k, e_k]$, it holds that

$$f_{s_k+1} = \ldots = f_{\eta_k} \neq f_{\eta_k+1} = \ldots = f_{e_k}$$

which implies, for $s_k < t < \eta_k$

$$\widetilde{f}_t^{(s_k,e_k]} = \sqrt{\frac{e_k - t}{(e_k - s_k)(t - s_k)}} \sum_{l=s_k+1}^{t} f_{\eta_k} - \sqrt{\frac{t - s_k}{(e_k - s_k)(e_k - t)}} \sum_{l=t+1}^{\eta_k} f_{\eta_k}$$

$$- \sqrt{\frac{t - s_k}{(e_k - s_k)(e_k - t)}} \sum_{l=\eta_k+1}^{e_k} f_{\eta_k+1}$$

$$= (t - s_k)\sqrt{\frac{e_k - t}{(e_k - s_k)(t - s_k)}} f_{\eta_k} - (\eta_k - t)\sqrt{\frac{t - s_k}{(e_k - s_k)(e_k - t)}} f_{\eta_k}$$

$$- (e_k - \eta_k)\sqrt{\frac{t - s_k}{(e_k - s_k)(e_k - t)}} f_{\eta_k+1}$$

$$= \sqrt{\frac{(t - s_k)(e_k - t)}{(e_k - s_k)}} f_{\eta_k} - (\eta_k - t)\sqrt{\frac{t - s_k}{(e_k - s_k)(e_k - t)}} f_{\eta_k}$$

$$- (e_k - \eta_k)\sqrt{\frac{t - s_k}{(e_k - s_k)(e_k - t)}} f_{\eta_k+1}$$

$$= (e_k - t)\sqrt{\frac{t - s_k}{(e_k - t)(e_k - s_k)}} f_{\eta_k} - (\eta_k - t)\sqrt{\frac{t - s_k}{(e_k - s_k)(e_k - t)}} f_{\eta_k}$$

$$- (e_k - \eta_k)\sqrt{\frac{t - s_k}{(e_k - s_k)(e_k - t)}} f_{\eta_k+1}$$

$$= (e_k - \eta_k)\sqrt{\frac{t - s_k}{(e_k - t)(e_k - s_k)}} f_{\eta_k} - (e_k - \eta_k)\sqrt{\frac{t - s_k}{(e_k - s_k)(e_k - t)}} f_{\eta_k+1}$$

$$= (e_k - \eta_k)\sqrt{\frac{t - s_k}{(e_k - t)(e_k - s_k)}} (f_{\eta_k} - f_{\eta_k+1}).$$

Similarly, for $\eta_k \le t \le e_k$

$$f_t^{(s_k,e_k]} = \sqrt{\frac{e_k - t}{(e_k - s_k)(t - s_k)}} (\eta_k - s_k)(f_{\eta_k} - f_{\eta_k+1}).$$

Therefore,

$$\widetilde{f}_t^{(s_k,e_k]} = \begin{cases} \sqrt{\frac{t-s_k}{(e_k-s_k)(e_k-t)}}(e_k - \eta_k)(f_{\eta_k} - f_{\eta_k+1}), & s_k < t < \eta_k; \\ \sqrt{\frac{e_k-t}{(e_k-s_k)(t-s_k)}}(\eta_k - s_k)(f_{\eta_k} - f_{\eta_k+1}), & \eta_k \le t \le e_k. \end{cases} \tag{14}$$

Since $\rho = O(\log(T)T^{\frac{p}{2r+p}})$, by Assumption 3, we have that

$$\min\{\eta_k - s_k, e_k - \eta_k\} \ge \frac{1}{16}\Delta > \rho, \tag{15}$$

so that $\eta_k \in [s_k + \rho, e_k - \rho]$. Then, from (14), (15) and the fact that $|e_k - s_k| < \Delta$ and $|\eta_k - s_k| < \Delta$,

$$||\widetilde{f}_{\eta_k}^{(s_k,e_k]}||_{L_2} = \sqrt{\frac{e_k - \eta_k}{(e_k - s_k)(\eta_k - s_k)}}(\eta_k - s_k)||f_{\eta_k} - f_{\eta_k+1}||_{L_2} \ge c_2\sqrt{\Delta}\frac{3}{4}\kappa_k. \tag{16}$$

Therefore, it holds that

$$\max_{t=s_k+\rho}^{t=e_k-\rho} ||\widetilde{F}_{t,h}^{(s_k,e_k]}||_{L_2} \ge ||\widetilde{F}_{\eta_k,h}^{(s_k,e_k]}||_{L_2}$$

$$\ge ||\widetilde{f}_{\eta_k}^{(s_k,e_k]}||_{L_2} - \lambda$$

$$\ge c_2\frac{3}{4}\sqrt{\Delta}\kappa_k - \lambda,$$

where the first inequality follows from the fact that $\eta_k \in [s_k + \rho, e_k - \rho]$, the second inequality follows from the good event in (10) and Remark 2, and the last inequality follows from (16). Next, we observe that by Assumption 3

$$\log^{\frac{1}{2}}(T)\sqrt{\frac{1}{h^p}} = \sqrt{T^{\frac{p}{2r+p}}}\sqrt{\log(T)} \leq \frac{c_2}{4}\sqrt{\Delta}\kappa_k,$$

and,

$$\sqrt{T}h^r = T^{\frac{1}{2} - \frac{r}{2r+p}} = T^{\frac{p}{4r+2p}}.$$

In consequence, since $\kappa_k$ is a positive constant, by the upper bound of $\lambda$ on Equation (9), for sufficiently large $T$, it holds that

$$\frac{c_2}{4}\sqrt{\Delta}\kappa_k \geq \lambda.$$

Therefore,

$$\max_{t=s_k+\rho}^{t=e_k-\rho} ||\widetilde{F}_{t,h}^{(s_k,e_k)}||_{L_2} \geq \frac{c_2}{2}\sqrt{\Delta}\kappa_k.$$

Therefore Equation (13) holds with $c_1 = \frac{c_2}{2}$.

**Step 3.** In this step, it is shown that $\mathrm{SBS}((s,e],h,\tau)$ can consistently detect or reject the existence of undetected change points within $(s,e]$.

Suppose $\eta_k \in (s,e]$ is any undetected change point. Then by the second half of **Step 1**, $\mathcal{I}_k \subseteq (s,e]$, and moreover

$$a_{\mathcal{I}^*} \geq \max_{t=s_k+\rho}^{t=e_k-\rho} ||\widetilde{F}_{t,h}^{(s_k,e_k)}||_{L_2} \geq c_1\sqrt{\Delta}\kappa_k > \tau,$$

where the second inequality follows from Equation (13), and the last inequality follows from Assumption 3 and the choice of $\tau = C_\tau\left(\log^{\frac{1}{2}}(T)\sqrt{\frac{1}{h^p}}\right)$. Therefore, $\mathcal{M}^{s,e} \neq \emptyset$, since $\mathcal{I}_k \in \mathcal{M}^{s,e}$.

Suppose there does not exist any undetected change point in $(s,e]$. Then for any $\mathcal{I} = (\alpha, \beta] \subseteq (s,e]$, one of the following situations must hold,

(a) There is no change point within $(\alpha, \beta]$;

(b) there exists only one change point $\eta_k$ within $(\alpha, \beta]$ and $\min\{\eta_k - \alpha, \beta - \eta_k\} \leq \Upsilon_k$;

(c) there exist two change points $\eta_k, \eta_{k+1}$ within $(\alpha, \beta]$ and

$$\eta_k - \alpha \leq \Upsilon_k \quad \text{and} \quad \beta - \eta_{k+1} \leq \Upsilon_{k+1}.$$

Observe that if (a) holds, then we have

$$\max_{\alpha+\rho<t<\beta-\rho} ||\widetilde{F}_{t,h}^{(\alpha,\beta)}||_{L_2} \leq \max_{\alpha+\rho<t<\beta-\rho} ||\widetilde{f}_{t}^{(\alpha,\beta)}||_{L_2} + \lambda = \lambda.$$

Cases (b) and (c) can be dealt with using similar arguments. We will only work on (c) here. It follows that, in the good event in Equation (10),

$$\max_{\alpha+\rho<t<\beta-\rho} ||\widetilde{F}_{t,h}^{(\alpha,\beta)}||_{L_2} \leq \max_{\alpha<t<\beta} ||\widetilde{f}_{t}^{(\alpha,\beta)}||_{L_2} + \lambda \tag{17}$$

$$\leq \sqrt{e - \eta_k}\kappa_{k+1} + \sqrt{\eta_k - s}\kappa_k + \lambda \tag{18}$$

$$\leq 2\sqrt{C}\log^{\frac{1}{2}}(T)\sqrt{T^{\frac{p}{2r+p}}} + \lambda \tag{19}$$

where the second inequality is followed by Lemma 7. Therefore in the good event in Equation (10), for any $\mathcal{I} = (\alpha, \beta] \subseteq (s,e]$, it holds that

$$a_{\mathcal{I}} = \max_{t=\alpha+\rho}^{\beta-\rho} ||\widetilde{F}_{t,h}^{(\alpha,\beta)}||_{L_2} \leq 2\sqrt{C}\log^{\frac{1}{2}}(T)\sqrt{T^{\frac{p}{2r+p}}} + \lambda,$$

Then,

$$2\sqrt{C}\log^{\frac{1}{2}}(T)\sqrt{1 + T^{\frac{p}{2r+p}}} + \lambda$$

$$=2\sqrt{C}\log^{\frac{1}{2}}(T)\sqrt{\frac{1}{h^p} + 1} + 2C\sqrt{\frac{\log T}{h^p}} + \frac{2C_1\sqrt{p}}{\sqrt{h^p}} + 2C_2\sqrt{T}h^r.$$

We observe that $\sqrt{\frac{\log(T)}{h^p}} = O\left(\log(T)^{1/2}\sqrt{\frac{1}{h^p}}\right)$. Moreover,

$$\sqrt{T}h^r = \sqrt{T}\left(\frac{1}{T}\right)^{\frac{r}{2r+p}} \leq \left(T^{\frac{1}{2}-\frac{r}{2r+p}}\right),$$

and given that,

$$\frac{1}{2} - \frac{r}{2r+p} = \frac{p}{2(2r+p)}$$

we get,

$$\sqrt{T}h^r = o\left(\log^{\frac{1}{2}}(T)\sqrt{\frac{1}{h^p}}\right).$$

Therefore, by the choice of $\tau$, we will always correctly reject the existence of undetected change points, since

$$2\sqrt{C}\log^{\frac{1}{2}}(T)\sqrt{T^{\frac{p}{2r+p}}} + \lambda \leq \tau.$$

Thus, by the choice of $\tau$, it holds that with sufficiently large constant $C_\tau$,

$$a_{\mathcal{I}} \leq \tau \quad \text{for all} \quad \mathcal{I} \subseteq (s,e]. \tag{20}$$

As a result, MNSBS$((s,e],h,\tau)$ will correctly reject if $(s,e]$ contains no undetected change points.

**Step 4.** Assume that there exists an undetected change point $\eta_{\widetilde{k}} \in (s,e]$ such that

$$\min\{\eta_{\widetilde{k}} - s, \eta_{\widetilde{k}} - e\} = \frac{3}{4}\Delta.$$

Then, $\mathcal{M}^{s,e} \neq \emptyset$. Let $\mathcal{I}^*$ be defined as in MNSBS $((s,e],h,\tau)$ with

$$\mathcal{I}^* = (\alpha^*, \beta^*].$$

To complete the induction, it suffices to show that, there exists a change point $\eta_k \in (s,e]$ such that $\min\{\eta_k - s, \eta_k - e\} \geq \frac{3}{4}\Delta$ and $|b_{\mathcal{I}^*} - \eta_k| \leq \Upsilon_k$. To this end, we consider the collection of change points of $\{f_t\}_{t \in (\alpha^*, \beta^*]}$ We are to ensure that the assumptions of Lemma 12 are satisfied. In the following, $\lambda$ is used in Lemma 12. Then Equation (68) and Equation (69) are directly consequence of Equation (10), Equation (11), Equation (12). By the narrowest of $\mathcal{I}^*$,

$$|\mathcal{I}^*| \leq |\mathcal{I}_k| \leq \Delta,$$

and by **Step 1** with $\mathcal{I}_k = (s_k, e_k]$, it holds that

$$\min\{\eta_k - s_k, e_k - \eta_k\} \geq \frac{1}{16}\zeta_k \geq c_2\Delta,$$

Therefore for all $k \in \{\widetilde{k} : \min\{\eta_{\widetilde{k}} - s, e - \eta_{\widetilde{k}}\} \geq c_2\Delta\}$,

$$\max_{t=\alpha^*+\rho}^{t=\beta^*-\rho} \|\widetilde{F}_{t,h}^{(\alpha^*,\beta^*]}\|_{L_2} \geq \max_{t=s_k+\rho}^{t=e_k-\rho} \|\widetilde{F}_{t,h}^{(s_k,e_k]}\|_{L_2} \geq c_1\sqrt{\Delta}\kappa_k,$$

where the last inequality follows from Equation (13). Therefore (70) holds in Lemma 12. Finally, Equation (71) is a direct consequence of the choices that

$$h = C_h(T)^{\frac{-1}{2r+d}} \quad \text{and} \quad \rho = \frac{\log(T)}{nh^d}.$$

Thus, all the conditions in Lemma 12 are met. So that, there exists a change point $\eta_k$ of $\{f_t\}_{t \in \mathcal{I}^*}$, satisfying

$$\min\{\beta^* - \eta_k, \eta_k - \alpha^*\} > c\Delta, \tag{21}$$

and

$$|b_{\mathcal{I}^*} - \eta_k| \leq \max\{C_3\lambda^2\kappa_k^{-2}, \rho\} \leq C_4\log(T)\left(\frac{1}{h^p} + Th^{2r}\right)\kappa_k^{-2}$$

$$\leq C\log(T)\left(T^{\frac{p}{2r+p}}\right)\kappa_k^{-2}$$

for sufficiently large constant $C$, where we have followed the same line of arguments as for the conclusion of (20). Observe that

**i)** The change points of $\{f_t\}_{t \in \mathcal{I}^*}$ belong to $(s, e] \cap \{\eta_k\}_{k=1}^K$; and

**ii)** Equation (21) and $(\alpha^*, \beta^*] \subseteq (s, e]$ imply that

$$\min\{e - \eta_k, \eta_k - s\} > c\Delta \geq \Upsilon_{\max}.$$

As discussed in the argument before **Step 1**, this implies that $\eta_k$ must be an undetected change point of $\{f_t\}_{t \in \mathcal{I}^*}$. $\qquad \square$

# E   Proof of Theorem 2

In this section, we present the proof of theorem Theorem 2.

*Proof of Theorem 2.* **Uniform tightness** of $\kappa_k^{2+\frac{p}{r}}\left|\widetilde{\eta}_k - \eta_k\right|$. Here we show **a.1** and **b.1**. For this purpose, we will follow a series of steps. On **step 1**, we rewrite (6) in order to derive a uniform bound. **Step 2** analyses the lower bound while **Step 3** the upper bound.

**Step 1:** Denote $\widetilde{r} = \widetilde{\eta}_k - \eta_k$. Without loss of generality, suppose $\widetilde{r} \geq 0$. Since $\widetilde{\eta}_k = \eta_k + \widetilde{r}$, defined in (6), is the minimizer of $\widehat{Q}_k(\eta)$, it follows that

$$\widehat{Q}_k(\eta_k + \widetilde{r}) - \widehat{Q}_k(\eta_k) \leq 0.$$

Let

$$Q^*(\eta) = \sum_{t=s_k+1}^{\eta} ||F_{t,h_2} - f_{(s_k,\eta_k]} * \mathcal{K}_{h_2}||_{L_2}^2 + \sum_{t=\eta+1}^{e_k} ||F_{t,h_2} - f_{(\eta_k,e_k]} * \mathcal{K}_{h_2}||_{L_2}^2, \tag{22}$$

where,

$$f_{(s_k,\eta_k]} = \frac{1}{\eta_k - s_k} \sum_{i=s_k+1}^{\eta_k} f_i, \ f_{(\eta_k,e_k]} = \frac{1}{e_k - \eta_k} \sum_{i=\eta_k+1}^{e_k} f_i. \tag{23}$$

Observe that,

$$Q^*(\eta_k + \widetilde{r}) - Q^*(\eta_k) \leq \widehat{Q}_k(\eta_k) - \widehat{Q}_k(\eta_k + \widetilde{r}) - Q^*(\eta_k) + Q^*(\eta_k + \widetilde{r}). \tag{24}$$

If $\widetilde{r} \leq 1/\kappa_k^{2+\frac{p}{r}}$, then there is nothing to show. So for the rest of the argument, for contradiction, assume that

$$\widetilde{r} \geq \frac{1}{\kappa_k^{2+\frac{p}{r}}}. \tag{25}$$

**Step 2: Finding a lower bound.** In this step, we will find a lower bound of the inequality (24). To this end, we observe that,

$$Q^*(\eta_k + \widetilde{r}) - Q^*(\eta_k) = \sum_{t=\eta_k+1}^{\eta_k+\widetilde{r}} ||F_{t,h_2} - f_{(s_k,\eta_k]} * \mathcal{K}_{h_2}||_{L_2}^2 - \sum_{t=\eta_k+1}^{\eta_k+\widetilde{r}} ||F_{t,h_2} - f_{(\eta_k,e_k]} * \mathcal{K}_{h_2}||_{L_2}^2$$

$$= \sum_{t=\eta_k+1}^{\eta_k+\widetilde{r}} ||f_{(s_k,\eta_k]} * \mathcal{K}_{h_2} - f_{(\eta_k,e_k]} * \mathcal{K}_{h_2}||_{L_2}^2$$

$$- 2 \sum_{t=\eta_k+1}^{\eta_k+\widetilde{r}} \langle f_{(s_k,\eta_k]} * \mathcal{K}_{h_2} - f_{(\eta_k,e_k]} * \mathcal{K}_{h_2}, F_{t,h_2} - f_{(\eta_k,e_k]} * \mathcal{K}_{h_2} \rangle_{L_2}$$

$$= \sum_{t=\eta_k+1}^{\eta_k+\widetilde{r}} \frac{1}{2}||f_{(s_k,\eta_k]} - f_{(\eta_k,e_k]}||_{L_2}^2 - 2||f_{(s_k,\eta_k]} * \mathcal{K}_{h_2} - f_{(s_k,\eta_k]} + f_{(\eta_k,e_k]} * \mathcal{K}_{h_2} - f_{(\eta_k,e_k]}||_{L_2}^2$$

$$- 2 \sum_{t=\eta_k+1}^{\eta_k+\widetilde{r}} \langle f_{(s_k,\eta_k]} * \mathcal{K}_{h_2} - f_{(\eta_k,e_k]} * \mathcal{K}_{h_2}, F_{t,h_2} - f_{(\eta_k,e_k]} * \mathcal{K}_{h_2} \rangle_{L_2}$$

$$\geq \frac{1}{2}\widetilde{r}\kappa_k^2 - 2 \sum_{t=\eta_k+1}^{\eta_k+\widetilde{r}} ||f_{(s_k,\eta_k]} * \mathcal{K}_{h_2} - f_{(s_k,\eta_k]} + f_{(\eta_k,e_k]} * \mathcal{K}_{h_2} - f_{(\eta_k,e_k]}||_{L_2}^2$$

$$- 2 \sum_{t=\eta_k+1}^{\eta_k+\widetilde{r}} \langle f_{(s_k,\eta_k]} * \mathcal{K}_{h_2} - f_{(\eta_k,e_k]} * \mathcal{K}_{h_2}, F_{t,h_2} - f_{(\eta_k,e_k]} * \mathcal{K}_{h_2} \rangle_{L_2}$$

We consider,

$$I_1 := 2 \sum_{t=\eta_k+1}^{\eta_k+\widetilde{r}} ||f_{(s_k,\eta_k]} * \mathcal{K}_{h_2} - f_{(s_k,\eta_k]} + f_{(\eta_k,e_k]} * \mathcal{K}_{h_2} - f_{(\eta_k,e_k]}||_{L_2}^2, \text{ and,}$$

$$I_2 := 2 \sum_{t=\eta_k+1}^{\eta_k+\widetilde{r}} \langle f_{(s_k,\eta_k]} * \mathcal{K}_{h_2} - f_{(\eta_k,e_k]} * \mathcal{K}_{h_2}, F_{t,h_2} - f_{(\eta_k,e_k]} * \mathcal{K}_{h_2} \rangle_{L_2}.$$

From above, we have that,

$$Q^*(\eta_k + \widetilde{r}) - Q^*(\eta_k) \geq \frac{1}{2}\widetilde{r}\kappa_k^2 - I_1 - I_2.$$

We now analyze the order of magnitude of term $I_1$. Then, we get a lower bound for the term $-I_1$. In fact $I_1$, has an upper bound of the form $o_p(\widetilde{r}k^{\frac{p}{r}+2})$, where we use that $||f_{\eta_k} * \mathcal{K}_{h_2} - f_{\eta_k}||_{L_2} = o(1)$ and $||f_{\eta_{k+1}} * \mathcal{K}_{h_2} - f_{\eta_{k+1}}||_{L_2} = o(1)$. For the term $I_2$, we consider the random variable,

$$Y_i = \frac{\langle f_{[s_k+1,\eta_k]} * \mathcal{K}_{h_2} - f_{[\eta_k+1,e_k]} * \mathcal{K}_{h_2}, F_{t,h_2} - f_{[\eta_k+1,e_k]} * \mathcal{K}_{h_2} \rangle_{L_2}}{\kappa_k \mathbb{E}(||F_{t,h_2} - f_{\eta_{k+1}} * \mathcal{K}_{h_2}||_{L^2}^3)^{1/3}}.$$

In order to use Lemma 3, we need to bound $\mathbb{E}(|Y_i|^3)$. For this, first we use Cauchy Schwartz inequality,

$$\mathbb{E}(|Y_i|^3) \leq \frac{(||(f_{\eta_k+1} - f_{\eta_k}) * \mathcal{K}_{h_2}||_{L^2})^3 \mathbb{E}(||F_{t,h_2} - f_{\eta_{k+1}} * \mathcal{K}_{h_2}||_{L^2}^3)}{\kappa_k^3 \mathbb{E}(||F_{t,h_2} - f_{\eta_{k+1}} * \mathcal{K}_{h_2}||_{L^2}^3)}$$

then, by Minkowski's inequality,

$$\begin{aligned}
||(f_{\eta_k+1} - f_{\eta_k}) * \mathcal{K}_{h_2}||_{L^2} &= \left|\left| \int_{\mathbb{R}^p} (f_{\eta_k+1} - f_{\eta_k})(\cdot - y)\mathcal{K}_{h_2}(y)dy \right|\right|_{L_2} \\
&\leq \int_{\mathbb{R}^p} \left|\left| (f_{\eta_k+1} - f_{\eta_k})(\cdot - y)\mathcal{K}_{h_2}(y) \right|\right|_{L_2} dy \\
&= \left( \int_{\mathbb{R}^p} |\mathcal{K}_{h_2}(y)|dy \right) \left|\left| (f_{\eta_k+1} - f_{\eta_k})(\cdot - y) \right|\right|_{L_2} \\
&= ||f_{\eta_k+1} - f_{\eta_k}||_{L^2} ||\mathcal{K}_{h_2}||_{L^1}.
\end{aligned}$$

Therefore, by Assumption 2, we have

$$\begin{aligned}
&\frac{(||(f_{\eta_k+1} - f_{\eta_k}) * \mathcal{K}_{h_2}||_{L^2})^3 \mathbb{E}(||F_{t,h_2} - f_{\eta_{k+1}} * \mathcal{K}_{h_2}||_{L^2}^3)}{\kappa_k^3 \mathbb{E}(||F_{t,h_2} - f_{\eta_{k+1}} * \mathcal{K}_{h_2}||_{L^2}^3)} \\
&\leq \frac{(||f_{\eta_k+1} - f_{\eta_k}||_{L^2} ||\mathcal{K}_{h_2}||_{L^1})^3 \mathbb{E}(||F_{t,h_2} - f_{\eta_{k+1}} * \mathcal{K}_{h_2}||_{L^2}^3)}{\kappa_k^3 \mathbb{E}(||F_{t,h_2} - f_{\eta_{k+1}} * \mathcal{K}_{h_2}||_{L^2}^3)} \\
&\leq C_K.
\end{aligned}$$

for any $t \in (\eta_k, e_k]$. Moreover, we have that

$$\begin{aligned}
\mathbb{E}(||F_{t,h_2} - f_{\eta_{k+1}} * \mathcal{K}_{h_2}||_{L^2}^3)^{\frac{1}{3}} &= \left( \int \left( \int (\mathcal{K}_{h_2}(x-z) - \mathbb{E}(\mathcal{K}_{h_2}(x-X_t)))^2 dx \right)^{\frac{3}{2}} f_t(z)dz \right)^{1/3} \\
&\leq \left( \int \left( \int (\mathcal{K}_{h_2}(x-z))^2 dx \right)^{\frac{3}{2}} f_t(z)dz \right)^{\frac{1}{3}} \\
&= \frac{1}{\kappa^{p/2r}}.
\end{aligned}$$
(26)

Therefore, by Lemma 3, we have that $I_2 = o_p\left( \sqrt{\widetilde{r}}\kappa_k \kappa_k^{-\frac{p}{2r}} (\log(\widetilde{r}\kappa_k^{\frac{p}{r}+2}) + 1) \right)$. Thus,

$$Q^*(\eta_k + \widetilde{r}) - Q^*(\eta_k) \geq \frac{1}{2}\widetilde{r}\kappa_k^2 - O_p\left( \sqrt{\widetilde{r}}\kappa_k \kappa_k^{-\frac{p}{2r}} (\log(\widetilde{r}\kappa_k^{\frac{p}{r}+2}) + 1) \right) - o_p(\widetilde{r}\kappa_k^{\frac{p}{r}+2}). \quad (27)$$

**Step 3: Finding an upper bound.** Now, we proceeded to get an upper bound of (24). This is, an upper bound of the following expression,

$$\widehat{Q}_k(\eta_k) - \widehat{Q}_k(\eta_k + \widetilde{r}) - Q^*(\eta_k) + Q^*(\eta_k + \widetilde{r}). \quad (28)$$

Observe that, this expression can be written as,

$$\widehat{Q}_k(\eta_k) - \widehat{Q}_k(\eta_k + \widetilde{r}) - Q^*(\eta_k) + Q^*(\eta_k + \widetilde{r})$$

$$= -\sum_{t=\eta_k+1}^{\eta_k+\widetilde{r}} ||F_{t,h_1} - F_{(s_k,\eta_k],h_1}||^2_{L_2} + \sum_{t=\eta_k+1}^{\eta_k+\widetilde{r}} ||F_{t,h_1} - F_{(\widehat{\eta}_k,e_k],h_1}||^2_{L_2}$$

$$+ \sum_{t=\eta_k+1}^{\eta_k+\widetilde{r}} ||F_{t,h_2} - f_{(s_k,\eta_k]} * \mathcal{K}_{h_2}||^2_{L_2} - \sum_{t=\eta_k+1}^{\eta_k+\widetilde{r}} ||F_{t,h_2} - f_{(\widehat{\eta}_k,e_k]} * \mathcal{K}_{h_2}||^2_{L_2}$$

So that,

$$\widehat{Q}_k(\eta_k) - \widehat{Q}_k(\eta_k + \widetilde{r}) - Q^*(\eta_k) + Q^*(\eta_k + \widetilde{r}) = U_1 + U_2,$$

where,

$$U_1 = \sum_{t=\eta_k+1}^{\eta_k+\widetilde{r}} ||F_{t,h_2} - f_{(s_k,\eta_k]} * \mathcal{K}_{h_2}||^2_{L_2} - \sum_{t=\eta_k+1}^{\eta_k+\widetilde{r}} ||F_{t,h_1} - F_{(s_k,\widehat{\eta}_k],h_1}||^2_{L_2}, \text{ and,}$$

$$U_2 = \sum_{t=\eta_k+1}^{\eta_k+\widetilde{r}} ||F_{t,h_1} - F_{(\widehat{\eta}_k,e_k],h_1}||^2_{L_2} - \sum_{t=\eta_k+1}^{\eta_k+\widetilde{r}} ||F_{t,h_2} - f_{(\eta_k,e_k]} * \mathcal{K}_{h_2}||^2_{L_2}.$$

Now, we analyze each of the terms above. For $U_1$, observe that

$$\sum_{t=\eta_k+1}^{\eta_k+\widetilde{r}} ||F_{t,h_2} - f_{(s_k,\eta_k]} * \mathcal{K}_{h_2}||^2_{L_2} - \sum_{t=\eta_k+1}^{\eta_k+\widetilde{r}} ||F_{t,h_1} - F_{(s_k,\widehat{\eta}_k],h_1}||^2_{L_2}$$

$$= \sum_{t=\eta_k+1}^{\eta_k+\widetilde{r}} ||F_{t,h_2} - f_{(s_k,\eta_k]} * \mathcal{K}_{h_2}||^2_{L_2} - \sum_{t=\eta_k+1}^{\eta_k+\widetilde{r}} ||F_{t,h_2} - F_{(s_k,\eta_k],h_2}||^2_{L_2}$$

$$+ \sum_{t=\eta_k+1}^{\eta_k+\widetilde{r}} ||F_{t,h_2} - F_{(s_k,\eta_k],h_2}||^2_{L_2} - \sum_{t=\eta_k+1}^{\eta_k+\widetilde{r}} ||F_{t,h_1} - F_{(s_k,\eta_k],h_1}||^2_{L_2}$$

$$= I_3 + I_4,$$

where,

$$I_3 = \sum_{t=\eta_k+1}^{\eta_k+\widetilde{r}} ||F_{t,h_2} - f_{(s_k,\eta_k]} * \mathcal{K}_{h_2}||^2_{L_2} - \sum_{t=\eta_k+1}^{\eta_k+\widetilde{r}} ||F_{t,h_2} - F_{(s_k,\eta_k],h_2}||^2_{L_2}, \text{ and,}$$

$$I_4 = \sum_{t=\eta_k+1}^{\eta_k+\widetilde{r}} ||F_{t,h_2} - F_{(s_k,\eta_k],h_2}||^2_{L_2} - \sum_{t=\eta_k+1}^{\eta_k+\widetilde{r}} ||F_{t,h_1} - F_{(s_k,\widehat{\eta}_k],h_1}||^2_{L_2}.$$

To analyze $I_3$, we rewrite it as follow,

$$I_3 = \sum_{t=\eta_k+1}^{\eta_k+\widetilde{r}} ||f_{(s_k,\eta_k]} * \mathcal{K}_{h_2} - F_{(s_k,\eta_k],h_2}||^2_{L_2} - 2\sum_{t=\eta_k+1}^{\eta_k+\widetilde{r}} \langle f_{(s_k,\eta_k]} * \mathcal{K}_{h_2} - F_{(s_k,\eta_k],h_2}, F_{t,h_2} - f_{(s_k,\eta_k]} * \mathcal{K}_{h_2}\rangle_{L_2}$$

$$= I_{3,1} + I_{3,2},$$

where,

$$I_{3,1} = \sum_{t=\eta_k+1}^{\eta_k+\widetilde{r}} ||f_{(s_k,\eta_k]} * \mathcal{K}_{h_2} - F_{(s_k,\eta_k],h_2}||^2_{L_2}, \text{ and,}$$

$$I_{3,2} = -2\sum_{t=\eta_k+1}^{\eta_k+\widetilde{r}} \langle f_{(s_k,\eta_k]} * \mathcal{K}_{h_2} - F_{(s_k,\eta_k],h_2}, F_{t,h_2} - f_{(s_k,\eta_k]} * \mathcal{K}_{h_2}\rangle_{L_2}.$$

Now, we will get an upper bound for each of the terms above. The term $I_{3,1} = O_p\left(\widetilde{r}\frac{1}{T}\frac{\log(T)}{\kappa_k^{\frac{p}{r}}}\right)$, which is followed by the use of Remark 1 . Even more, by Assumption 4, we get

$$I_{3,1} = o_p(\widetilde{r}\kappa_k^{\frac{p}{r}+2}). \tag{29}$$

For the term $I_{3,2}$, by Cauchy Schwartz inequality and triangle inequality,

$$\langle f_{(s_k,\eta_k]} * \mathcal{K}_\kappa - F_{(s_k,\eta_k],\kappa}, F_{t,h_2} - f_{(s_k,\eta_k]} * \mathcal{K}_{h_2}\rangle_{L_2}$$
$$\leq ||f_{(s_k,\eta_k]} * \mathcal{K}_{h_2} - F_{(s_k,\eta_k],h_2}||_{L_2}||F_{t,h_2} - f_{(s_k,\eta_k]} * \mathcal{K}_{h_2}||_{L_2}$$
$$\leq ||f_{(s_k,\eta_k]} * \mathcal{K}_{h_2} - F_{(s_k,\eta_k],h_2}||_{L_2}\left(||F_{t,h_2} - f_{[\eta_k+1,e_k]} * \mathcal{K}_{h_2}||_{L_2} + ||f_{[\eta_k+1,e_k]} * \mathcal{K}_{h_2} - f_{[s_k+1,\eta_k]} * \mathcal{K}_{h_2}||_{L_2}\right)$$

for any $t \in (\eta_k, \eta_k + \widetilde{r}]$. By the Remark 1, we have that

$$||f_{(s_k,\eta_k]} * \mathcal{K}_{h_2} - F_{(s_k,\eta_k],h_2}||_{L_2} = O_p\left(\frac{1}{\sqrt{T}}\sqrt{\frac{\log(T)}{\kappa_k^{\frac{p}{r}}}}\right)$$

and using basic properties of integrals $||f_{[\eta_k+1,e_k]} * \mathcal{K}_{h_2} - f_{[s_k+1,\eta_k]} * \mathcal{K}_{h_2}||_{L_2} = O(\kappa_k)$. Therefore,

$$I_{3,2} \leq O_p\left(\frac{1}{\sqrt{T}}\sqrt{\frac{\log(T)}{\kappa_k^{\frac{p}{r}}}}\right)\left(O(\widetilde{r}\kappa_k) + \sum_{t=\eta_k+1}^{\eta_k+\widetilde{r}}||F_{t,h_2} - f_{[\eta_k+1,e_k]} * \mathcal{K}_{h_2}||_{L_2}\right)$$

Now, we need to get a bound of the magnitude of

$$\sum_{t=\eta_k+1}^{\eta_k+\widetilde{r}}||F_{t,h_2} - f_{[\eta_k+1,e_k]} * \mathcal{K}_{h_2}||_{L_2},$$

in order to get an upper for $I_{3,2}$. This is done similarly to $I_2$. We consider the random variable

$$\widetilde{Y}_i = \frac{\langle F_{t,h_2} - f_{(\eta_k,e_k]} * \mathcal{K}_{h_2}, F_{t,h_2} - f_{(\eta_k,e_k]} * \mathcal{K}_{h_2}\rangle_{L_2}^{\frac{1}{2}} - \mathbb{E}(||F_{t,h_2} - f_{\eta_{k+1}} * \mathcal{K}_{h_2}||_{L_2})}{\mathbb{E}(||F_{t,h_2} - f_{\eta_{k+1}} * \mathcal{K}_{h_2}||_{L_2}^3)^{\frac{1}{3}}}.$$

In order to use Lemma 3, we observe that since $||F_{t,h_2} - f_{\eta_{k+1}} * \mathcal{K}_{h_2}||_{L^2} \geq 0$,

$$\mathbb{E}(|\widetilde{Y}_i|^3) \leq \frac{\mathbb{E}(||F_{t,h_2} - f_{\eta_{k+1}} * \mathcal{K}_{h_2}||_{L^2}^3)}{\mathbb{E}(||F_{t,h_2} - f_{\eta_{k+1}} * \mathcal{K}_{h_2}||_{L^2}^3)} = 1.$$

Therefore, using Lemma 3 and that $\mathbb{E}(||F_{t,h_2} - f_{\eta_{k+1}} * \mathcal{K}_{h_2}||_{L^2}) = O(\kappa_k^{\frac{-p}{2r}})$ by (26), we get that

$$\sum_{t=\eta_k+1}^{\eta_k+\widetilde{r}}||F_{t,h_2} - f_{[\eta_k+1,e_k]} * \mathcal{K}_{h_2}||_{L_2}^2 = O_p(\sqrt{\widetilde{r}\kappa_k^{-\frac{p}{r}}}(\log(\widetilde{r}\kappa_k^{\frac{p}{r}+2}) + 1)) + O_p(\widetilde{r}\kappa_k^{\frac{-p}{2r}}).$$

Thus, by Assumption 4 and above,

$$I_{3,2} \leq O_p\left(\frac{1}{\sqrt{T}}\sqrt{\frac{\log(T)}{\kappa_k^{\frac{p}{r}}}}\right)\left(O(\widetilde{r}\kappa_k) + O_p(\sqrt{\widetilde{r}\kappa_k^{-\frac{p}{r}}}(\log(\widetilde{r}\kappa_k^{\frac{p}{r}+2}) + 1)) + O_p(\widetilde{r}\kappa_k^{\frac{-p}{2r}})\right) = o_p(\widetilde{r}\kappa_k^{\frac{p}{r}+2}). \tag{30}$$

Consequently, $I_3$ has been bounded, and we only need to go over the term $I_4$, to finalize the analysis for $U_1$. To analyze $I_4$, we observe that

$$I_4 = \sum_{t=\eta_k+1}^{\eta_k+\widetilde{r}}||F_{t,h_2} - F_{(s_k,\eta_k],h_2}||_{L_2}^2 - \sum_{t=\eta_k+1}^{\eta_k+\widetilde{r}}||F_{t,h_1} - F_{(s_k,\widehat{\eta}_k],h_1}||_{L_2}^2$$

$$= \sum_{t=\eta_k+1}^{\eta_k+\widetilde{r}}\left[\langle F_{t,h_2}, F_{t,h_2}\rangle_{L_2} - 2\langle F_{t,h_2}, F_{(s_k,\eta_k],h_2}\rangle_{L_2} + \langle F_{(s_k,\eta_k],h_2}, F_{(s_k,\eta_k],h_2}\rangle_{L_2}\right]$$

$$+ \sum_{t=\eta_k+1}^{\eta_k+\widetilde{r}}\left[-\langle F_{t,h_1}, F_{t,h_1}\rangle_{L_2} + 2\langle F_{t,h_1}, F_{(s_k,\widehat{\eta}_k],h_1}\rangle_{L_2} - \langle F_{(s_k,\widehat{\eta}_k],h_1}, F_{(s_k,\widehat{\eta}_k],h_1}\rangle_{L_2}\right]$$

$$= I_{4,1} + I_{4,2} + I_{4,3},$$

where,

$$I_{4,1} = \sum_{t=\eta_k+1}^{\eta_k+\widetilde{r}} \langle F_{t,h_2}, F_{t,h_2} \rangle_{L_2} - \langle F_{t,h_1}, F_{t,h_1} \rangle_{L_2}$$

$$I_{4,2} = \sum_{t=\eta_k+1}^{\eta_k+\widetilde{r}} 2\langle F_{t,h_1}, F_{(s_k,\widehat{\eta}_k],h_1} \rangle_{L_2} - 2\langle F_{t,h_2}, F_{(s_k,\eta_k],h_2} \rangle_{L_2}, \text{ and,}$$

$$I_{4,3} = \sum_{t=\eta_k+1}^{\eta_k+\widetilde{r}} \langle F_{(s_k,\eta_k],h_2}, F_{(s_k,\eta_k],h_2} \rangle_{L_2} - \langle F_{(s_k,\widehat{\eta}_k],h_1}, F_{(s_k,\widehat{\eta}_k],h_1} \rangle_{L_2}.$$

Now, we explore each of the terms $I_{4,1}$, $I_{4,2}$, and $I_{4,3}$. First, $I_{4,1}$ can be bounded as follows, we add and subtract $\langle F_{t,h_1}, F_{t,h_2} \rangle_{L_2}$, to get

$$\sum_{t=\eta_k+1}^{\eta_k+\widetilde{r}} \langle F_{t,h_2}, F_{t,h_2} \rangle_{L_2} - \langle F_{t,h_1}, F_{t,h_1} \rangle_{L_2}$$

$$= \sum_{t=\eta_k+1}^{\eta_k+\widetilde{r}} \langle F_{t,h_2}, F_{t,h_2} \rangle_{L_2} - \langle F_{t,h_1}, F_{t,h_1} \rangle_{L_2} + \langle F_{t,h_1}, F_{t,h_2} \rangle_{L_2} - \langle F_{t,h_1}, F_{t,h_2} \rangle_{L_2}$$

$$= \sum_{t=\eta_k+1}^{\eta_k+\widetilde{r}} \langle F_{t,h_2} - F_{t,h_1}, F_{t,h_2} \rangle_{L_2} + \langle F_{t,h_1}, F_{t,h_2} - F_{t,h_1} \rangle_{L_2}$$

which, by Hölder's inequality, is bounded by

$$\sum_{t=\eta_k+1}^{\eta_k+\widetilde{r}} ||F_{t,h_2}||_{L_2}||F_{t,h_2} - F_{t,h_1}||_{L_2} + ||F_{t,h_1}||_{L_2}||F_{t,h_2} - F_{t,h_1}||_{L_2} = \widetilde{r}O_p\left(\frac{T^{-\frac{r}{2r+p}}}{\kappa_k^{\frac{p}{2r}+\frac{1}{2}+\frac{p}{2r}}} \log^{\frac{r}{2r+p}}(T)\right))$$

since $||F_{t,h_1} - F_{t,h_2}||_{L_2} = O\left(\frac{|\kappa-\widehat{\kappa}|^{\frac{1}{2}}}{\kappa_k^{\frac{p}{2r}+\frac{1}{2}}}\right) = O_p\left(\frac{1}{\kappa_k^{\frac{p}{2r}+\frac{1}{2}}}\left(\left(\frac{\log(T)}{\Delta}\right)^{\frac{2r}{2r+p}} + \frac{T^{\frac{p}{2r+p}}\log(T)}{\kappa\Delta}\right)^{\frac{1}{2}}\right)$, for
any $t$, see Remark 2 for more detail. Similarly, for $I_{4,2}$, we have that adding and subtracting $2\langle F_{t,h_1}, F_{(s_k,\eta_k],h_2} \rangle_{L_2}$,

$$\sum_{t=\eta_k+1}^{\eta_k+\widetilde{r}} 2\langle F_{t,h_1}, F_{(s_k,\widehat{\eta}_k],h_1} \rangle_{L_2} - 2\langle F_{t,h_2}, F_{(s_k,\eta_k],h_2} \rangle_{L_2}$$

$$= \sum_{t=\eta_k+1}^{\eta_k+\widetilde{r}} 2\langle F_{t,h_1}, F_{(s_k,\widehat{\eta}_k],h_1} \rangle_{L_2} - 2\langle F_{t,h_2}, F_{(s_k,\eta_k],h_2} \rangle_{L_2} + 2\langle F_{t,h_1}, F_{(s_k,\eta_k],h_2} \rangle_{L_2} - 2\langle F_{t,h_1}, F_{(s_k,\eta_k],h_2} \rangle_{L_2}$$

$$= \sum_{t=\eta_k+1}^{\eta_k+\widetilde{r}} 2\langle F_{t,h_1} - F_{t,h_2}, F_{(s_k,\eta_k],h_2} \rangle_{L_2} + 2\langle F_{t,h_1}, F_{(s_k,\widehat{\eta}_k],h_1} - F_{(s_k,\eta_k],h_2} \rangle_{L_2},$$

and by Hölder's inequality and Remark 2, it is bounded by

$$\sum_{t=\eta_k+1}^{\eta_k+\widetilde{r}} ||F_{t,h_1} - F_{t,h_2}||_{L_2}||F_{(s_k,\eta_k],h_2}||_{L_2} + ||F_{t,h_1}||_{L_2}||F_{(s_k,\eta_k],h_2} - F_{(s_k,\widehat{\eta}_k],h_1}||_{L_2}$$

$$= \widetilde{r}O_p\left(\frac{1}{\kappa_k^{\frac{p}{2r}+\frac{1}{2}+\frac{p}{2r}}}\left(\left(\frac{\log(T)}{\Delta}\right)^{\frac{2r}{2r+p}} + \frac{T^{\frac{p}{2r+p}}\log(T)}{\kappa\Delta}\right)^{\frac{1}{2}} + \frac{T^{\frac{p}{2r+p}}\log(T)}{\kappa^2}\right)\right).$$

Finally, for $I_{4,3}$, we notice that, adding and subtracting $\langle F_{(s_k,\widehat{\eta}_k],h_1}, F_{(s_k,\eta_k],h_2}\rangle_{L_2}$, it is written as,

$$\sum_{t=\eta_k+1}^{\eta_k+\widetilde{r}} \langle F_{(s_k,\eta_k],h_2}, F_{(s_k,\eta_k],h_2}\rangle_{L_2} - \langle F_{(s_k,\eta_k],h_1}, F_{(s_k,\eta_k],h_1}\rangle_{L_2}$$

$$= \sum_{t=\eta_k+1}^{\eta_k+\widetilde{r}} \langle F_{(s_k,\eta_k],h_2}, F_{(s_k,\eta_k],h_2}\rangle_{L_2} - \langle F_{(s_k,\widehat{\eta}_k],h_1}, F_{(s_k,\widehat{\eta}_k],h_1}\rangle_{L_2}$$

$$+ \langle F_{(s_k,\widehat{\eta}_k],h_1}, F_{(s_k,\eta_k],h_2}\rangle_{L_2} - \langle F_{(s_k,\widehat{\eta}_k],h_1}, F_{(s_k,\eta_k],h_2}\rangle_{L_2}$$

$$= \sum_{t=\eta_k+1}^{\eta_k+\widetilde{r}} \langle F_{(s_k,\eta_k],h_2} - F_{(s_k,\widehat{\eta}_k],h_1}, F_{(s_k,\eta_k],h_2}\rangle_{L_2} + \langle F_{(s_k,\widehat{\eta}_k],h_1}, F_{(s_k,\eta_k],h_2} - F_{(s_k,\widehat{\eta}_k],h_1}\rangle_{L_2}$$

which, by Hölder's inequality and Remark 2, is bounded by

$$\sum_{t=\eta_k+1}^{\eta_k+\widetilde{r}} ||F_{(s_k,\eta_k],h_2}||_{L_2}||F_{(s_k,\widehat{\eta}_k],h_1} - F_{(s_k,\eta_k],h_2}||_{L_2} + ||F_{(s_k,\widehat{\eta}_k],h_1}||_{L_2}||F_{(s_k,\eta_k],h_2} - F_{(s_k,\widehat{\eta}_k],h_1}||_{L_2}$$

$$= \widetilde{r} O_p\Big( \frac{1}{\kappa_k^{\frac{p}{2r}+\frac{1}{2}+\frac{p}{2r}}} \Big( \Big( \frac{\log(T)}{\Delta} \Big)^{\frac{2r}{2r+p}} + \frac{T^{\frac{p}{2r+p}}\log(T)}{\kappa\Delta} \Big)^{\frac{1}{2}} \Big)$$

Then, by above and Assumption 4, we conclude

$$I_4 = o_p(\widetilde{r}\kappa_k^{\frac{p}{r}+2}). \tag{31}$$

From (29), (30) and (31), we find that $U_1$ has the following upper bound,

$$\sum_{t=\eta_k+1}^{\eta_k+\widetilde{r}} ||F_{t,h_2} - f_{(s_k,\eta_k]} * \mathcal{K}_{h_2}||_{L_2}^2 - \sum_{t=\eta_k+1}^{\eta_k+\widetilde{r}} ||F_{t,h_1} - F_{(s_k,\widehat{\eta}_k],h_1}||_{L_2}^2 = o_p(\widetilde{r}\kappa_k^{\frac{p}{r}+2}). \tag{32}$$

Now, making an analogous analysis, we have that $U_2$ is upper bounded by,

$$\sum_{t=\eta_k+1}^{\eta_k+\widetilde{r}} ||F_{t,h_1} - F_{(\widehat{\eta}_k,e_k],h_1}||_{L_2}^2 - \sum_{t=\eta_k+1}^{\eta_k+\widetilde{r}} ||F_{t,h_2} - f_{(\eta_k,e_k]} * \mathcal{K}_{h_2}||_{L_2}^2 = o_p(\widetilde{r}\kappa_k^{\frac{p}{r}+2}). \tag{33}$$

In fact, we observe that

$$\sum_{t=\eta_k+1}^{\eta_k+\widetilde{r}} ||F_{t,h_1} - F_{(\widehat{\eta}_k,e_k],h_1}||_{L_2}^2 - \sum_{t=\eta_k+1}^{\eta_k+\widetilde{r}} ||F_{t,h_2} - f_{(\eta_k,e_k]} * \mathcal{K}_{h_2}||_{L_2}^2$$

$$= \sum_{t=\eta_k+1}^{\eta_k+\widetilde{r}} ||F_{t,h_2} - F_{(\eta_k,e_k],h_2}||_{L_2}^2 - \sum_{t=\eta_k+1}^{\eta_k+\widetilde{r}} ||F_{t,h_2} - f_{(\eta_k,e_k]} * \mathcal{K}_{h_2}||_{L_2}^2$$

$$+ \sum_{t=\eta_k+1}^{\eta_k+\widetilde{r}} ||F_{t,h_1} - F_{(\widehat{\eta}_k,e_k],h_1}||_{L_2}^2 - \sum_{t=\eta_k+1}^{\eta_k+\widetilde{r}} ||F_{t,h_2} - F_{(\eta_k,e_k],h_2}||_{L_2}^2$$

$$= I_5 + I_6,$$

where,

$$I_5 = \sum_{t=\eta_k+1}^{\eta_k+\widetilde{r}} ||F_{t,h_2} - F_{(\eta_k,e_k],h_2}||_{L_2}^2 - \sum_{t=\eta_k+1}^{\eta_k+\widetilde{r}} ||F_{t,h_2} - f_{(\eta_k,e_k]} * \mathcal{K}_{h_2}||_{L_2}^2, \text{ and,}$$

$$I_6 = \sum_{t=\eta_k+1}^{\eta_k+\widetilde{r}} ||F_{t,h_1} - F_{(\widehat{\eta}_k,e_k],h_1}||_{L_2}^2 - \sum_{t=\eta_k+1}^{\eta_k+\widetilde{r}} ||F_{t,h_2} - F_{(\eta_k,e_k],h_2}||_{L_2}^2.$$

Then, $I_5$ is bounded as follows

$$I_5 = \sum_{t=\eta_k+1}^{\eta_k+\widetilde{r}} ||f_{(\eta_k,e_k]} * \mathcal{K}_{h_2} - F_{(\eta_k,e_k],h_2}||_{L_2}^2 + 2 \sum_{t=\eta_k+1}^{\eta_k+\widetilde{r}} \langle f_{(\eta_k,e_k]} * \mathcal{K}_{h_2} - F_{(\eta_k,e_k],h_2}, F_{t,h_2} - f_{(\eta_k,e_k]} * \mathcal{K}_{h_2}\rangle_{L_2}$$

where,

$$I_{5,1} = \sum_{t=\eta_k+1}^{\eta_k+\widetilde{r}} ||f_{(\eta_k,e_k]} * \mathcal{K}_{h_2} - F_{(\eta_k,e_k],h_2}||_{L_2}^2, \text{ and,}$$

$$I_{5,2} = 2 \sum_{t=\eta_k+1}^{\eta_k+\widetilde{r}} \langle f_{(\eta_k,e_k]} * \mathcal{K}_{h_2} - F_{(\eta_k,e_k],h_2}, F_{t,h_2} - f_{(\eta_k,e_k]} * \mathcal{K}_{h_2} \rangle_{L_2}.$$

The term $I_{5,1} = O_p\left(\widetilde{r}\frac{1}{T}\frac{\log(T)}{\kappa_k^{\frac{p}{r}}}\right)$, using Remark 1. Even more, by Assumption 4, we get

$$I_{5,1} = o_p(\widetilde{r}\kappa_k^{\frac{p}{r}+2}). \tag{34}$$

For the term $I_{5,2}$, by Cauchy Schwartz inequality,

$$\langle f_{(\eta_k,e_k]} * \mathcal{K}_{h_2} - F_{(\eta_k,e_k],h_2}, F_{t,h_2} - f_{(\eta_k,e_k]} * \mathcal{K}_{h_2} \rangle_{L_2}$$
$$\leq ||f_{(\eta_k,e_k]} * \mathcal{K}_{h_2} - F_{(\eta_k,e_k],h_2}||_{L_2}||F_{t,h_2} - f_{(\eta_k,e_k]} * \mathcal{K}_{h_2}||_{L_2}$$

for any $t \in (\eta_k, \eta_k + \widetilde{r}]$. By Remark 1, we have that $||f_{(\eta_k,e_k]} * \mathcal{K}_{h_2} - F_{(\eta_k,e_k],h_2}||_{L_2} = O_p\left(\frac{1}{\sqrt{T}}\sqrt{\frac{\log(T)}{\kappa_k^{\frac{p}{r}}}}\right)$. Therefore,

$$I_{5,2} \leq O_p\left(\frac{1}{\sqrt{T}}\sqrt{\frac{\log(T)}{\kappa_k^{\frac{p}{r}}}}\right)\left(\sum_{t=\eta_k+1}^{\eta_k+\widetilde{r}} ||F_{t,h_2} - f_{[\eta_k+1,e_k]} * \mathcal{K}_{h_2}||_{L_2}\right)$$

Now, similarly to the bound for $I_2$, we consider the random variable

$$\bar{Y}_i = \frac{\langle F_{t,h_2} - f_{(\eta_k,e_k]} * \mathcal{K}_{\kappa}, F_{t,h_2} - f_{(\eta_k,e_k]} * \mathcal{K}_{h_2} \rangle_{L_2}^{\frac{1}{2}} - \mathbb{E}(||F_{t,h_2} - f_{[\eta_k+1,e_k]} * \mathcal{K}_{h_2}||_{L_2})}{\mathbb{E}(||F_{t,h_2} - f_{\eta_{k+1}} * \mathcal{K}_{h_2}||_{L_2}^3)^{\frac{1}{3}}}.$$

In order to use Lemma 3, we observe

$$\mathbb{E}(|\bar{Y}_i|^3) = \frac{\mathbb{E}(||F_{t,h_2} - f_{\eta_{k+1}} * \mathcal{K}_{h_2}||_{L_2}^3)}{\mathbb{E}(||F_{t,h_2} - f_{\eta_{k+1}} * \mathcal{K}_{h_2}||_{L_2}^3)} = 1.$$

so that, by Lemma 3,

$$I_{5,2} \leq O_p\left(\frac{1}{\sqrt{T}}\sqrt{\frac{\log(T)}{\kappa_k^{\frac{p}{r}}}}\right)\left(O_p(\sqrt{\widetilde{r}\kappa_k^{-\frac{p}{r}}}(\log(\widetilde{r}\kappa_k^{\frac{p}{r}+2}) + 1)) + O_p(\kappa_k^{\frac{-p}{2r}})\right) = o_p(\widetilde{r}\kappa_k^{\frac{p}{r}+2}). \tag{35}$$

To analyze $I_6$, we observe that

$$I_6 = \sum_{t=\eta_k+1}^{\eta_k+\widetilde{r}} ||F_{t,h_2} - F_{(\eta_k,e_k],h_2}||_{L_2}^2 - \sum_{t=\eta_k+1}^{\eta_k+\widetilde{r}} ||F_{t,h_1} - F_{(\widehat{\eta}_k,e_k],h_1}||_{L_2}^2$$

$$= \sum_{t=\eta_k+1}^{\eta_k+\widetilde{r}} \left[\langle F_{t,h_2}, F_{t,h_2}\rangle_{L_2} - 2\langle F_{t,h_2}, F_{(\eta_k,e_k],h_2}\rangle_{L_2} + \langle F_{(\eta_k,e_k],h_2}, F_{(\eta_k,e_k],h_2}\rangle_{L_2}\right]$$

$$\sum_{t=\eta_k+1}^{\eta_k+\widetilde{r}} \left[-\langle F_{t,h_1}, F_{t,h_1}\rangle_{L_2} + 2\langle F_{t,h_1}, F_{(\widehat{\eta}_k,e_k],h_1}\rangle_{L_2} - \langle F_{(\widehat{\eta}_k,e_k],h_1}, F_{(\widehat{\eta}_k,e_k],h_1}\rangle_{L_2}\right]$$

$$= I_{6,1} + I_{6,2} + I_{6,3},$$

where,

$$I_{6,1} = \sum_{t=\eta_k+1}^{\eta_k+\widetilde{r}} \langle F_{t,h_2}, F_{t,h_2}\rangle_{L_2} - \langle F_{t,h_1}, F_{t,h_1}\rangle_{L_2},$$

$$I_{6,2} = \sum_{t=\eta_k+1}^{\eta_k+\widetilde{r}} 2\langle F_{t,h_1}, F_{(\widehat{\eta}_k,e_k],h_1}\rangle_{L_2} - 2\langle F_{t,h_2}, F_{(\eta_k,e_k],h_2}\rangle_{L_2}$$

$$I_{6,3} = \sum_{t=\eta_k+1}^{\eta_k+\widetilde{r}} \langle F_{(\eta_k,e_k],h_2}, F_{(\eta_k,e_k],h_2}\rangle_{L_2} - \langle F_{(\widehat{\eta}_k,e_k],h_1}, F_{(\widehat{\eta}_k,e_k],h_1}\rangle_{L_2}.$$

Then we bound each of these terms. First, we rewrite $I_{6,1}$, as

$$\sum_{t=\eta_k+1}^{\eta_k+\widetilde{r}} \langle F_{t,h_2}, F_{t,h_2}\rangle_{L_2} - \langle F_{t,h_1}, F_{t,h_1}\rangle_{L_2}$$

$$= \sum_{t=\eta_k+1}^{\eta_k+\widetilde{r}} \langle F_{t,h_2}, F_{t,h_2}\rangle_{L_2} - \langle F_{t,h_1}, F_{t,h_1}\rangle_{L_2} + \langle F_{t,h_1}, F_{t,h_2}\rangle_{L_2} - \langle F_{t,h_1}, F_{t,h_2}\rangle_{L_2}$$

$$= \sum_{t=\eta_k+1}^{\eta_k+\widetilde{r}} \langle F_{t,h_2} - F_{t,h_1}, F_{t,h_2}\rangle_{L_2} + \langle F_{t,h_1}, F_{t,h_2} - F_{t,h_1}\rangle_{L_2}$$

which, by Hölder's inequality, is bounded by

$$\sum_{t=\eta_k+1}^{\eta_k+\widetilde{r}} ||F_{t,h_2}||_{L_2}||F_{t,h_2} - F_{t,h_1}||_{L_2} + ||F_{t,h_1}||_{L_2}||F_{t,h_2} - F_{t,h_1}||_{L_2}$$

$$= \widetilde{r}O_p\Big(\frac{1}{\kappa_k^{\frac{p}{2r}+\frac{1}{2}+\frac{p}{2r}}}\Big)\Big(\frac{\log(T)}{\Delta}\Big)^{\frac{2r}{2r+p}} + \frac{T^{\frac{p}{2r+p}}\log(T)}{\kappa\Delta}\Big)^{\frac{1}{2}}\Big)$$

since $||F_{t,\kappa} - F_{t,\widehat{\kappa}}||_{L_2}^2 = O(\frac{|\kappa-\widehat{\kappa}|^{\frac{1}{2}}}{\kappa_k^{\frac{p}{2r}+\frac{1}{2}}}) = O_p(\frac{1}{\kappa_k^{\frac{p}{2r}+\frac{1}{2}}}\Big(\frac{\log(T)}{\Delta}\Big)^{\frac{2r}{2r+p}} + \frac{T^{\frac{p}{2r+p}}\log(T)}{\kappa\Delta}\Big)^{\frac{1}{2}}))$, for any $t$, see
Remark 2 for more detail. Similarly, for $I_{6,2}$ we have,

$$\sum_{t=\eta_k+1}^{\eta_k+\widetilde{r}} 2\langle F_{t,h_1}, F_{(\widehat{\eta}_k,e_k],h_1}\rangle_{L_2} - 2\langle F_{t,h_2}, F_{(\eta_k,e_k],h_2}\rangle_{L_2}$$

$$= \sum_{t=\eta_k+1}^{\eta_k+\widetilde{r}} 2\langle F_{t,h_1}, F_{(\widehat{\eta}_k,e_k],h_1}\rangle_{L_2} - 2\langle F_{t,h_2}, F_{(\eta_k,e_k],h_2}\rangle_{L_2} + 2\langle F_{t,h_1}, F_{(\eta_k,e_k],h_2}\rangle_{L_2} - 2\langle F_{t,h_1}, F_{(\eta_k,e_k],h_2}\rangle_{L_2}$$

$$= \sum_{t=\eta_k+1}^{\eta_k+\widetilde{r}} 2\langle F_{t,h_1} - F_{t,h_2}, F_{(\eta_k,e_k],h_2}\rangle_{L_2} + 2\langle F_{t,h_1}, F_{(\eta_k,e_k],h_2} - F_{(\widehat{\eta}_k,e_k],h_1}\rangle_{L_2}$$

and by Hölder's inequality and Remark 2, it is bounded by

$$\sum_{t=\eta_k+1}^{\eta_k+\widetilde{r}} ||F_{t,h_1} - F_{t,h_2}||_{L_2}||F_{(\eta_k,e_k],h_2}||_{L_2} + ||F_{t,h_1}||_{L_2}||F_{(\eta_k,e_k],h_2} - F_{(\widehat{\eta}_k,e_k],h_1}||_{L_2}$$

$$= \widetilde{r}O_p\Big(\frac{1}{\kappa_k^{\frac{p}{2r}+\frac{1}{2}+\frac{p}{2r}}}\Big(\Big(\frac{\log(T)}{\Delta}\Big)^{\frac{2r}{2r+p}} + \frac{T^{\frac{p}{2r+p}}\log(T)}{\kappa\Delta}\Big)^{\frac{1}{2}}\Big)\Big).$$

Now for $I_{6,3}$, we write it as

$$\sum_{t=\eta_k+1}^{\eta_k+\widetilde{r}} \langle F_{(\eta_k,e_k],h_2}, F_{(\eta_k,e_k],h_2}\rangle_{L_2} - \langle F_{(\widehat{\eta}_k,e_k],h_1}, F_{(\widehat{\eta}_k,e_k],h_1}\rangle_{L_2}$$

$$= \sum_{t=\eta_k+1}^{\eta_k+\widetilde{r}} \langle F_{(\eta_k,e_k],h_2}, F_{(\eta_k,e_k],h_2}\rangle_{L_2} - \langle F_{(\widehat{\eta}_k,e_k],h_1}, F_{(\widehat{\eta}_k,e_k],h_1}\rangle_{L_2} + \langle F_{(\widehat{\eta}_k,e_k],h_1}, F_{(\eta_k,e_k],h_2}\rangle_{L_2} - \langle F_{(\widehat{\eta}_k,e_k],h_1}, F_{(\eta_k,e_k],h_2}\rangle_{L_2}$$

$$= \sum_{t=\eta_k+1}^{\eta_k+\widetilde{r}} \langle F_{(\eta_k,e_k],h_2} - F_{(\widehat{\eta}_k,e_k],h_1}, F_{(s_k,\eta_k],h_2}\rangle_{L_2} + \langle F_{(\widehat{\eta}_k,e_k],h_1}, F_{(\eta_k,e_k],h_2} - F_{(\widehat{\eta}_k,e_k],h_1}\rangle_{L_2}$$

which, by Hölder's inequality and Remark 2, is bounded by

$$\sum_{t=\eta_k+1}^{\eta_k+\widetilde{r}} ||F_{(\eta_k,e_k],h_2}||_{L_2}||F_{(\eta_k,e_k],h_2} - F_{(\widehat{\eta}_k,e_k],h_1}||_{L_2} + ||F_{(\widehat{\eta}_k,e_k],h_1}||_{L_2}||F_{(\eta_k,e_k],h_2} - F_{(\widehat{\eta}_k,e_k],h_1}||_{L_2}$$

$$= \widetilde{r}O_p\Big(\frac{1}{\kappa_k^{\frac{p}{2r}+\frac{1}{2}+\frac{p}{2r}}}\Big(\Big(\frac{\log(T)}{\Delta}\Big)^{\frac{2r}{2r+p}} + \frac{T^{\frac{p}{2r+p}}\log(T)}{\kappa\Delta}\Big)^{\frac{1}{2}}\Big)\Big)$$

By above and Assumption 4, we conclude

$$I_6 = o_p(\widetilde{r}\kappa_k^{\frac{p}{r}+2}). \tag{36}$$

From, (34), (35) and (36), we get that $U_2$ is bounded by

$$\sum_{t=\eta_k+1}^{\eta_k+\widetilde{r}} ||F_{t,\widehat{\kappa}} - F_{(\widehat{\eta}_k,e_k],\widehat{\kappa}}||_{L_2}^2 - \sum_{t=\eta_k+1}^{\eta_k+\widetilde{r}} ||F_{t,\kappa} - f_{(\eta_k,e_k]} * \mathcal{K}_\kappa||_{L_2}^2 = o_p(\widetilde{r}\kappa_k^{\frac{p}{r}+2})$$

Therefore, from (32) and (33)

$$\widehat{Q}_k(\eta_k) - \widehat{Q}_k(\eta_k + \widetilde{r}) - Q^*(\eta_k) + Q^*(\eta_k + \widetilde{r}) = o_p(\widetilde{r}\kappa_k^{\frac{p}{r}+2}) \tag{37}$$

**Step 4: Combination of all the steps above.** Finally, combining (24), (27) and (37), uniformly for any $\widetilde{r} \geq \frac{1}{\kappa_k^{\frac{p}{r}+2}}$ we have that

$$\frac{1}{2}\widetilde{r}\kappa_k^2 - O_p\left(\sqrt{\widetilde{r}}\kappa_k\kappa_k^{-\frac{p}{2r}}(\log(\widetilde{r}\kappa_k^{\frac{p}{r}+2}) + 1)\right) - o_p(\widetilde{r}\kappa_k^{\frac{p}{r}+2}) \leq o_p(\widetilde{r}\kappa_k^{\frac{p}{r}+2})$$

which implies,

$$\widetilde{r}\kappa_k^{\frac{p}{r}+2} = O_p(1) \tag{38}$$

and complete the proofs of **a.1** and **b.1**.

**Limiting distributions.** For any $k \in \{1, \ldots, K\}$, due to the uniform tightness of $\widetilde{r}\kappa_k^{\frac{p}{r}+2}$, (24) and (37), as $T \to \infty$

$$Q^*(\eta) = \sum_{t=s_k+1}^{\eta} ||F_{t,h_2} - f_{(s_k,\eta_k]} * \mathcal{K}_{h_2}||_{L_2}^2 + \sum_{t=\eta+1}^{e_k} ||F_{t,h_2} - f_{(\eta_k,e_k]} * \mathcal{K}_{h_2}||_{L_2}^2,$$

satisfies

$$\left|\widehat{Q}\left(\eta_k + \widetilde{r}\right) - \widehat{Q}\left(\eta_k\right) - \left(Q^*\left(\eta_k + \widetilde{r}\right) - Q^*\left(\eta_k\right)\right)\right| \xrightarrow{p} 0.$$

Therefore, it is sufficient to find the limiting distributions of $Q^*\left(\eta_k + \widetilde{r}\right) - Q^*\left(\eta_k\right)$ when $T \to \infty$.

**Non-vanishing regime.** Observe that for $\widetilde{r} > 0$, we have that when $T \to \infty$,

$$Q^*(\eta_k + \widetilde{r}) - Q^*(\eta_k) = \sum_{t=\eta_k+1}^{\eta_k+\widetilde{r}} ||F_{t,h_2} - f_{(s_k,\eta_k]} * \mathcal{K}_{h_2}||_{L_2}^2 - \sum_{t=\eta_k+1}^{\eta_k+\widetilde{r}} ||F_{t,h_2} - f_{(\eta_k,e_k]} * \mathcal{K}_{h_2}||_{L_2}^2$$

$$= \sum_{t=\eta_k+1}^{\eta_k+\widetilde{r}} ||f_{(s_k,\eta_k]} * \mathcal{K}_{h_2} - f_{(\eta_k,e_k]} * \mathcal{K}_{h_2}||_{L_2}^2$$

$$- 2\sum_{t=\eta_k+1}^{\eta_k+\widetilde{r}} \langle f_{(s_k,\eta_k]} * \mathcal{K}_{h_2} - f_{(\eta_k,e_k]} * \mathcal{K}_{h_2}, F_{t,h_2} - f_{(\eta_k,e_k]} * \mathcal{K}_{h_2}\rangle_{L_2}$$

$$\xrightarrow{\mathcal{D}} \sum_{t=1}^{\widetilde{r}} 2\left\langle F_{h_2,\eta_k+t} - f_{\eta_k+t} * \mathcal{K}_{h_2}, (f_{\eta_{k+1}} - f_{\eta_k}) * \mathcal{K}_{h_2}\right\rangle_{L_2} + \widetilde{r}||(f_{\eta_{k+1}} - f_{\eta_k}) * \mathcal{K}_{h_2}||_{L_2}^2.$$

When $\widetilde{r} < 0$ and $T \to \infty$, we have that

$$Q^*(\eta_k + \widetilde{r}) - Q^*(\eta_k) = \sum_{t=\eta_k+\widetilde{r}}^{\eta_k-1} ||F_{t,h_2} - f_{(s_k,\eta_k]} * \mathcal{K}_{h_2}||_{L_2}^2 - \sum_{t=\eta_k+\widetilde{r}}^{\eta_k-1} ||F_{t,h_2} - f_{(\eta_k,e_k]} * \mathcal{K}_{h_2}||_{L_2}^2$$

$$= \sum_{t=\eta_k+\widetilde{r}}^{\eta_k-1} ||f_{(s_k,\eta_k]} * \mathcal{K}_{h_2} - f_{(\eta_k,e_k]} * \mathcal{K}_{h_2}||_{L_2}^2$$

$$- 2\sum_{t=\eta_k+\widetilde{r}}^{\eta_k-1} \langle f_{(s_k,\eta_k]} * \mathcal{K}_{h_2} - f_{(\eta_k,e_k]} * \mathcal{K}_{h_2}, F_{t,h_2} - f_{(\eta_k,e_k]} * \mathcal{K}_{h_2}\rangle_{L_2}$$

$$\xrightarrow{\mathcal{D}} \sum_{t=\widetilde{r}+1}^{0} 2\left\langle F_{h_2,\eta_k+t} - f_{\eta_k+t} * \mathcal{K}_{h_2}, (f_{\eta_k} - f_{\eta_{k+1}}) * \mathcal{K}_{h_2}\right\rangle_{L_2} + \widetilde{r}||(f_{\eta_{k+1}} - f_{\eta_k}) * \mathcal{K}_{h_2}||_{L_2}^2.$$

Therefore, using Slutsky's theorem and the Argmax (or Argmin) continuous mapping theorem (see 3.2.2 Theorem van der Vaart and Wellner, 1996) we conclude

$$(\widetilde{\eta}_k - \eta_k)\kappa_k^{\frac{p}{r}+2} \xrightarrow{\mathcal{D}} \operatorname*{arg\,min}_{\widetilde{r} \in \mathbb{Z}} P_k(\widetilde{r}) \tag{39}$$

**Vanishing regime.** Vanishing regime. Let $m = \kappa_k^{-2-\frac{p}{r}}$, and we have that $m \to \infty$ as $T \to \infty$. Observe that for $\widetilde{r} > 0$, we have that

$$Q_k^*\left(\eta_k + \widetilde{r}m\right) - Q_k^*\left(\eta_k\right) = \sum_{t=\eta_k}^{\eta_k+\widetilde{r}m-1} ||f_{(s_k,\eta_k]} * \mathcal{K}_{h_2} - f_{(\eta_k,e_k]} * \mathcal{K}_{h_2}||_{L_2}^2$$

$$-2\sum_{t=\eta_k}^{\eta_k+\widetilde{r}m-1} \langle f_{(s_k,\eta_k]} * \mathcal{K}_{h_2} - f_{(\eta_k,e_k]} * \mathcal{K}_{h_2}, F_{t,h_2} - f_{(\eta_k,e_k]} * \mathcal{K}_{h_2}\rangle_{L_2}$$

Following the Central Limit Theorem for $\alpha-$mixing, see Lemma 4, we get

$$\frac{1}{\sqrt{m}}\sum_{t=\eta_k}^{\eta_k+rm-1} \frac{\langle f_{(s_k,\eta_k]} * \mathcal{K}_{h_2} - f_{(\eta_k,e_k]} * \mathcal{K}_{h_2}, F_{t,h_2} - f_{(\eta_k,e_k]} * \mathcal{K}_{h_2}\rangle_{L_2}}{\kappa_k^{\frac{p}{2r}+1}} \xrightarrow{\mathcal{D}} \kappa_k^{-\frac{p}{r}}\widetilde{\sigma}_\infty(k)\mathbb{B}(\widetilde{r}),$$

where $\mathbb{B}(\widetilde{r})$ is a standard Brownian motion and $\widetilde{\sigma}(k)$ is the long-run variance given in (8). Therefore, it holds that when $T \to \infty$

$$Q_k^*\left(\eta_k + \widetilde{r}m\right) - Q_k^*\left(\eta_k\right) \xrightarrow{\mathcal{D}} \kappa_k^{-\frac{p}{r}}\widetilde{\sigma}_\infty(k)\mathbb{B}_1(r) + \widetilde{r}\kappa_k^{-\frac{p}{r}-2}||f_{(s_k,\eta_k]} * \mathcal{K}_{h_2} - f_{(\eta_k,e_k]} * \mathcal{K}_{h_2}||_{L_2}^2.$$

Similarly, for $\widetilde{r} < 0$, we have that when $n \to \infty$

$$Q_k^*\left(\eta_k + rm\right) - Q_k^*\left(\eta_k\right) \xrightarrow{\mathcal{D}} \kappa_k^{-\frac{p}{r}}\widetilde{\sigma}_\infty(k)\mathbb{B}_1(-\widetilde{r}) - \widetilde{r}\kappa_k^{-\frac{p}{r}-2}||f_{(s_k,\eta_k]} * \mathcal{K}_{h_2} - f_{(\eta_k,e_k]} * \mathcal{K}_{h_2}||_{L_2}^2..$$

Then, using Slutsky's theorem and the Argmax (or Argmin) continuous mapping theorem (see 3.2.2 Theorem in van der Vaart & Wellner (1996)), and the fact that, $\mathbb{E}(||f_{(s_k,\eta_k]} * \mathcal{K}_{h_2} - f_{(\eta_k,e_k]} * \mathcal{K}_{h_2}||_{L_2}^2) = O(\kappa_k^2)$, we conclude that

$$\kappa_k^{2+\frac{p}{r}}\left(\widetilde{\eta}_k - \eta_k\right) \xrightarrow{\mathcal{D}} \operatorname*{arg\,min}_{r \in \mathbb{Z}} \widetilde{\sigma}_\infty(k)B(\widetilde{r}) + |\widetilde{r}|,$$

which completes the proof of **b.2**. $\qquad\square$

# F   Proof of Theorem 3

In this section, we present the proof of theorem Theorem 3.

*Proof of Theorem 3.* First, letting $h_2 = c_\kappa \kappa_k^{\frac{1}{r}}$ and $R = O(\frac{T^{\frac{2r}{2r+p}}}{\kappa_k^{\frac{p}{2r}+\frac{3}{2}}})$, we consider

$$\check{\sigma}_\infty^2(k) = \frac{1}{R} \sum_{r=1}^R \left( \frac{1}{\sqrt{S}} \sum_{i \in \mathcal{S}_r} \check{Y}_i \right)^2, \text{ where, } \check{Y}_i = \kappa_k^{\frac{p}{2r}-1} \left\langle F_{h_2,i} - f_i * \mathcal{K}_{h_2}, (f_{\eta_k} - f_{\eta_{k+1}}) * \mathcal{K}_{h_2} \right\rangle_{L_2}. \tag{40}$$

We will show that

(i) $\left| \widehat{\sigma}_\infty^2(k) - \check{\sigma}_\infty^2(k) \right| \xrightarrow{P} 0, \quad T \to \infty,$ and

(ii) $\left| \check{\sigma}_\infty^2(k) - \widetilde{\sigma}_\infty^2(k) \right| \xrightarrow{P} 0, \quad T \to \infty$

in order to conclude the result. For (i), we use $a^2 - b^2 = (a+b)(a-b)$, to write,

$$\left| \widehat{\sigma}_\infty^2(k) - \check{\sigma}_\infty^2(k) \right| = \left| \frac{1}{R} \sum_{r=1}^R \left( \frac{1}{\sqrt{S}} \sum_{i \in \mathcal{S}_r} \check{Y}_i \right)^2 - \frac{1}{R} \sum_{r=1}^R \left( \frac{1}{\sqrt{S}} \sum_{i \in \mathcal{S}_r} Y_i \right)^2 \right|$$

$$= \left| \frac{1}{R} \sum_{r=1}^R \left( \frac{1}{\sqrt{S}} \sum_{i \in \mathcal{S}_r} \check{Y}_i - Y_i \right) \left( \frac{1}{\sqrt{S}} \sum_{i \in \mathcal{S}_r} \check{Y}_i + Y_i \right) \right|$$

$$= \left| \frac{1}{R} \sum_{r=1}^R I_1 I_2 \right|$$

Then, we bound each of the terms $I_1$ and $I_2$. For $I_1$, we observe that,

$$I_1 = \left| \frac{1}{\sqrt{S}} \sum_{i \in \mathcal{S}_r} \check{Y}_i - Y_i \right| \le \frac{1}{\sqrt{S}} \sum_{i \in \mathcal{S}_r} \left| \check{Y}_i - Y_i \right|.$$

Then, adding and subtracting, $\widehat{\kappa}_k^{\frac{p}{2r}-1} \left\langle F_{h_1,i} - f_i * \mathcal{K}_{h_1}, (f_{\eta_k} - f_{\eta_{k+1}}) * \mathcal{K}_{h_2} \right\rangle_{L_2}$ and

$$\widehat{\kappa}_k^{\frac{p}{2r}-1} \left\langle F_{h_2,i} - f_i * \mathcal{K}_{h_2} - F_{h_1,i} + f_i * \mathcal{K}_{h_1}, (f_{\widehat{\eta}_k} - f_{\widehat{\eta}_{k+1}}) * \mathcal{K}_{h_1} \right\rangle_{L_2},$$

we get that,

$$\left| \check{Y}_i - Y_i \right|$$

$$= \left| \kappa_k^{\frac{p}{2r}-1} \left\langle F_{h_2,i} - f_i * \mathcal{K}_{h_2}, (f_{\eta_k} - f_{\eta_{k+1}}) * \mathcal{K}_{h_2} \right\rangle_{L_2} - \widehat{\kappa}_k^{\frac{p}{2r}-1} \left\langle F_{h_1,i} - f_i * \mathcal{K}_{h_1}, (f_{\widehat{\eta}_k} - f_{\widehat{\eta}_{k+1}}) * \mathcal{K}_{h_1} \right\rangle_{L_2} \right|$$

$$= \left| \kappa_k^{\frac{p}{2r}-1} \left\langle F_{h_2,i} - f_i * \mathcal{K}_{h_2}, (f_{\eta_k} - f_{\eta_{k+1}}) * \mathcal{K}_{h_2} \right\rangle_{L_2} - \widehat{\kappa}_k^{\frac{p}{2r}-1} \left\langle F_{h_1,i} - f_i * \mathcal{K}_{h_1}, (f_{\eta_k} - f_{\eta_{k+1}}) * \mathcal{K}_{h_2} \right\rangle_{L_2} \right.$$

$$+ \widehat{\kappa}_k^{\frac{p}{2r}-1} \left\langle F_{h_1,i} - f_i * \mathcal{K}_{h_1}, (f_{\eta_k} - f_{\eta_{k+1}}) * \mathcal{K}_{h_2} \right\rangle_{L_2} - \widehat{\kappa}_k^{\frac{p}{2r}-1} \left\langle F_{h_1,i} - f_i * \mathcal{K}_{h_1}, (f_{\widehat{\eta}_k} - f_{\widehat{\eta}_{k+1}}) * \mathcal{K}_{h_1} \right\rangle_{L_2} \right|$$

$$+ \widehat{\kappa}_k^{\frac{p}{2r}-1} \left\langle F_{h_2,i} - f_i * \mathcal{K}_{h_2} - F_{h_1,i} + f_i * \mathcal{K}_{h_1}, (f_{\widehat{\eta}_k} - f_{\widehat{\eta}_{k+1}}) * \mathcal{K}_{h_1} \right\rangle_{L_2}$$

$$- \widehat{\kappa}_k^{\frac{p}{2r}-1} \left\langle F_{h_2,i} - f_i * \mathcal{K}_{h_2} - F_{h_1,i} + f_i * \mathcal{K}_{h_1}, (f_{\widehat{\eta}_k} - f_{\widehat{\eta}_{k+1}}) * \mathcal{K}_{h_1} \right\rangle_{L_2}$$

which can be written as,

$$\left| \left\langle \kappa_k^{\frac{p}{2r}-1}(F_{h_2,i} - f_i * \mathcal{K}_{h_2}) - \widehat{\kappa}_k^{\frac{p}{2r}-1}(F_{h_1,i} - f_i * \mathcal{K}_{h_1}), (f_{\eta_k} - f_{\eta_{k+1}}) * \mathcal{K}_{h_2} - (f_{\widehat{\eta}_k} - f_{\widehat{\eta}_{k+1}}) * \mathcal{K}_{h_1} \right\rangle_{L_2} \right.$$

$$+ \left\langle \widehat{\kappa}_k^{\frac{p}{2r}-1}(F_{h_1,i} - f_i * \mathcal{K}_{h_1}) - \kappa_k^{\frac{p}{2r}-1}(F_{h_2,i} - f_i * \mathcal{K}_{h_2}), (f_{\widehat{\eta}_k} - f_{\widehat{\eta}_{k+1}}) * \mathcal{K}_{h_1} - (f_{\eta_k} - f_{\eta_{k+1}}) * \mathcal{K}_{h_2} \right\rangle_{L_2}$$

$$+ \left\langle \kappa_k^{\frac{p}{2r}-1}(F_{h_2,i} - f_i * \mathcal{K}_{h_2}), (f_{\eta_k} - f_{\eta_{k+1}}) * \mathcal{K}_{h_2} - (f_{\widehat{\eta}_k} - f_{\widehat{\eta}_{k+1}}) * \mathcal{K}_{h_1} \right\rangle_{L_2}$$

$$+ \left. \left\langle \widehat{\kappa}_k^{\frac{p}{2r}-1}(F_{h_1,i} - f_i * \mathcal{K}_{h_1}) - \kappa_k^{\frac{p}{2r}-1}(F_{h_2,i} - f_i * \mathcal{K}_{h_2}), (f_{\widehat{\eta}_k} - f_{\widehat{\eta}_{k+1}}) * \mathcal{K}_{h_1} \right\rangle_{L_2} \right|.$$

Now, we bound the expression above. For this purpose, by triangle inequality, it is enough to bound each of the terms above. Then, we use Hölder's inequality. First,

$$\left| \left\langle \kappa_k^{\frac{p}{2r}-1}(F_{h_2,i} - f_i * \mathcal{K}_{h_2}) - \widehat{\kappa}_k^{\frac{p}{2r}-1}(F_{h_1,i} - f_i * \mathcal{K}_{h_1}), (f_{\eta_k} - f_{\eta_{k+1}}) * \mathcal{K}_{h_2} - (f_{\widehat{\eta}_k} - f_{\widehat{\eta}_{k+1}}) * \mathcal{K}_{h_1} \right\rangle_{L_2} \right|$$

$$\leq |\kappa_k^{\frac{p}{2r}-1} - \widehat{\kappa}_k^{\frac{p}{2r}-1}| \, ||F_{h_2,i} - f_i * \mathcal{K}_{h_2} - F_{h_1,i} + f_i * \mathcal{K}_{h_1}||_{L_2} ||(f_{\eta_k} - f_{\eta_{k+1}}) * \mathcal{K}_{h_2} - (f_{\widehat{\eta}_k} - f_{\widehat{\eta}_{k+1}}) * \mathcal{K}_{h_1}||_{L_2}.$$

Then, using (54), we have that $|\kappa_k^{\frac{p}{2r}-1} - \widehat{\kappa}_k^{\frac{p}{2r}-1}| = O_p\left( \frac{1}{\kappa_k^{2-\frac{p}{2r}}} \left( \left( \frac{\log(T)}{\Delta} \right)^{\frac{2r}{2r+p}} + \frac{T^{\frac{p}{2r+p}} \log(T)}{\kappa \Delta} \right)^{\frac{1}{2}} \right)$,

and using Remark 2, it follows that

$$||F_{h_2,i} - f_i * \mathcal{K}_{h_2} - F_{h_1,i} + f_i * \mathcal{K}_{h_1}||_{L_2} \leq ||F_{h_2,i} - F_{h_1,i}||_{L_2} + ||f_i * \mathcal{K}_{h_1} - f_i * \mathcal{K}_{h_2}||_{L_2}$$

$$= O_p\left( \frac{1}{\kappa_k^{\frac{p}{2r}+\frac{1}{2}}} \left( \left( \frac{\log(T)}{\Delta} \right)^{\frac{2r}{2r+p}} + \frac{T^{\frac{p}{2r+p}} \log(T)}{\kappa \Delta} \right)^{\frac{1}{2}} \right)$$

and,

$$||(f_{\eta_k} - f_{\eta_{k+1}}) * \mathcal{K}_{h_2} - (f_{\widehat{\eta}_k} - f_{\widehat{\eta}_{k+1}}) * \mathcal{K}_{h_1}||_{L_2}$$

$$\leq ||f_{\eta_k} * \mathcal{K}_{h_2} - f_{\widehat{\eta}_k} * \mathcal{K}_{h_1}||_{L_2} + ||f_{\widehat{\eta}_{k+1}} * \mathcal{K}_{h_1} - f_{\eta_{k+1}} * \mathcal{K}_{h_2}||_{L_2}$$

$$= O_p\left( \frac{1}{\kappa_k^{\frac{p}{2r}+\frac{1}{2}}} \left( \left( \frac{\log(T)}{\Delta} \right)^{\frac{2r}{2r+p}} + \frac{T^{\frac{p}{2r+p}} \log(T)}{\kappa \Delta} \right)^{\frac{1}{2}} \right).$$

So that,

$$\left| \left\langle \kappa_k^{\frac{p}{2r}-1}(F_{h_2,i} - f_i * \mathcal{K}_{h_2}) - \widehat{\kappa}_k^{\frac{p}{2r}-1}(F_{h_1,i} - f_i * \mathcal{K}_{h_1}), (f_{\eta_k} - f_{\eta_{k+1}}) * \mathcal{K}_{h_2} - (f_{\widehat{\eta}_k} - f_{\widehat{\eta}_{k+1}}) * \mathcal{K}_{h_1} \right\rangle_{L_2} \right|$$

$$= O_p\left( \frac{1}{\kappa_k^{2-\frac{p}{2r}}} \left( \left( \frac{\log(T)}{\Delta} \right)^{\frac{2r}{2r+p}} + \frac{T^{\frac{p}{2r+p}} \log(T)}{\kappa \Delta} \right)^{\frac{1}{2}} \right) O_p\left( \frac{1}{\kappa_k^{\frac{p}{2r}+\frac{1}{2}}} \left( \left( \frac{\log(T)}{\Delta} \right)^{\frac{2r}{2r+p}} + \frac{T^{\frac{p}{2r+p}} \log(T)}{\kappa \Delta} \right)^{\frac{1}{2}} \right)$$

$$\cdot O_p\left( \frac{1}{\kappa_k^{\frac{p}{2r}+\frac{1}{2}}} \left( \left( \frac{\log(T)}{\Delta} \right)^{\frac{2r}{2r+p}} + \frac{T^{\frac{p}{2r+p}} \log(T)}{\kappa \Delta} \right)^{\frac{1}{2}} \right).$$

Now, in a similar way, we observe that

$$\left\langle \kappa_k^{\frac{p}{2r}-1}(F_{h_2,i} - f_i * \mathcal{K}_{h_2}), (f_{\eta_k} - f_{\eta_{k+1}}) * \mathcal{K}_{h_2} - (f_{\widehat{\eta}_k} - f_{\widehat{\eta}_{k+1}}) * \mathcal{K}_{h_1} \right\rangle_{L_2}$$

$$\leq ||\kappa_k^{\frac{p}{2r}-1}(F_{h_2,i} - f_i * \mathcal{K}_{h_2})||_{L_2} ||(f_{\eta_k} - f_{\eta_{k+1}}) * \mathcal{K}_{h_2} - (f_{\widehat{\eta}_k} - f_{\widehat{\eta}_{k+1}}) * \mathcal{K}_{h_1}||_{L_2}$$

$$= O_p(\kappa_k^{\frac{p}{2r}-1} \kappa_k^{-\frac{p}{2r}}) O_p\left( \frac{1}{\kappa_k^{\frac{p}{2r}+\frac{1}{2}}} \left( \left( \frac{\log(T)}{\Delta} \right)^{\frac{2r}{2r+p}} + \frac{T^{\frac{p}{2r+p}} \log(T)}{\kappa \Delta} \right)^{\frac{1}{2}} \right)$$

where equality is followed by noticing that

$$||F_{h_2,i} - f_i * \mathcal{K}_{h_2}||_{L_2} \leq ||F_{h_2,i}||_{L_2} + ||f_i * \mathcal{K}_{h_2}||_{L_2} \tag{41}$$

$$= O(\kappa_k^{-\frac{p}{2r}}) + O(1), \tag{42}$$

and then using Remark 2 and Assumption 1. Finally,

$$\left\langle \widehat{\kappa}_k^{\frac{p}{2r}-1}(F_{h_1,i} - f_i * \mathcal{K}_{h_1}) - \kappa_k^{\frac{p}{2r}-1}(F_{h_2,i} - f_i * \mathcal{K}_{h_2}), (f_{\widehat{\eta}_k} - f_{\widehat{\eta}_{k+1}}) * \mathcal{K}_{h_1} \right\rangle_{L_2}$$

$$\leq |\widehat{\kappa}_k^{\frac{p}{2r}-1} - \kappa_k^{\frac{p}{2r}-1}| \|(F_{h_1,i} - f_i * \mathcal{K}_{h_1}) - (F_{h_2,i} - f_i * \mathcal{K}_{h_2})\|_{L_2} \|(f_{\widehat{\eta}_k} - f_{\widehat{\eta}_{k+1}}) * \mathcal{K}_{h_1}\|_{L_2}$$

$$= O_p\left(\frac{1}{\kappa_k^{2-\frac{p}{2r}}}\left(\left(\frac{\log(T)}{\Delta}\right)^{\frac{2r}{2r+p}} + \frac{T^{\frac{p}{2r+p}}\log(T)}{\kappa\Delta}\right)^{\frac{1}{2}}\right) O_p\left(\frac{1}{\kappa_k^{\frac{p}{2r}+\frac{1}{2}}}\left(\left(\frac{\log(T)}{\Delta}\right)^{\frac{2r}{2r+p}} + \frac{T^{\frac{p}{2r+p}}\log(T)}{\kappa\Delta}\right)^{\frac{1}{2}}\right)\kappa_k$$

where equality is followed by Remark 2, Assumption 1 and Minkowski's inequality. Therefore,

$$I_1 \leq \frac{1}{\sqrt{S}}\sum_{i \in \mathcal{S}_r}\left|\breve{Y}_i - Y_i\right|$$

$$\sqrt{S}\left(O_p\left(\frac{1}{\kappa_k^{2-\frac{p}{2r}}}\left(\left(\frac{\log(T)}{\Delta}\right)^{\frac{2r}{2r+p}} + \frac{T^{\frac{p}{2r+p}}\log(T)}{\kappa\Delta}\right)\right)\right.$$

$$\cdot O_p\left(\frac{1}{\kappa_k^{\frac{p}{2r}+\frac{1}{2}}}\left(\left(\frac{\log(T)}{\Delta}\right)^{\frac{2r}{2r+p}} + \frac{T^{\frac{p}{2r+p}}\log(T)}{\kappa\Delta}\right)^{\frac{1}{2}}\right) O_p\left(\frac{1}{\kappa_k^{\frac{p}{2r}+\frac{1}{2}}}\left(\left(\frac{\log(T)}{\Delta}\right)^{\frac{2r}{2r+p}} + \frac{T^{\frac{p}{2r+p}}\log(T)}{\kappa\Delta}\right)^{\frac{1}{2}}\right)$$

$$+ O_p(\kappa_k^{\frac{p}{2r}-1}\kappa_k^{-\frac{p}{2r}}) O_p\left(\frac{T^{-\frac{r}{2r+p}}}{\kappa_k^{\frac{p}{2r}+\frac{1}{2}}}\log^{\frac{r}{2r+p}}(T)\right) + O_p\left(\frac{T^{-\frac{r}{4r+2p}}}{\kappa_k^{2-\frac{p}{2r}}}\log^{\frac{r}{2r+p}}(T)\right) O_p\left(\frac{T^{-\frac{r}{2r+p}}}{\kappa_k^{\frac{p}{2r}+\frac{1}{2}}}\log^{\frac{r}{2r+p}}(T)\kappa_k\right)$$

$$= \sqrt{S}\left(O_p\left(\frac{T^{-\frac{2r}{2r+p}}}{\kappa_k^{2-\frac{p}{2r}}}\log(T)^{\frac{2r}{2r+p}}\right) O_p\left(\frac{T^{-\frac{2r}{2r+p}}}{\kappa_k^{\frac{p}{r}+1}}\log(T)^{\frac{2r}{2r+p}}\right) + O_p(\kappa_k^{-1}) O_p\left(\frac{T^{-\frac{2r}{2r+p}}}{\kappa_k^{\frac{p}{2r}+\frac{1}{2}}}\log(T)^{\frac{r}{2r+p}}\right)\right).$$

To bound the $I_2$ term, we add and subtract $\kappa_k^{\frac{p}{2r}-1}\left\langle F_{h_2,i} - f_i * \mathcal{K}_{h_2}, (f_{\widehat{\eta}_k} - f_{\widehat{\eta}_{k+1}}) * \mathcal{K}_{h_1} \right\rangle_{L_2}$, to get

$$\left|\breve{Y}_i + Y_i\right|$$

$$= \left|\kappa_k^{\frac{p}{2r}-1}\left\langle F_{h_2,i} - f_i * \mathcal{K}_{h_2}, (f_{\eta_k} - f_{\eta_{k+1}}) * \mathcal{K}_{h_2} \right\rangle_{L_2} + \widehat{\kappa}_k^{\frac{p}{2r}-1}\left\langle F_{h_1,i} - f_i * \mathcal{K}_{h_1}, (f_{\widehat{\eta}_k} - f_{\widehat{\eta}_{k+1}}) * \mathcal{K}_{h_1} \right\rangle_{L_2}\right|$$

$$= \left|\kappa_k^{\frac{p}{2r}-1}\left\langle F_{h_2,i} - f_i * \mathcal{K}_{h_2}, (f_{\eta_k} - f_{\eta_{k+1}}) * \mathcal{K}_{h_2} \right\rangle_{L_2} - \kappa_k^{\frac{p}{2r}-1}\left\langle F_{h_2,i} - f_i * \mathcal{K}_{h_2}, (f_{\widehat{\eta}_k} - f_{\widehat{\eta}_{k+1}}) * \mathcal{K}_{h_1} \right\rangle_{L_2}\right.$$

$$+ \kappa_k^{\frac{p}{2r}-1}\left\langle F_{h_2,i} - f_i * \mathcal{K}_{h_2}, (f_{\eta_k} - f_{\eta_{k+1}}) * \mathcal{K}_{h_1} \right\rangle_{L_2} + \widehat{\kappa}_k^{\frac{p}{2r}-1}\left\langle F_{h_1,i} - f_i * \mathcal{K}_{h_1}, (f_{\widehat{\eta}_k} - f_{\widehat{\eta}_{k+1}}) * \mathcal{K}_{h_1} \right\rangle_{L_2}\right|$$

$$= \left|\kappa_k^{\frac{p}{2r}-1}\left\langle F_{h_2,i} - f_i * \mathcal{K}_{h_2}, (f_{\eta_k} - f_{\eta_{k+1}}) * \mathcal{K}_{h_2} \right\rangle_{L_2} + \kappa_k^{\frac{p}{2r}-1}\left\langle F_{h_2,i} - f_i * \mathcal{K}_{h_2}, (f_{\widehat{\eta}_k} - f_{\widehat{\eta}_{k+1}}) * \mathcal{K}_{h_1} \right\rangle_{L_2}\right.$$

$$\left.+ \left\langle \widehat{\kappa}_k^{\frac{p}{2r}-1}(F_{h_1,i} - f_i * \mathcal{K}_{h_1}) - \kappa_k^{\frac{p}{2r}-1}(F_{h_2,i} - f_i * \mathcal{K}_{h_2}), (f_{\widehat{\eta}_k} - f_{\widehat{\eta}_{k+1}}) * \mathcal{K}_{h_1} \right\rangle_{L_2}\right|.$$

Then, as before, we bound each of the terms above using Hölder's inequality. We start with the term

$$\left|\kappa_k^{\frac{p}{2r}-1}\left\langle F_{h_2,i} - f_i * \mathcal{K}_{h_2}, (f_{\eta_k} - f_{\eta_{k+1}}) * \mathcal{K}_{h_2} \right\rangle_{L_2}\right|$$

$$\leq \kappa_k^{\frac{p}{2r}-1}\|F_{h_2,i} - f_i * \mathcal{K}_{h_2}\|_{L_2}\|(f_{\eta_k} - f_{\eta_{k+1}}) * \mathcal{K}_{h_2}\|_{L_2}$$

$$\leq \kappa_k^{\frac{p}{2r}-1}O_p(\kappa_k^{-\frac{p}{2r}})\kappa_k = O_p(1)$$

where the second inequality is followed by (41). Similarly,

$$\left|\kappa_k^{\frac{p}{2r}-1}\left\langle F_{h_2,i} - f_i * \mathcal{K}_{h_2}, (f_{\widehat{\eta}_k} - f_{\widehat{\eta}_{k+1}}) * \mathcal{K}_{h_1} \right\rangle_{L_2}\right|$$

$$\leq \kappa_k^{\frac{p}{2r}-1}\|F_{h_2,i} - f_i * \mathcal{K}_{h_2}\|_{L_2}\|(f_{\widehat{\eta}_k} - f_{\widehat{\eta}_{k+1}}) * \mathcal{K}_{h_1}\|_{L_2}$$

$$\leq \kappa_k^{\frac{p}{2r}-1}O_p(\kappa_k^{-\frac{p}{2r}})\kappa_k = O_p(1)$$

where the second inequality is followed by the (41). Finally, the term

$$\left|\left\langle \widehat{\kappa}_k^{\frac{p}{2r}-1}(F_{h_1,i} - f_i * \mathcal{K}_{h_1}) - \kappa_k^{\frac{p}{2r}-1}(F_{h_2,i} - f_i * \mathcal{K}_{h_2}), (f_{\widehat{\eta}_k} - f_{\widehat{\eta}_{k+1}}) * \mathcal{K}_{h_1} \right\rangle_{L_2}\right|$$

was previously bounded by,

$$O_p\Big(\frac{1}{\kappa_k^{2-\frac{p}{2r}}}\Big(\Big(\frac{\log(T)}{\Delta}\Big)^{\frac{2r}{2r+p}}+\frac{T^{\frac{p}{2r+p}}\log(T)}{\kappa\Delta}\Big)\Big)O_p\Big(\frac{1}{\kappa_k^{\frac{p}{2r}+\frac{1}{2}}}\Big(\Big(\frac{\log(T)}{\Delta}\Big)^{\frac{2r}{2r+p}}+\frac{T^{\frac{p}{2r+p}}\log(T)}{\kappa\Delta}\Big)^{\frac{1}{2}}\Big)\kappa_k.$$

Therefore,

$$
\begin{aligned}
I_2 \le & \frac{1}{\sqrt{S}}\sum_{i\in\mathcal{S}_r}\Big|\breve{Y}_i+Y_i\Big|\\
=&\sqrt{S}\Big(O_p(1)+O_p\Big(\frac{1}{\kappa_k^{2-\frac{p}{2r}}}\Big(\Big(\frac{\log(T)}{\Delta}\Big)^{\frac{2r}{2r+p}}+\frac{T^{\frac{p}{2r+p}}\log(T)}{\kappa\Delta}\Big)\Big)O_p\Big(\frac{1}{\kappa_k^{\frac{p}{2r}+\frac{1}{2}}}\Big(\Big(\frac{\log(T)}{\Delta}\Big)^{\frac{2r}{2r+p}}+\frac{T^{\frac{p}{2r+p}}\log(T)}{\kappa\Delta}\Big)^{\frac{1}{2}}\Big)\kappa_k\Big).
\end{aligned}
$$

In consequences,

$$
\Big|\widehat{\sigma}_\infty^2(k)-\breve{\sigma}_\infty^2(k)\Big|=\Big|\frac{1}{R}\sum_{r=1}^{R}\Big(\frac{1}{\sqrt{S}}\sum_{i\in\mathcal{S}_r}\breve{Y}_i\Big)^2-\frac{1}{R}\sum_{r=1}^{R}\Big(\frac{1}{\sqrt{S}}\sum_{i\in\mathcal{S}_r}Y_i\Big)^2\Big|
$$

$$
=\Big|\frac{1}{R}\sum_{r=1}^{R}I_1I_2\Big|
$$

$$
=S\Big(O_p\Big(\frac{1}{\kappa_k^{4-\frac{p}{r}}}\Big(\Big(\frac{\log(T)}{\Delta}\Big)^{\frac{2r}{2r+p}}+\frac{T^{\frac{p}{2r+p}}\log(T)}{\kappa\Delta}\Big)^2\Big)
$$

$$
\cdot O_p\Big(\frac{1}{\kappa_k^{\frac{p}{r}+1}}\Big(\Big(\frac{\log(T)}{\Delta}\Big)^{\frac{2r}{2r+p}}+\frac{T^{\frac{p}{2r+p}}\log(T)}{\kappa\Delta}\Big)\Big)O_p\Big(\frac{1}{\kappa_k^{\frac{p}{2r}+\frac{1}{2}}}\Big(\Big(\frac{\log(T)}{\Delta}\Big)^{\frac{2r}{2r+p}}+\frac{T^{\frac{p}{2r+p}}\log(T)}{\kappa\Delta}\Big)^{\frac{1}{2}}\Big)\kappa_k
$$

$$
+O_p\Big(\frac{1}{\kappa_k^{2-\frac{p}{2r}}}\Big(\Big(\frac{\log(T)}{\Delta}\Big)^{\frac{2r}{2r+p}}+\frac{T^{\frac{p}{2r+p}}\log(T)}{\kappa\Delta}\Big)\Big)
$$

$$
\cdot O_p\Big(\frac{1}{\kappa_k^{\frac{p}{2r}+\frac{1}{2}}}\Big(\Big(\frac{\log(T)}{\Delta}\Big)^{\frac{2r}{2r+p}}+\frac{T^{\frac{p}{2r+p}}\log(T)}{\kappa\Delta}\Big)^{\frac{1}{2}}\Big)O_p\Big(\frac{1}{\kappa_k^{\frac{p}{2r}+\frac{1}{2}}}\Big(\Big(\frac{\log(T)}{\Delta}\Big)^{\frac{2r}{2r+p}}+\frac{T^{\frac{p}{2r+p}}\log(T)}{\kappa\Delta}\Big)^{\frac{1}{2}}\Big)
$$

$$
+O_p(\kappa_k^{-1})O_p\Big(\frac{1}{\kappa_k^{\frac{p}{2r}+\frac{1}{2}}}\Big(\Big(\frac{\log(T)}{\Delta}\Big)^{\frac{2r}{2r+p}}+\frac{T^{\frac{p}{2r+p}}\log(T)}{\kappa\Delta}\Big)^{\frac{1}{2}}\Big)
$$

$$
+O_p(\kappa_k^{-1})O_p\Big(\frac{1}{\kappa_k^{\frac{p}{2r}+\frac{1}{2}}}\Big(\Big(\frac{\log(T)}{\Delta}\Big)^{\frac{2r}{2r+p}}+\frac{T^{\frac{p}{2r+p}}\log(T)}{\kappa\Delta}\Big)^{\frac{1}{2}}\Big)
$$

$$
\cdot O_p\Big(\frac{1}{\kappa_k^{2-\frac{p}{2r}}}\Big(\Big(\frac{\log(T)}{\Delta}\Big)^{\frac{2r}{2r+p}}+\frac{T^{\frac{p}{2r+p}}\log(T)}{\kappa\Delta}\Big)\Big)O_p\Big(\frac{1}{\kappa_k^{\frac{p}{2r}+\frac{1}{2}}}\Big(\Big(\frac{\log(T)}{\Delta}\Big)^{\frac{2r}{2r+p}}+\frac{T^{\frac{p}{2r+p}}\log(T)}{\kappa\Delta}\Big)^{\frac{1}{2}}\Big)\kappa_k\Big).
$$

In order to conclude (i), we notice that by Assumption 4 and that $S=O(T^{\frac{-p}{2r+p}}\kappa_k^{\frac{p}{2r}+\frac{3}{2}})$, which implies,

$$\Big|\widehat{\sigma}_\infty^2(k)-\breve{\sigma}_\infty^2(k)\Big|=o_p(1).$$

Now, we are going to see that $\Big|\breve{\sigma}_\infty^2(k)-\widetilde{\sigma}_\infty^2(k)\Big|\xrightarrow{P}0,\quad T\to\infty.$ To this end, we will show that the estimator is asymptotically unbiased, and its variance $\to 0$ as $T\to\infty$. First, we notice that, by Hölder's inequality and Minkowsky's inequality,

$$
\begin{aligned}
|\breve{Y}_i|=&\kappa_k^{\frac{p}{2r}-1}\Big\langle F_{h_2,i}-f_i*\mathcal{K}_{h_2},(f_{\eta_k}-f_{\eta_{k+1}})*\mathcal{K}_{h_2}\Big\rangle_{L_2}\Big|\\
\le&\kappa_k^{\frac{p}{2r}-1}||F_{h_2,i}-f_i*\mathcal{K}_{h_2}||_{L_2}||(f_{\eta_k}-f_{\eta_{k+1}})*\mathcal{K}_{h_2}||_{L_2}\\
\le&\kappa_k^{\frac{p}{2r}-1}\kappa_k^{-\frac{p}{2r}}\kappa_k=1.
\end{aligned}
$$

Now, we analyze the Bias. We observe that,

$$\mathbb{E}(\check{\sigma}_\infty^2(k)) = \frac{1}{R}\sum_{r=1}^{R}\mathbb{E}\left(\left(\frac{1}{\sqrt{S}}\sum_{i\in\mathcal{S}_r}\check{Y}_i\right)^2\right) = \frac{1}{S}\mathbb{E}\left(\left(\sum_{i\in\mathcal{S}_r}\check{Y}_i\right)^2\right) = \sum_{l=-S+1}^{S+1}\frac{S-l}{S}\mathbb{E}(\check{Y}_i\check{Y}_{i+l})$$

and,

$$\tilde{\sigma}_\infty^2(k) = \sum_{l=-\infty}^{\infty}\mathbb{E}(\check{Y}_i\check{Y}_{i+l}).$$

so that, the bias has the following form,

$$\tilde{\sigma}_\infty^2(k) - \mathbb{E}(\check{\sigma}_\infty^2(k)) = 2\sum_{l=S}^{\infty}\mathbb{E}(\check{Y}_i\check{Y}_{i+l}) + 2\sum_{l=1}^{S}\frac{l}{S}\mathbb{E}(\check{Y}_i\check{Y}_{i+l}).$$

Now, we show that each of the above terms vanishes as $T\to\infty$. We have that, by condition (2) and covariance inequality

$$2\sum_{l=S}^{\infty}\mathbb{E}(\check{Y}_i\check{Y}_{i+l}) \leq 8\sum_{l=S}^{\infty}||\check{Y}_i||_{L_\infty}^2\alpha_l \leq 8\sum_{l=S}^{\infty}\alpha_l \to 0, \text{ as } T\to\infty$$

where $\alpha_l$ is the mixing coefficient. Then,

$$2\sum_{l=1}^{S}\frac{l}{S}\mathbb{E}(\check{Y}_i\check{Y}_{i+l}) \leq 8\sum_{l=1}^{S}\frac{l}{S}||\check{Y}_i||_{L_\infty}^2\alpha_l \leq \frac{C}{S} \to 0,$$

by condition (2), choice of $S$ and Assumption 4. Therefore, we conclude that the Bias vanishes as $T\to\infty$. To analyze the Variance, we observe that, if $Y_r = \frac{1}{S}\left(\sum_{i\in\mathcal{S}_r}\check{Y}_i\right)^2$

$$Var(\check{\sigma}_\infty^2(k)) = \mathbb{E}((\check{\sigma}_\infty^2(k) - \mathbb{E}(\check{\sigma}_\infty^2(k)))^2)$$

$$= \frac{1}{R^2}\mathbb{E}\left(\left(\sum_{r=1}^{R}Y_r - \mathbb{E}(Y_r)\right)^2\right)$$

$$= \frac{1}{R}\sum_{l=-R+1}^{R-1}\frac{R-l}{R}cov(Y_r, Y_{l+r})$$

$$\leq \frac{8}{R}||Y_r||_{L_\infty}^2\sum_{l=0}^{\infty}\tilde{\alpha}_l \leq \frac{8CS}{R} \to 0, \text{ as, } T\to\infty.$$

where, $\tilde{\alpha}_l$ are the mixing coefficients of $\{Y_r\}_{r\in\mathbb{Z}}$, which is bounded by the mixing coefficient $\alpha_l$. From here, we conclude the result (ii). $\qquad\square$

# G   Large probability events

In this section, we deal with all the large probability events that occurred in the proof of Theorem 1. Recall that, for any $(s, e] \subseteq (0, T]$,

$$\widetilde{f}_t^{s,e}(x) = \sqrt{\frac{e-t}{(e-s)(t-s)}} \sum_{l=s+1}^{t} f_l(x) - \sqrt{\frac{t-s}{(e-s)(e-t)}} \sum_{l=t+1}^{e} f_l(x), \ x \in \mathcal{X}.$$

**Proposition 1.** *For any $x$,*

$$\mathbb{P}\Big( \max_{\rho \leqslant k \leqslant T-\widetilde{r}} \Big| \frac{1}{\sqrt{k}} \sum_{t=\widetilde{r}+1}^{\widetilde{r}+k} \Big( \mathcal{K}_h(x-X_t) - \int \mathcal{K}_h(x-z)dF_t(z) \Big) \Big| \geq C\sqrt{\frac{\log T}{h^p}} \Big) \leqslant T^{-p-3}.$$

*Proof.* We have that the random variables $\{Z_t = \mathcal{K}_h\big(x-X_t\big)\}_{t=1}^{T}$ satisfies

$$\sigma\Big( \mathcal{K}_h\big(x-X_t\big) \Big) \subset \sigma\Big( X_t \Big)$$

and,

$$\Big| \mathcal{K}_h\big(x-X_t\big) \Big| \leqslant \frac{1}{h^p}C_K.$$

Moreover, let

$$V^2 = \sup_{t>0} \Big( var(\mathcal{K}_h(x-X_t) + 2\sum_{j>t} |cov(Z_t, Z_j)|) \Big).$$

We observe that,

$$var(\mathcal{K}_h(x-X_t)) \leq E((\frac{1}{h^p}\mathcal{K}(\frac{x-X_t}{h}))^2)$$

$$\leq \int \frac{1}{h^{2p}}\mathcal{K}(\frac{x-z}{h})dF_t(z)$$

making $\mu = \frac{x-z}{h}$, the last inequality is equal to

$$\int \frac{1}{h^{2p}}\mathcal{K}(\frac{x-z}{h})dF_t(z) \leq \frac{1}{h^p} \int \mathcal{K}^2(u)dF_t(z)$$

$$\leq \frac{1}{h^p}C_kC_f.$$

Then, by proposition 2.5 on Fan & Yao (2008), $|cov(Z_1, Z_1 + t)| \leq C\alpha(t)\frac{1}{h^{2p}}C_K^2$. On the other hand,

$$cov(Z_1, Z_1 + t) = |E(Z_1 Z_{t+1}) - E(Z_1)^2|$$

$$\leq \int \int \mathcal{K}_h(x-z_1)\mathcal{K}_h(x-z_2)g_t(z_1, z_2)dz_1 dz_2 + E(Z_1)^2$$

$$\leq ||g_t||_{L_\infty} + E(Z_1)^2.$$

Since by assumption Assumption 1**b** we have that $||g_t||_{L_\infty} < \infty$, and

$$E(Z_1) = E(\mathcal{K}_h(x-X_1)) = \int \frac{1}{h^p}\mathcal{K}(\frac{x-z}{h})dF_t(z) = \int \mathcal{K}(u)f_t(x-hu)du = O(1),$$

we obtain that $|cov(Z_1, Z_{1+t})|$. Therefore, $\sum_{t=1}^{\frac{1}{h}-1} |cov(Z_1, Z_{t+1})| \leq C\frac{1}{h}$ and, using the mixing condition bound, inequality (2),

$$\sum_{t=\frac{1}{h^p}}^{T-1} |cov(Z_1, Z_1 + t)| \leq D \sum_{t=\frac{1}{h}}^{\infty} \frac{e^{-2Ct}}{h^{2p}}$$

$$\leq D\frac{e^{-2C\frac{1}{h^p}}}{h^{2p}}$$

$$\leq \widetilde{D}\frac{1}{h^{2p}}h^p = \widetilde{D}h^p$$

where the last inequity is followed by the fact $e^{-x} < \frac{1}{x}$ for $x > -1$. In consequence,

$$V^2 = \sup_{t>0}\left(var(\mathcal{K}_h(x - X_t) + 2\sum_{j>t}|cov(Z_t, Z_j)|)\right)$$

$$= \widetilde{C}\frac{1}{h^p} + \widetilde{D}\frac{1}{h^p} = \widetilde{\widetilde{C}}\frac{1}{h^p}.$$

Then, by Bernstein inequality for mixing dependence, see Merlevède et al. (2009) for more details, letting

$$\lambda = C_p\left(\sqrt{\frac{k\log(T)}{h^p}} + \sqrt{\frac{\log(T)}{h^{2p}}} + \sqrt{\frac{\log(T)\log^2(k)}{h^p}}\right)$$

we get that,

$$\mathbb{P}\left(\left|\sum_{t=\widetilde{r}+1}^{\widetilde{r}+k}\left(\mathcal{K}_h(x - X_t) - \int \mathcal{K}_h(x - z)dF_t(z)\right)\right| > \lambda\right) \leq T^{-p-3}.$$

in consequence,

$$\mathbb{P}\left(\left|\frac{1}{\sqrt{k}}\sum_{t=\widetilde{r}+1}^{\widetilde{r}+k}\left(\mathcal{K}_h(x - X_t) - \int \mathcal{K}_h(x - z)dF_t(z)\right)\right| > \frac{\lambda}{\sqrt{k}}\right) \leq T^{-p-3}.$$

Since $kh^p \geq \log(T)$ if $k > \rho$, and $\log^2(k) = O(k)$,

$$\frac{\lambda}{\sqrt{k}} = \frac{C_p\left(\sqrt{\frac{k\log(T)}{h^p}} + \sqrt{\frac{\log(T)}{h^{2p}}} + \sqrt{\frac{\log(T)\log^2(k)}{h^p}}\right)}{\sqrt{k}}$$

$$= C_p\left(\sqrt{\frac{\log(T)}{h^p}} + \sqrt{\frac{\log(T)}{kh^{2p}}} + \sqrt{\frac{\log(T)\log^2(k)}{kh^p}}\right)$$

$$\leq C_p\left(\sqrt{\frac{\log(T)}{h^p}} + \sqrt{\frac{1}{h^p}} + \sqrt{\frac{\log(T)}{h^p}}\right)$$

$$\leq C_1\sqrt{\frac{\log(T)}{h^p}}.$$

It follows that,

$$\mathbb{P}\left(\left|\frac{1}{\sqrt{k}}\sum_{t=\widetilde{r}+1}^{\widetilde{r}+k}\left(\mathcal{K}_h(x - X_t) - \int \mathcal{K}_h(x - z)dF_t(z)\right)\right| > C_1\sqrt{\frac{\log(T)}{h^p}}\right) \leq T^{-p-3}.$$

$\square$

**Proposition 2.** *Define the events*

$$\mathcal{A}_1 = \left\{\max_{t=s+\rho+1}^{e-\rho}\sup_{x\in\mathbb{R}^p}\left|\widetilde{F}_{t,h}^{s,e}(x) - \widetilde{f}_t^{s,e}(x)\right| \geq 2C\sqrt{\frac{\log T}{h^p}} + \frac{2C_1\sqrt{p}}{h^p} + 2C_2\sqrt{T}h^r\right\}$$

*and,*

$$\mathcal{A}_2 = \left\{\max_{\rho\leqslant k\leq T-\widetilde{r}}\sup_{x\in\mathbb{R}^p}\left|\frac{1}{\sqrt{k}}\sum_{t=\widetilde{r}+1}^{\widetilde{r}+k}\left(\mathcal{K}_h(x - X_t) - f_t(x)\right)\right| \geq C\sqrt{\frac{\log T}{h^p}} + \frac{C_1\sqrt{p}}{h^p} + C_2\sqrt{T}h^r\right\}.$$

*Then*

$$\mathbb{P}\left(\mathcal{A}_1\right) \leqslant 2R^pT^{-2} \tag{43}$$

$$\mathbb{P}\left(\mathcal{A}_2\right) \leqslant R^pT^{-2} \tag{44}$$

*where $R$ is a positive constant.*

*Proof.* First, we notice that

$$\max_{\rho \leqslant k \leqslant T-\widetilde{r}} \sup_{x \in \mathbb{R}^p} \left| \frac{1}{\sqrt{k}} \sum_{t=\widetilde{r}+1}^{\widetilde{r}+k} \mathcal{K}_h\left(x - X_t\right) - f_t(x) \right|$$

$$\leq \max_{\rho \leq k \leq T-\widetilde{r}} \sup_{x \in \mathbb{R}^p} \left| \frac{1}{\sqrt{k}} \sum_{t=\widetilde{r}+1}^{\widetilde{r}+k} \left( \mathcal{K}_h(x - X_t) - \int \mathcal{K}_h(x - z) dF_t(z) \right) \right|$$

$$+ \max_{\rho \leqslant k \leqslant T-\widetilde{r}} \sup_{x \in \mathbb{R}^p} \left| \frac{1}{\sqrt{k}} \sum_{t=\widehat{r}+1}^{r+k} \left( \int \mathcal{K}_h(x - z) dF_z(z) - f_t(x) \right) \right| = I_1 + I_2$$

Now we will bound each of the terms $I_1$, $I_2$. For $I_1$, we consider

$$A = \{x_1, ..., x_{(R\sqrt{T}/\sqrt{h^p}h)^p}\}$$

with $\cup_{x_i \in \{x_1,...,x_{(R\sqrt{T}/\sqrt{h^p}h)^p}\}} Rec(x_i, \frac{\sqrt{h^p}h}{\sqrt{T}}) \supset D$, where $D$ is the support of $K$ and $Rec(x_i, \frac{\sqrt{h^p}h}{\sqrt{T}})$ are boxes centered at $x_i$ of size $\frac{\sqrt{h^p}h}{\sqrt{T}}$ and $R$ is the size of the boxe containing $D$. Then by Proposition 1, for any $x_i$,

$$\mathbb{P}\left( \max_{\rho \leqslant k \leq T-\widetilde{r}} \left| \frac{1}{\sqrt{k}} \sum_{t=\widetilde{r}+1}^{\widetilde{r}+k} \left( \mathcal{K}_h(x_i - X_t) - \int \mathcal{K}_h(x_i - z) dF_t(z) \right) \right| \geq C\sqrt{\frac{\log T}{h^p}} \right) \leqslant T^{-p-3},$$

$$(45)$$

by an union bound argument,

$$\mathbb{P}\left( \max_{\rho \leq k \leq T-\widetilde{r}} \sup_{x \in A} \left| \frac{1}{\sqrt{k}} \sum_{t=\widetilde{r}+1}^{\widetilde{r}+k} \left( \mathcal{K}_h(x - X_t) - \int \mathcal{K}_h(x - z) dF_t(z) \right) \right| \geq C\sqrt{\frac{\log T}{h^p}} \right) \leqslant T^{-p-3}|A|.$$

$$(46)$$

Let $I_{1,1} = \{\max_{\rho \leqslant k \leq T-\widetilde{r}} \sup_{x \in A} \left| \frac{1}{\sqrt{k}} \sum_{t=\widetilde{r}+1}^{\widetilde{r}+k} \left( \mathcal{K}_h(x - X_t) - \int \mathcal{K}_h(x - z) dF_t(z) \right) \right|\}$. For any $x \in \mathbb{R}^p$, there exist $x_i \in A$ such that

$$\max_{\rho \leqslant k \leq T-\widetilde{r}} \left| \frac{1}{\sqrt{k}} \sum_{t=\widetilde{r}+1}^{\widetilde{r}+k} \left( \mathcal{K}_h(x - X_t) - \int \mathcal{K}_h(x - z) dF_t(z) \right) \right|$$

$$\leq \max_{\rho \leqslant k \leq T-\widetilde{r}} \left| \frac{1}{\sqrt{k}} \sum_{t=\widetilde{r}+1}^{\widetilde{r}+k} \left( \mathcal{K}_h(x_i - X_t) - \int \mathcal{K}_h(x_i - z) dF_t(z) \right) \right|$$

$$+ \max_{\rho \leqslant k \leq T-\widetilde{r}} \left| \frac{1}{\sqrt{k}} \sum_{t=\widetilde{r}+1}^{\widetilde{r}+k} \left( \mathcal{K}_h(x - X_t) - \mathcal{K}_h(x_i - X_t) \right) \right|$$

$$+ \max_{\rho \leqslant k \leq T-\widetilde{r}} \left| \frac{1}{\sqrt{k}} \sum_{t=\widetilde{r}+1}^{\widetilde{r}+k} \left( \int \mathcal{K}_h(x_i - z) dF_t(z) - \int \mathcal{K}_h(x - z) dF_t(z) \right) \right|$$

$$\leq \max_{\rho \leqslant k \leq T-\widetilde{r}} \sup_{x \in A} \left| \frac{1}{\sqrt{k}} \sum_{t=\widetilde{r}+1}^{\widetilde{r}+k} \left( \mathcal{K}_h(x_i - X_t) - \int \mathcal{K}_h(x_i - z) dF_t(z) \right) \right|$$

$$+ \max_{\rho \leqslant k \leq T-\widetilde{r}} \left| \frac{1}{\sqrt{k}} \sum_{t=\widetilde{r}+1}^{\widetilde{r}+k} \left( \mathcal{K}_h(x - X_t) - \mathcal{K}_h(x_i - X_t) \right) \right|$$

$$+ \max_{\rho \leqslant k \leq T-\widetilde{r}} \left| \frac{1}{\sqrt{k}} \sum_{t=\widetilde{r}+1}^{\widetilde{r}+k} \left( \int \mathcal{K}_h(x_i - z) dF_t(z) - \int \mathcal{K}_h(x - z) dF_t(z) \right) \right|$$

$$= I_{1,1} + I_{1,2} + I_{1,3}$$

The term $I_{1,2}$ is bounded as followed.

$$\max_{\rho \leqslant k \leq T-\widetilde{r}} \left| \frac{1}{\sqrt{k}} \sum_{t=\widetilde{r}+1}^{\widetilde{r}+k} \left( \mathcal{K}_h(x - X_t) - \mathcal{K}_h(x_i - X_t) \right) \right|$$

$$\leq \max_{\rho \leqslant k \leq T-\widetilde{r}} \frac{1}{\sqrt{k}} \sum_{t=\widetilde{r}+1}^{\widetilde{r}+k} \left| \mathcal{K}_h(x - X_t) - \mathcal{K}_h(x_i - X_t) \right|$$

$$\leq \max_{\rho \leqslant k \leq T-\widetilde{r}} \frac{1}{\sqrt{k}} \sum_{t=\widetilde{r}+1}^{\widetilde{r}+k} \frac{|x - x_i|}{h^{p+1}}$$

$$\leq \max_{\rho \leqslant k \leq T-\widetilde{r}} \frac{1}{\sqrt{k}} \sum_{t=\widetilde{r}+1}^{\widetilde{r}+k} \frac{\sqrt{h^p} h \sqrt{p}}{\sqrt{T} h^{p+1}} \leq \frac{\sqrt{p}}{\sqrt{h^p}}$$

For the term $I_{1,3}$, since the random variables $\{\mathcal{K}_h(x - X_t)\}_{t=1}^{T}$ have bounded expected value for any $x \in \mathbb{R}^p$

$$\max_{\rho \leqslant k \leq T-\widetilde{r}} \left| \frac{1}{\sqrt{k}} \sum_{t=\widetilde{r}+1}^{\widetilde{r}+k} \left( \int \mathcal{K}_h(x_i - z) dF_t(z) - \int \mathcal{K}_h(x - z) dF_t(z) \right) \right|$$

$$\leq \max_{\rho \leqslant k \leq T-\widetilde{r}} \left| \frac{1}{\sqrt{k}} \sum_{t=\widetilde{r}+1}^{\widetilde{r}+k} \left( \frac{\sqrt{h^p} h 2C \sqrt{p}}{h^{p+1} \sqrt{T}} \right) \right| \leq \frac{2C \sqrt{p}}{\sqrt{h^p}}.$$

Thus,

$$\max_{\rho \leqslant k \leq T-\widetilde{r}} \sup_{x \in \mathbb{R}^p} \left| \frac{1}{\sqrt{k}} \sum_{t=\widetilde{r}+1}^{\widetilde{r}+k} \left( \mathcal{K}_h(x - X_t) - \int \mathcal{K}_h(x - z) dF_t(z) \right) \right|$$

$$\leq I_{1,1} + C_2 \frac{\sqrt{p}}{\sqrt{h^p}}$$

From here,

$$\mathbb{P}\left( I_1 > C_1 \sqrt{\frac{\log T}{h^p}} + \frac{C_2 \sqrt{p}}{\sqrt{h^p}} \right) \leq \mathbb{P}\left( I_{1,1} + I_{1,2} + I_{1,3} > C_1 \sqrt{\frac{\log T}{h^p}} + \frac{C_2 \sqrt{p}}{\sqrt{h^p}} \right) \tag{47}$$

$$\leq \mathbb{P}\left( I_{1,1} \right) \leq T^{-p-3} |A| = T^{-p-2} (R\sqrt{T}/\sqrt{h^p} h)^p \tag{48}$$

$$\leq T^{-p-3} (R\sqrt{T} \sqrt{T} T^{\frac{1}{p}})^p = R^p T^{-2} \tag{49}$$

Finally, we analyze the term $I_2$. By the adaptive assumption, the following is satisfied,

$$\max_{\rho \leqslant k \leqslant T-\widetilde{r}} \sup_{x \in \mathbb{R}^p} \left| \frac{1}{\sqrt{k}} \sum_{t=\widehat{r}+1}^{r+k} \left( \int \mathcal{K}_h(x - z) dF_z(z) - f_t(x) \right) \right|$$

$$\leq \max_{\rho \leqslant k \leqslant T-\widetilde{r}} \frac{1}{\sqrt{k}} \sum_{t=\widehat{r}+1}^{r+k} \sup_{x \in \mathbb{R}^p} \left| \int \mathcal{K}_h(x - z) dF_z(z) - f_t(x) \right|$$

$$\leq \max_{\rho \leqslant k \leqslant T-\widetilde{r}} \frac{1}{\sqrt{k}} \sum_{t=\widehat{r}+1}^{r+k} C_2 h^r$$

$$\leq C_2 \sqrt{T} h^r$$

We conclude the bound for event $\mathcal{A}_2$. We conclude the bound for event $\mathcal{A}_2$. Next, to derive the bound for event $\mathcal{A}_1$, by definition of $\widetilde{F}_{t,h}^{s,e}$ and $\widetilde{f}_t^{s,e}$, we have that

$$\left| \widetilde{F}_{t,h}^{s,e}(x) - \widetilde{f}_t^{s,e}(x) \right| \leq \left| \sqrt{\frac{e-t}{(e-s)(t-s)}} \sum_{l=s+1}^{t} (F_{l,h}(x) - f_{l,h}(x)) \right|$$

$$+ \left| \sqrt{\frac{t-s}{(e-s)(e-t)}} \sum_{l=t+1}^{e} (F_{l,h}(x) - f_{l,h}(x)) \right|.$$

Then, we observe that,

$$\sqrt{\frac{e-t}{(e-s)(t-s)}} \leq \sqrt{\frac{1}{t-s}} \text{ if } s \leq t, \text{ and } \sqrt{\frac{t-s}{(e-s)(e-t)}} \leq \sqrt{\frac{1}{e-t}} \text{ if } t \leq e.$$

Therefore,

$$X = \max_{t=s+\rho+1}^{e-\rho} \left| \widetilde{F}_{t,h}^{(s,e)}(x) - \widetilde{f}_t^{s,e}(x) \right| \leq \max_{t=s+\rho+1}^{e-\rho} \left| \sqrt{\frac{1}{t-s}} \sum_{l=s+1}^{t} \left( F_{l,h}(x) - \{f_{l,h}(x)\} \right) \right|$$

$$+ \max_{t=s+\rho+1}^{e-\rho} \left| \sqrt{\frac{1}{e-t}} \sum_{l=t+1}^{e} \left( F_{l,h}(x) - f_{l,h}(x) \right) \right| = X_1 + X_2.$$

Finally, letting $\lambda = 2C_1 \sqrt{\frac{\log T}{h^p}} + \frac{2C_2\sqrt{p}}{\sqrt{h^p}} + 2C_2\sqrt{T}h^r$, we get that

$$\mathbb{P}(X \geq \lambda) \leq \mathbb{P}(X_1 + X_2 \geq \frac{\lambda}{2} + \frac{\lambda}{2})$$

$$\leq \mathbb{P}(X_1 \geq \frac{\lambda}{2}) + \mathbb{P}(X_2 \geq \frac{\lambda}{2})$$

$$\leq 2R^p T^{-2},$$

where the last inequality follows from above. This concludes the bound for $\mathcal{A}_1$. $\square$

**Remark 1.** *On the events $(\mathcal{A}_1)^c$ and, $(\mathcal{A}_2)^c$, by Assumption 1, we have that*

$$\max_{t=s+\rho+1}^{e-\rho} \|\widetilde{F}_{t,h}^{s,e}(x) - \widetilde{f}_t^{s,e}(x)\|_{L_2} \leq \max_{t=s+\rho+1}^{e-\rho} \widetilde{C}_{\mathcal{X}} \sup_{x \in \mathbb{R}^p} \left| \widetilde{F}_{t,h}^{s,e}(x) - \widetilde{f}_t^{s,e}(x) \right|$$

$$\leq 2\widetilde{C}_{\mathcal{X}} C \sqrt{\frac{\log T}{h^p}} + \frac{2\widetilde{C}_{\mathcal{X}} C_1 \sqrt{p}}{h^p} + 2\widetilde{C}_{\mathcal{X}} C_2 \sqrt{T} h^r$$

*where $\widetilde{C}_{\mathcal{X}}$ is the volume of the set $\mathcal{X}$. Moreover, using inequality (47), we have that*

$$\max_{\rho \leqslant k \leq T-\widetilde{r}} \left\| \frac{1}{\sqrt{k}} \sum_{t=\widetilde{r}+1}^{\widetilde{r}+k} \left( \mathcal{K}_h(\cdot - X_t) - \int \mathcal{K}_h(\cdot - z)dF_t(z) \right) \right\|_{L_2}$$

$$\leq C_{\mathcal{X}} \max_{\rho \leqslant k \leq T-\widetilde{r}} \sup_{x \in \mathbb{R}^p} \left| \frac{1}{\sqrt{k}} \sum_{t=\widetilde{r}+1}^{\widetilde{r}+k} \left( \mathcal{K}_h(x - X_t) - \int \mathcal{K}_h(x - z)dF_t(z) \right) \right|$$

$$= O_p\left( \sqrt{\frac{\log T}{h^p}} \right).$$

# H   $\alpha$-mixing condition

A process $(X_t, t \in \mathbb{Z})$ is said to be $\alpha$-mixing if

$$\alpha_k = \sup_{t \in \mathbb{Z}} \alpha(\sigma(X_s, s \leq t), \sigma(X_s, s \geq t+k)) \longrightarrow_{k \to \infty} 0.$$

The strong mixing, or $\alpha$-mixing coefficient between two $\sigma$-fields $\mathcal{A}$ and $\mathcal{B}$ is defined as

$$\alpha(\mathcal{A}, \mathcal{B}) = \sup_{A \in \mathcal{A}, B \in \mathcal{B}} |\mathbb{P}(A \cap B) - \mathbb{P}(A)\mathbb{P}(B)|.$$

Suppose $X$ and $Y$ are two random variables. Then for positive numbers $p^{-1} + q^{-1} + r^{-1} = 1$, it holds that

$$|\operatorname{Cov}(X,Y)| \leq 4\|X\|_{L_p}\|Y\|_{L_q}\{\alpha(\sigma(X), \sigma(Y))\}^{1/r}.$$

Let $\left\{Z_t\right\}_{t=-\infty}^{\infty}$ be a stationary time series vectors. Denote the alpha mixing coefficients of $k$ to be

$$\alpha(k) = \alpha\left(\sigma\left\{\ldots, Z_{t-1}, Z_t\right\}, \sigma\left\{Z_{t+k}, Z_{t+k+1}, \ldots\right\}\right).$$

Note that the definition is independent of $t$.

## H.1   Maximal Inequality

The unstationary version of the following lemma is in Lemma B.5. of Kirch (2006).

**Lemma 1.** *Suppose $\left\{y_i\right\}_{i=1}^{\infty}$ is a stationary alpha-mixing time series with mixing coefficient $\alpha(k)$ and that $\mathbb{E}\left(y_i\right) = 0$. Suppose that there exists $\delta, \Delta > 0$ such that*

$$\mathbb{E}\left(\left|y_i\right|^{2+\delta+\Delta}\right) \leq D_1$$

*and*

$$\sum_{k=0}^{\infty}(k+1)^{\delta/2}\alpha(k)^{\Delta/(2+\delta+\Delta)} \leq D_2.$$

*Then*

$$\mathbb{E}\left(\max_{k=1,\ldots,n}\left|\sum_{i=1}^{k}y_i\right|^{2+\delta}\right) \leq Dn^{(2+\delta)/2},$$

*where $D$ only depends on $\delta$ and the joint distribution of $\left\{y_i\right\}_{i=1}^{\infty}$.*

*Proof.* This is Lemma B.8. of Kirch (2006). $\qquad\square$

**Lemma 2.** *Suppose that there exists $\delta, \Delta > 0$ such that*

$$\mathbb{E}\left(\left|y_i\right|^{2+\delta+\Delta}\right) \leq D_1$$

*and*

$$\sum_{k=0}^{\infty}(k+1)^{\delta/2}\alpha(k)^{\Delta/(2+\delta+\Delta)} \leq D_2.$$

*Then it holds that for any $d > 0, 0 < \nu < 1$ and $x > 0$,*

$$\mathbb{P}\left(\max_{k \in [\nu d, d]} \frac{\left|\sum_{i=1}^{k}y_i\right|}{\sqrt{k}} \geq x\right) \leq Cx^{-2-\delta},$$

*where $C$ is some constant.*

*Proof.* Let

$$S_d^* = \max_{k=1,\dots,d} \Big| \sum_{i=1}^{k} y_i \Big|.$$

Then Lemma 1 implies that

$$\left\| S_d^* \right\|_{L_{2+\delta}} \leq C_1 d^{1/2}$$

Therefore it holds that

$$\mathbb{P}\Big( \Big| \frac{S_d^*}{\sqrt{d}} \Big| \geq x \Big) = \mathbb{P}\Big( \Big| \frac{S_d^*}{\sqrt{d}} \Big|^{2+\delta} \geq x^{2+\delta} \Big) \leq C_1 x^{-2-\delta}.$$

Observe that

$$\frac{|S_d^*|}{\sqrt{d}} = \max_{k=1,\dots,d} \frac{\Big| \sum_{i=1}^{k} y_i \Big|}{\sqrt{d}} \geq \max_{k \in [\nu d, d]} \frac{\Big| \sum_{i=1}^{k} y_i \Big|}{\sqrt{d}} \geq \max_{k \in [\nu d, d]} \frac{\Big| \sum_{i=1}^{k} y_i \Big|}{\sqrt{k/\nu}}$$

Therefore

$$\mathbb{P}\Big( \max_{k \in [\nu d, d]} \frac{\Big| \sum_{i=1}^{k} y_i \Big|}{\sqrt{k}} \geq x/\sqrt{\nu} \Big) \leq \mathbb{P}\Big( \Big| \frac{S_d^*}{\sqrt{d}} \Big| \geq x \Big) \leq C_1 x^{-2-\delta},$$

which gives

$$\mathbb{P}\Big( \max_{k \in [\nu d, d]} \frac{\Big| \sum_{i=1}^{k} y_i \Big|}{\sqrt{k}} \geq x \Big) \leq C_2 x^{-2-\delta}.$$

$\square$

**Lemma 3.** *Let $\nu > 0$ be given. Under the same assumptions as in Lemma 1, for any $0 < a < 1$ it holds that*

$$\mathbb{P}\Big( \Big| \sum_{i=1}^{r} y_i \Big| \leq \frac{C}{a}\sqrt{r}\{\log(r\nu) + 1\} \text{ for all } r \geq 1/\nu \Big) \geq 1 - a^2,$$

*where $C$ is some absolute constant.*

*Proof.* Let $s \in \mathbb{Z}^+$ and $\mathcal{T}_s = \left[ 2^s/\nu, 2^{s+1}/\nu \right]$. By Lemma 3, for all $x \geq 1$,

$$\mathbb{P}\Big( \sup_{r \in \mathcal{T}_s} \frac{\Big| \sum_{i=1}^{r} y_i \Big|}{\sqrt{r}} \geq x \Big) \leq C_1 x^{-2-\delta} \leq C_1 x^{-2}.$$

Therefore by a union bound, for any $0 < a < 1$,

$$\mathbb{P}\Big( \exists s \in \mathbb{Z}^+ : \sup_{r \in \mathcal{T}_s} \frac{\Big| \sum_{i=1}^{r} y_i \Big|}{\sqrt{r}} \geq \frac{\sqrt{C_1}}{a}(s+1) \Big) \leq \sum_{s=0}^{\infty} \frac{a^2}{(s+1)^2} = a^2\pi^2/6.$$

For any $r \in \left[ 2^s/\nu, 2^{s+1}/\nu \right], s \leq \log(r\nu)/\log(2)$, and therefore

$$\mathbb{P}\Big( \exists s \in \mathbb{Z}^+ : \sup_{r \in \mathcal{T}_s} \frac{\Big| \sum_{i=1}^{r} y_i \Big|}{\sqrt{r}} \geq \frac{\sqrt{C_1}}{a}\Big\{ \frac{\log(r\nu)}{\log(2)} + 1 \Big\} \Big) \leq a^2\pi^2/6.$$

Equation (2) directly gives

$$\mathbb{P}\Big( \sup_{r \in \mathcal{T}_s} \frac{\Big| \sum_{i=1}^{r} y_i \Big|}{\sqrt{r}} \geq \frac{C}{a}\{\log(r\nu) + 1\} \Big) \leq a^2.$$

$\square$

## H.2 Central Limit theorem

Below is the central limit theorem for $\alpha$-mixing random variable. We refer to Doukhan (1994) for more details.

**Lemma 4.** *Let $\left\{Z_t\right\}$ be a centred $\alpha$-mixing stationary time series. Suppose for the mixing coefficients and moments, for some $\delta > 0$ it holds*

$$\sum_{k=1}^{\infty} \alpha_k^{\delta/(2+\delta)} < \infty, \quad \mathbb{E}\left[\left|Z_1\right|^{2+\delta}\right] < \infty.$$

*Denote $S_n = \sum_{t=1}^{n} Z_t$ and $\sigma_n^2 = \mathbb{E}\left[\left|S_n\right|^2\right]$. Then*

$$\frac{S_{\lfloor nt \rfloor}}{\sigma_n} \to W(t),$$

*where convergence is in Skorohod topology and $W(t)$ is the standard Brownian motion on $[0, 1]$.*

# I Additional Technical Results

**Lemma 5.** *Let $\mathcal{J}$ be defined as in Definition 2 and suppose Assumption 1 **e** holds. Then for each change point $\eta_k$ there exists a seeded interval $\mathcal{I}_k = (s_k, e_k]$ such that*
**a.** *$\mathcal{I}_k$ contains exactly one change point $\eta_k$;*
**b.** *$\min\{\eta_k - s_k, e_k - \eta_k\} \geq \frac{1}{16}\Delta$; and*
**c.** *$\max\{\eta_k - s_k, e_k - \eta_k\} \leq \frac{9}{10}\Delta$;*

*Proof.* These are the desired properties of seeded intervals by construction. The proof is the same as theorem 3 of Kovács et al. (2020) and is provided here for completeness.

Let $k \in \{1, ..., \mathfrak{K}\}$, where $\mathfrak{K}$ is from definition 2. By construction of seeded intervals, we have that in $\mathcal{J}_k$, we can find an interval $(\lfloor (i-1)T2^{-k} \rfloor, \lceil (i-1)T2^{-k} + T2^{-k+1} \rceil]$, for some $i \in 1, ..., 2^k - 1$, such that

$$\min\{\eta_k - \lfloor (i-1)T2^{-k} \rfloor, \lceil (i-1)T2^{-k} + T2^{-k+1} \rceil - \eta_k\} \geq \frac{l_k}{4} \tag{50}$$

and

$$\max\{\eta_k - \lfloor (i-1)T2^{-k} \rfloor, \lceil (i-1)T2^{-k} + T2^{-k+1} \rceil - \eta_k\} \leq l_k, \tag{51}$$

where $l_k = T2^{-k+1}$, is the size of each interval in $\mathcal{J}_k$. By the choice of $\mathfrak{K}$, there exist $k$, such that $l_k = \frac{9}{10}\Delta$, from where the claim is followed. $\square$

**Lemma 6.** *Let $\{X_i\}_{i=1}^T$ be random grid points sampled from a common density function $f_t : \mathbb{R}^p \to \mathbb{R}$, satisfying Assumption 1-**a** and -**b**. Under Assumption (2), the density estimator of the sampling distribution $\mu$,*

$$\widehat{f}_t(x) = \frac{1}{T} \sum_{t=1}^T \mathcal{K}_h(x - X_i), \quad x \in \mathbb{R}^p,$$

*satisfies,*

$$\|\widehat{f}_T - f_t\|_{L_\infty} = O_p\left(\left(\frac{\log(T)}{T}\right)^{\frac{2r}{2r+p}}\right). \tag{52}$$

The verification of these bounds can be found in many places in the literature. See for example Yu (1993) and Tsybakov (2009).

**Remark 2.** *Even more, by Assumption 1,*

$$\|\widehat{f}_T - f_t\|_{L_2} \leq C_{\mathcal{X}}\|\widehat{f}_T - f_t\|_{L_\infty} = O\left(\left(\frac{\log(T)}{T}\right)^{\frac{2r}{2r+p}}\right) \tag{53}$$

*with high probability. Therefore, given that*

$$\kappa = \frac{\left\|\sqrt{\frac{\eta_{k+1} - \eta_k}{(\eta_{k+1} - \eta_{k-1})(\eta_k - \eta_{k-1})}} \sum_{i=\eta_{k-1}+1}^{\eta_k} f_i - \sqrt{\frac{(\eta_k - \eta_{k-1})}{(\eta_{k+1} - \eta_{k-1})(\eta_{k+1} - \eta_k)}} \sum_{i=\eta_k+1}^{\eta_{k+1}} f_i\right\|_{L_2}}{\sqrt{\frac{(\eta_k - \eta_{k-1})(\eta_{k+1} - \eta_k)}{\eta_{k+1} - \eta_{k-1}}}} \tag{54}$$

*and (5), by triangle inequality, (53), and Lemma 5, we have that*

$$|\kappa - \widehat{\kappa}| = O_p\left(\left(\frac{\log(T)}{\Delta}\right)^{\frac{2r}{2r+p}} + \frac{T^{\frac{p}{2r+p}}\log(T)}{\kappa\Delta}\right).$$

*From here, and Assumption 2, if $h_1 = O(\kappa^{\frac{1}{r}})$ and $h_2 = O(\widehat{\kappa}^{\frac{1}{r}})$, we conclude that*

$$\|F_{t,h_1} - F_{t,h_2}\|_{L_2}^2 = O\left(\frac{|\kappa - \widehat{\kappa}|}{\kappa^{\frac{p}{r}+1}}\right).$$

*In fact,*

$$\|F_{t,h_1} - F_{t,h_2}\|_{L_2}^2$$
$$= \int_{\mathbb{R}^p} \left(\frac{1}{h_1^p}\mathcal{K}\left(\frac{x - X_t}{h_1}\right) - \frac{1}{h_2^p}\mathcal{K}\left(\frac{x - X_t}{h_2}\right)\right)^2 dx$$
$$= \int_{\mathbb{R}^p} \left(\frac{1}{h_1^p}\mathcal{K}\left(\frac{x - X_t}{h_1}\right)\right)^2 - 2\frac{1}{h_1^p}\mathcal{K}\left(\frac{x - X_t}{h_1}\right)\frac{1}{h_2^p}\mathcal{K}\left(\frac{x - X_t}{h_2}\right) + \left(\frac{1}{h_2^p}\mathcal{K}\left(\frac{x - X_t}{h_2}\right)\right)^2 dx.$$

*Now, we analyze the two following terms,*

$$I_1 = \int_{\mathbb{R}^p} \Big( \frac{1}{h_1^p} \mathcal{K}(\frac{x - X_t}{h_1}) \Big)^2 - \frac{1}{h_1^p} \mathcal{K}(\frac{x - X_t}{h_1}) \frac{1}{h_2^p} \mathcal{K}(\frac{x - X_t}{h_2}) dx$$

*and*

$$I_2 = \int_{\mathbb{R}^p} \Big( \frac{1}{h_2^p} \mathcal{K}(\frac{x - X_t}{h_2}) \Big)^2 - \frac{1}{h_1^p} \mathcal{K}(\frac{x - X_t}{h_1}) \frac{1}{h_2^p} \mathcal{K}(\frac{x - X_t}{h_2}) dx.$$

*For $I_1$, letting $u = \frac{x - X_t}{h_1}$, we have that*

$$\int_{\mathbb{R}^p} \Big( \frac{1}{h_1^p} \mathcal{K}(\frac{x - X_t}{h_1}) \Big)^2 dx = \int_{\mathbb{R}^p} \frac{1}{h_1^p} \Big( \mathcal{K}(u) \Big)^2 du$$

*and, letting $v = \frac{x - X_t}{h_2}$, we have that*

$$\int_{\mathbb{R}^p} \frac{1}{h_1^p} \mathcal{K}(\frac{x - X_t}{h_1}) \frac{1}{h_2^p} \mathcal{K}(\frac{x - X_t}{h_2}) dx = \int_{\mathbb{R}^p} \frac{1}{h_1^p} \mathcal{K}(v \frac{h_2}{h_1}) \mathcal{K}(v) dv.$$

*Therefore, by Assumption 2 and the Mean Value Theorem,*

$$
\begin{aligned}
I_1 = \frac{1}{h_1^p} \int_{\mathbb{R}^p} \mathcal{K}(v) \Big( \mathcal{K}(v) - \mathcal{K}(v \frac{h_2}{h_1}) \Big) dv &\leq C \frac{1}{h_1^p} \Big| 1 - \frac{h_2}{h_1} \Big| \int_{\mathbb{R}^p} \mathcal{K}(v) ||v|| dv \\
&\leq C_1 \frac{|h_1 - h_2|}{h_1^{p+1}} \\
&= O\Big( \frac{|\kappa - \widehat{\kappa}|}{\kappa^{\frac{p+1}{r}}} \kappa^{\frac{1}{r} - 1} \Big) = O\Big( \frac{|\kappa - \widehat{\kappa}|}{\kappa^{\frac{p}{r} + 1}} \Big).
\end{aligned}
$$

*Similarly, we have,*

$$
\begin{aligned}
I_2 &= \int_{\mathbb{R}^p} \Big( \frac{1}{h_2^p} \mathcal{K}(\frac{x - X_t}{h_2}) \Big)^2 - \frac{1}{h_1^p} \mathcal{K}(\frac{x - X_t}{h_1}) \frac{1}{h_2^p} \mathcal{K}(\frac{x - X_t}{h_2}) dx \\
&= \frac{1}{h_2^p} \int_{\mathbb{R}^p} \mathcal{K}(v) \Big( \mathcal{K}(v) - \mathcal{K}(v \frac{h_1}{h_2}) \Big) dv \leq C \frac{1}{h_2^p} \Big| 1 - \frac{h_1}{h_2} \Big| \int_{\mathbb{R}^p} \mathcal{K}(v) ||v|| dv = O\Big( \frac{|\kappa - \widehat{\kappa}|}{\kappa^{\frac{p}{r} + 1}} \Big).
\end{aligned}
$$

## I.1 Multivariate change point detection lemmas

We present some technical results corresponding to the generalization of the univariate CUSUM to the Multivariate case. For more details, we refer the interested readers to Padilla et al. (2021) and Wang et al. (2020).

Let $\{X_t\}_{t=1}^T \subset \mathbb{R}^p$ a process with unknown densities $\{f_t\}_{t=1}^T$.

**Assumption 5.** *We assume there exist $\{\eta_k\}_{k=1}^K \subset \{2, ..., T\}$ with $1 = \eta_0 < \eta_1 < ... < \eta_k \leq T < \eta_{K+1} = T + 1$, such that*

$$f_t \neq f_{t+1} \text{ if and only if } t \in \{\eta_1, ..., \eta_K\}, \tag{55}$$

*Assume*

$$\min_{k=1,...,K+1} (\eta_k - \eta_{k-1}) \geq \Delta > 0,$$
$$0 < ||f_{\eta_k+1} - f_{\eta_k}||_{L_\infty} = \kappa_k \text{ for all } k = 1, \dots, K.$$

In the rest of this section, we use the notation

$$\widetilde{f}_t^{(s,e)}(x) = \sqrt{\frac{e - t}{(e - s)(t - s)}} \sum_{j=s+1}^t f_j(x) - \sqrt{\frac{t - s}{(e - s)(e - t)}} \sum_{j=t+1}^e f_j(x),$$

for all $0 \leq s < t < e \leq T$ and $x \in \mathbb{R}^p$.

**Lemma 7.** *If $[s, e]$ contain two and only two change points $\eta_r$ and $\eta_{r+1}$, then*

$$\sup_{s \leq t \leq e} ||\widetilde{f}_t^{s,e}||_{L_2} \leq \sqrt{e - \eta_{r+1}} ||f_{r+1} - f_r||_{L_2} + \sqrt{\eta_r - s} ||f_r - f_{r-1}||_{L_2}.$$

*Proof.* This is Lemma 15 in Wang et al. (2020). Consider the sequence $\left\{g_t\right\}_{t=s+1}^{e}$ be such that

$$g_t = \begin{cases} f_{\eta_k}, & \text{if } s+1 \le t < \eta_k, \\ f_t, & \text{if } \eta_k \le t \le e. \end{cases}$$

For any $t \ge \eta_k$,

$$\widetilde{f}_t^{s,e} - \widetilde{g}_t^{s,e}$$

$$= \sqrt{\frac{e-t}{(e-s)(t-s)}}\left( \sum_{i=s+1}^{t} f_i - \sum_{i=s+1}^{\eta_k} f_{\eta_k} - \sum_{i=\eta_k+1}^{t} f_i \right)$$

$$- \sqrt{\frac{t-s}{(e-s)(e-t)}}\left( \sum_{i=t+1}^{e} f_i - \sum_{i=t+1}^{e} f_i \right)$$

$$= \sqrt{\frac{e-t}{(e-s)(t-s)}}\left( \eta_k - s \right)\left( f_{\eta_k} - f_{\eta_{k-1}} \right).$$

So for $t \ge \eta_k, \|\widetilde{f}_t^{s,e} - \widetilde{g}_t^{s,e}\|_{L_2} \le \sqrt{\eta_k - s}\kappa_k$. Since $\sup_{s \le t \le e} \|\widetilde{f}_t^{s,e}\|_{L_2} = \max\left\{ \|\widetilde{f}_{\eta_k}^{s,e}\|_{L_2}, \|\widetilde{f}_{\eta_{k+1}}^{s,e}\|_{L_2} \right\}$, and that

$$\max\left\{ \|\widetilde{f}_{\eta_k}^{s,e}\|_{L_2}, \|\widetilde{f}_{\eta_{k+1}}^{s,e}\|_{L_2} \right\} \le \sup_{s \le t \le e} \|\widetilde{g}_t^{s,e}\|_{L_2} + \sqrt{\eta_k - s}\kappa_k$$

$$\le \sqrt{e - \eta_{k+1}}\kappa_{k+1} + \sqrt{\eta_r - s}\kappa_k$$

where the last inequality follows form the fact that $g_t$ has only one change point in $[s, e]$. $\qquad\square$

**Lemma 8.** *Suppose $e - s \le C_R\Delta$, where $C_R > 0$ is an absolute constant, and that*

$$\eta_{k-1} \le s \le \eta_k \le \ldots \le \eta_{k+q} \le e \le \eta_{k+q+1}, \quad q \ge 0$$

*Denote*

$$\kappa_{\max}^{s,e} = \max\left\{ \sup_{x \in \mathbb{R}^p} \left| f_{\eta_p}(x) - f_{\eta_{p-1}}(x) \right| : k \le p \le k+q \right\}.$$

*Then for any $k - 1 \le p \le k + q$, it holds that*

$$\sup_{x \in \mathbb{R}^p} \left| \frac{1}{e-s} \sum_{i=s+1}^{e} f_i(x) - f_{\eta_p}(x) \right| \le C_R\kappa_{\max}^{s,e}.$$

*Proof.* This is Lemma 18 in Wang et al. (2020). Since $e - s \le C_R\Delta$, the interval $[s, e]$ contains at most $C_R + 1$ change points. Observe that

$$\left\| \frac{1}{e-s} \sum_{i=s}^{e} f_i - f_{\eta_p} \right\|_{L_\infty}$$

$$= \frac{1}{e-s}\left\| \sum_{i=s}^{\eta_k} \left( f_{\eta_{k-1}} - f_{\eta_p} \right) + \sum_{i=\eta_k+1}^{\eta_{k+1}} \left( f_{\eta_k} - f_{\eta_p} \right) + \ldots + \sum_{i=\eta_{k+q}+1}^{e} \left( f_{\eta_{k+q}} - f_{\eta_p} \right) \right\|_{L_\infty}$$

$$\le \frac{1}{e-s} \sum_{i=s}^{\eta_k} |p-k|\kappa_{\max}^{s,e} + \sum_{i=\eta_k+1}^{\eta_{k+1}} |p-k-1|\kappa_{\max}^{s,e} + \ldots + \sum_{i=\eta_{k+q}+1}^{e} |p-k-q-1|\kappa_{\max}^{s,e}$$

$$\le \frac{1}{e-s} \sum_{i=s}^{e} \left( C_R + 1 \right)\kappa_{\max}^{s,e},$$

where $\left| p_1 - p_2 \right| \le C_R + 1$ for any $\eta_{p_1}, \eta_{p_2} \in [s, e]$ is used in the last inequality. $\qquad\square$

**Lemma 9.** *Let $(s, e) \subset (0, n)$ contains two or more change points such that*

$$\eta_{k-1} \le s \le \eta_k \le \ldots \le \eta_{k+q} \le e \le \eta_{k+q+1}, \quad q \ge 1$$

*If $\eta_k - s \le c_1\Delta$, for $c_1 > 0$, then*

$$\left\| \widetilde{f}_{\eta_k}^{s,e} \right\|_{L_\infty} \le \sqrt{c_1}\left\| \widetilde{f}_{\eta_{k+1}}^{s,e} \right\|_{L_\infty} + 2\kappa_k\sqrt{\eta_k - s}$$

*Proof.* This is Lemma 20 in Wang et al. (2020). Consider the sequence $\left\{g_t\right\}_{t=s+1}^{e}$ be such that

$$g_t = \begin{cases} f_{\eta_{r+1}}, & s+1 \leq t \leq \eta_k, \\ f_t, & \eta_k + 1 \leq t \leq e \end{cases}$$

For any $t \geq \eta_r$, it holds that

$$\|\widetilde{f}_{\eta_k}^{s,e} - \widetilde{g}_{\eta_k}^{s,e}\|_{L_\infty} = \left\|\sqrt{\frac{(e-s)-t}{(e-s)(t-s)}}\left(\eta_k - s\right)\left(f_{\eta_{k+1}} - f_{\eta_k}\right)\right\|_{L_\infty} \leq \sqrt{\eta_k - s}\kappa_k.$$

Thus,

$$\|\widetilde{f}_{\eta_k}^{s,e}\|_{L_\infty} \leq \|\widetilde{g}_{\eta_k}^{s,e}\|_{L_\infty} + \sqrt{\eta_k - s}\kappa_k \leq \sqrt{\frac{\left(\eta_k - s\right)\left(e - \eta_{k+1}\right)}{\left(\eta_{k+1} - s\right)\left(e - \eta_k\right)}}\|\widetilde{g}_{\eta_{k+1}}^{s,e}\|_{L_\infty} + \sqrt{\eta_k - s}\kappa_k$$

$$\leq \sqrt{\frac{c_1\Delta}{\Delta}}\|\widetilde{g}_{\eta_{k+1}}^{s,e}\|_{L_\infty} + \sqrt{\eta_k - s}\kappa_k \leq \sqrt{c_1}\|\widetilde{f}_{\eta_{k+1}}^{s,e}\|_{L_\infty} + 2\sqrt{\eta_k - s}\kappa_k,$$

where the first inequality follows from the observation that the first change point of $g_t$ in $(s,e)$ is at $\eta_{k+1}$. $\square$

**Lemma 10.** *Under Assumption 5, for any interval $(s,e) \subset (0,T)$ satisfying*

$$\eta_{k-1} \leq s \leq \eta_k \leq \ldots \leq \eta_{k+q} \leq e \leq \eta_{k+q+1}, \quad q \geq 0.$$

*Let*

$$b \in \arg\max_{t=s+1,\ldots,e} \sup_{x \in \mathbb{R}^p} \left|\widetilde{f}_t^{(s,e]}(x)\right|.$$

*Then $b \in \left\{\eta_1, \ldots, \eta_K\right\}$. For any fixed $z \in \mathbb{R}^p$, if $\widetilde{f}_t^{(s,e]}(z) > 0$ for some $t \in (s,e)$, then $\widetilde{f}_t^{(s,e]}(z)$ is either strictly monotonic or decreases and then increases within each of the interval $\left(s, \eta_k\right), \left(\eta_k, \eta_{k+1}\right), \ldots, \left(\eta_{k+q}, e\right)$.*

*Proof.* We prove this by contradiction. Assume that $b \notin \{\eta_1, \ldots, \eta_K\}$. Let $z_1 \in \arg\max_{x \in \mathbb{R}^p}\left|\bar{f}_b^{s,e}(x)\right|$. Due to the definition of $b$, we have

$$b \in \arg\max_{t=s+1,\ldots,e}\left|\widetilde{f}_t^{(s,e]}\left(z_1\right)\right|.$$

It is easy to see that the collection of change points $\{f_t(z_1)\}_{t=s+1}^{e}$ is a subset of the change points of $\{f\}_{t=s+1}^{e}$. Then, from Lemma 2.2 in Venkatraman (1992) that

$$\widetilde{f}_b^{(s,e]}\left(z_1\right) < \max_{j \in \{k,\ldots,k+q\}} \widetilde{f}_{\eta_j}^{(s,e]}\left(z_1\right) \leq \max_{t=s+1,\ldots,e} \sup_{x \in \mathbb{R}^p}\left|\widetilde{f}_t^{(s,e]}(x)\right|$$

which is a contradiction. $\square$

Recall that in Algorithm 1, when searching for change points in the interval $(s,e)$, we actually restrict to values $t \in \left(s + \rho, e - \rho\right)$. We now show that for intervals satisfying condition $SE$ from Lemma 1, taking the maximum of the CUSUM statistic over $\left(s + \rho, e - \rho\right)$ is equivalent to searching on $(s,e)$, when there are change points in $\left(s + \rho, e - \rho\right)$.

**Lemma 11.** *Let $z_0 \in \mathbb{R}^p, (s,e) \subset (0,T)$. Suppose that there exists a true change point $\eta_k \in (s,e)$ such that*

$$\min\left\{\eta_k - s, e - \eta_k\right\} \geq c_1\Delta, \tag{56}$$

*and*

$$\left|\widetilde{f}_{\eta_k}^{(s,e]}\left(z_0\right)\right| \geq \left(c_1/2\right)\frac{\kappa\Delta}{\sqrt{e-s}}, \tag{57}$$

*where $c_1 > 0$ is a sufficiently small constant. In addition, assume that*

$$\max_{t=s+1,\ldots,e} \left| \widetilde{f}_t^{(s,e]}(z_0) \right| - \left| \widetilde{f}_{\eta_k}^{(s,e]}(z_0) \right| \leq c_2 \Delta^4 (e-s)^{-7/2} \kappa, \tag{58}$$

*where $c_2 > 0$ is a sufficiently small constant. Then for any $d \in (s, e)$ satisfying*

$$\left| d - \eta_k \right| \leq c_1 \Delta/32, \tag{59}$$

*it holds that*

$$\left| \widetilde{f}_{\eta_k}^{(s,e]}(z_0) \right| - \left| \widetilde{f}_d^{(s,e]}(z_0) \right| > c \left| d - \eta_k \right| \Delta \left| \widetilde{f}_{\eta_k}^{(s,e]}(z_0) \right| (e-s)^{-2}, \tag{60}$$

*where $c > 0$ is a sufficiently small constant, depending on all the other absolute constants.*

*Proof.* Without loss of generality, we assume that $d \geq \eta_k$ and $\widetilde{f}_{\eta_k}(z_0) \geq 0$. Following the arguments in Lemma 2.6 in Venkatraman (1992), it suffices to consider two cases: (i) $\eta_{k+1} > e$ and (ii) $\eta_{k+1} \leq e$
**Case (i)**. Note that

$$\widetilde{f}_{\eta_k}^{(s,e]}(z_0) = \sqrt{\frac{(e-\eta_k)(\eta_k - s)}{e-s}} \left\{ f_{\eta_k}(z_0) - f_{\eta_{k+1}}(z_0) \right\}$$

and

$$\widetilde{f}_d^{(s,e]}(z_0) = (\eta_k - s) \sqrt{\frac{e-d}{(e-s)(d-s)}} \left\{ f_{\eta_k}(z_0) - f_{\eta_{k+1}}(z_0) \right\}.$$

Therefore, it follows from (56) that

$$\widetilde{f}_{\eta_k}^{(s,e]}(z_0) - \widetilde{f}_d^{(s,e]}(z_0) = \left( 1 - \sqrt{\frac{(e-d)(\eta_k - s)}{(d-s)(e-\eta_k)}} \right) \widetilde{f}_{\eta_k}^{(s,e]}(z_0) \geq c\Delta \left| d - \eta_k \right| (e-s)^{-2} \widetilde{f}_{\eta_k}^{(s,e]}(z_0).$$

$$\tag{61}$$

The inequality follows from the following arguments. Let $u = \eta_k - s$, $v = e - \eta_k$ and $w = d - \eta_k$. Then

$$1 - \sqrt{\frac{(e-d)(\eta_k - s)}{(d-s)(e-\eta_k)}} - c\Delta \left| d - \eta_k \right| (e-s)^2$$

$$= 1 - \sqrt{\frac{(v-w)u}{(u+w)v}} - c\frac{\Delta w}{(u+v)^2}$$

$$= \frac{w(u+v)}{\sqrt{(u+w)v}(\sqrt{(v-w)u} + \sqrt{(u+w)v})} - c\frac{\Delta w}{(u+v)^2}.$$

The numerator of the above equals

$$w(u+v)^3 - c\Delta w(u+w)v - c\Delta w \sqrt{uv(u+w)(v-w)}$$

$$\geq 2c_1 \Delta w \left\{ (u+v)^2 - \frac{c(u+w)v}{2c_1} - \frac{c\sqrt{uv(u+w)(v-w)}}{2c_1} \right\}$$

$$\geq 2c_1 \Delta w \left\{ \left( 1 - c/(2c_1) \right)(u+v)^2 - 2^{-1/2} c/c_1 uv \right\} > 0$$

as long as

$$c < \frac{\sqrt{2}c_1}{4 + 1/(\sqrt{2}c_1)}.$$

**Case (ii)**. Let $g = c_1 \Delta/16$. We can write

$$\widetilde{f}_{\eta_k}^{(s,e]}(z_0) = a\sqrt{\frac{e-s}{(\eta_k - s)(e-\eta_k)}}, \quad \widetilde{f}_{\eta_k+g}^{(s,e]}(z_0) = (a+g\theta)\sqrt{\frac{e-s}{(e-\eta_k - g)(\eta_k + g - s)}},$$

where

$$a = \sum_{j=s+1}^{\eta_k} \left\{ f_j\left(z_0\right) - \frac{1}{e-s} \sum_{j=s+1}^{e} f_j\left(z_0\right) \right\}$$

$$\theta = \frac{a\sqrt{\left(\eta_k+g-s\right)\left(e-\eta_k-g\right)}}{g} \left\{ \frac{1}{\sqrt{\left(\eta_k-s\right)\left(e-\eta_k\right)}} - \frac{1}{\left(\eta_k+g-s\right)\left(e-\eta_k-g\right)} + \frac{b}{a\sqrt{e-s}} \right\},$$

and $b = \widetilde{f}_{\eta_k+g}^{(s,e]}\left(z_0\right) - \widetilde{f}_{\eta_k}^{(s,e]}\left(z_0\right)$. To ease notation, let $d - \eta_k = l \le g/2, N_1 = \eta_k - s$ and $N_2 = e - \eta_k - g$. We have

$$E_l = \widetilde{f}_{\eta_k}^{(s,e]}\left(z_0\right) - \widetilde{f}_{d}^{(s,e]}\left(z_0\right) = E_{1l}\left(1 + E_{2l}\right) + E_{3l}, \tag{62}$$

where

$$E_{1l} = \frac{al(g-l)\sqrt{e-s}}{\sqrt{N_1\left(N_2+g\right)}\sqrt{\left(N_1+l\right)\left(g+N_2-l\right)}\left(\sqrt{\left(N_1+l\right)\left(g+N_2-l\right)} + \sqrt{N_1\left(g+N_2\right)}\right)},$$

$$E_{2l} = \frac{\left(N_2-N_1\right)\left(N_2-N_1-l\right)}{\left(\sqrt{\left(N_1+l\right)\left(g+N_2-l\right)} + \sqrt{\left(N_1+g\right)N_2}\right)\left(\sqrt{N_1\left(g+N_2\right)} + \sqrt{\left(N_1+g\right)N_2}\right)},$$

and

$$E_{3l} = -\frac{bl}{g}\sqrt{\frac{\left(N_1+g\right)N_2}{\left(N_1+l\right)\left(g+N_2-l\right)}}.$$

Next, we notice that $g - l \ge c_1\Delta/32$. It holds that

$$E_{1l} \ge c_{1l}\left|d - \eta_k\right|\Delta\widetilde{f}_{\eta_k}^{(s,e]}\left(z_0\right)(e-s)^{-2}, \tag{63}$$

where $c_{1l} > 0$ is a sufficiently small constant depending on $c_1$. As for $E_{2l}$, due to (59), we have

$$E_{2l} \ge -1/2. \tag{64}$$

As for $E_{3l}$, we have

$$E_{3l} \ge -c_{3l,1}b\left|d - \eta_k\right|(e-s)\Delta^{-2} \ge -c_{3l,2}b\left|d - \eta_k\right|\Delta^{-3}(e-s)^{3/2}\widetilde{f}_{\eta_k}^{(s,e]}\left(z_0\right)\kappa^{-1} \tag{65}$$

$$\ge -c_{1l}/2\left|d - \eta_k\right|\Delta\widetilde{f}_{\eta_k}^{(s,e]}\left(z_0\right)(e-s)^{-2}, \tag{66}$$

where the second inequality follows from (57) and the third inequality follows from (58), $c_{3l,1}, c_{3l,2} > 0$ are sufficiently small constants, depending on all the other absolute constants. Combining (62), (63), (64) and (65), we have

$$\widetilde{f}_{\eta_k}^{(s,e]}\left(z_0\right) - \widetilde{f}_{d}^{(s,e]}\left(z_0\right) \ge c\left|d - \eta_k\right|\Delta\widetilde{f}_{\eta_k}^{(s,e]}\left(z_0\right)(e-s)^{-2}, \tag{67}$$

where $c > 0$ is a sufficiently small constant. In view of (61) and (67), the proof is complete. □

Consider the following events

$$\mathcal{A}((s,e],\rho,\gamma) = \left\{ \max_{t=s+\rho+1,\dots,e-\rho} \sup_{z\in\mathbb{R}^p} |\widetilde{F}_{t,h}^{s,e}(z) - \widetilde{f}_t^{s,e}(z)| \le \gamma \right\};$$

$$\mathcal{B}(r,\rho,\gamma) = \left\{ \max_{N=\rho,\dots,T-r} \sup_{z\in\mathbb{R}^p} \left| \frac{1}{\sqrt{N}} \sum_{t=r+1}^{r+N} (F_{t,h} - f_t) \right| \le \gamma \right\}$$

$$\bigcup \left\{ \max_{N=\rho,\dots,r} \left| \frac{1}{\sqrt{N}} \sum_{t=r-N+1}^{r} \sup_{z\in\mathbb{R}^p} (F_{t,h}(z) - f_t(z)) \right| \le \gamma \right\}.$$

**Lemma 12.** *Suppose Assumption 5 holds. Let $[s, e]$ be an subinterval of $[1, T]$ with $e - s \leq C_R \Delta$, and contain at least one change point $\eta_r$ with $\min\{\eta_r - s, e - \eta_r\} \geq cT$ for some constant $c > 0$. Let $\kappa_{\max}^{s,e} = \max\{\kappa_p : \min\{\eta_p - s, e - \eta_p\} \geq cT\}$. Let*

$$b \in \arg \max_{t = s + \rho, \ldots, e - \rho} \|\widetilde{F}_{t,h}^{s,e}\|_{L_2}.$$

*For some $c_1 > 0$, $\lambda > 0$ and $\delta > 0$, suppose that the following events hold*

$$\mathcal{A}((s, e], \rho, \gamma), \tag{68}$$

$$\mathcal{B}(s, \rho, \gamma) \cup \mathcal{B}(e, \rho, \gamma) \cup \bigcup_{\eta \in \{\eta_k\}_{k=1}^{K}} \mathcal{B}(\eta, \rho, \gamma) \tag{69}$$

*and that*

$$\max_{t = s + \rho, \ldots, e - \rho} \|\widetilde{F}_{t,h}^{s,e}\|_{L_2} = \|\widetilde{F}_{b,h}^{s,e}\|_{L_2} \geq c_1 \kappa_{\max}^{s,e} \sqrt{T} \tag{70}$$

*If there exists a sufficiently small $c_2 > 0$ such that*

$$\gamma \leq c_2 \kappa_{\max}^{s,e} \sqrt{T} \quad \text{and that} \quad \rho \leq c_2 T, \tag{71}$$

*then there exists a change point $\eta_k \in (s, e)$ such that*

$$\min\{e - \eta_k, \eta_k - s\} > c_3 T \quad \text{and} \quad |\eta_k - b| \leq C_3 \max\{\gamma^2 \kappa_k^{-2}, \rho\},$$

*where $c_3$ is some sufficiently small constant independent of $T$.*

*Proof.* Let $z_1 \in \arg \max_{z \in \mathbb{R}^p} \left| \widetilde{f}_b^{(s,e)}(z) \right|$. Without loss of generality, assume that $\widetilde{f}_b^{(s,e)}\left(z_1\right) > 0$ and that $\widetilde{f}_b^{(s,e)}\left(z_1\right)$ as a function of $t$ is locally decreasing at $b$. Observe that there has to be a change point $\eta_k \in (s, b)$, or otherwise $\widetilde{f}_b^{(s,e)}\left(z_1\right) > 0$ implies that $\widetilde{f}_t^{(s,e)}\left(z_1\right)$ is decreasing, as a consequence of Lemma 10. Thus, there exists a change point $\eta_k \in (s, b)$ satisfying that

$$\sup_{z \in \mathbb{R}^p} \left| \widetilde{f}_{\eta_k}^{(s,e)}(z) \right| \geq \left| \widetilde{f}_{\eta_k}^{(s,e)}\left(z_1\right) \right| > \left| \widetilde{f}_b^{(s,e)}\left(z_1\right) \right| \geq \sup_{z \in \mathbb{R}^p} \left| \widetilde{F}_b^{(s,e)}(z) \right| - \gamma \geq c \kappa_k \sqrt{\Delta} \tag{72}$$

where the second inequality follows from Lemma 10, the third because of the good event $\mathcal{A}$, and fourth inequalities by (70) and Assumption 1, and $c > 0$ is an absolute constant. Observe that $(s, e)$ has to contain at least one change point or otherwise $\sup_{z \in \mathbb{R}} \left| \widetilde{f}_{\eta_k}^{(s,e)}(z) \right| = 0$ which contradicts (72).

**Step 1**. In this step, we are to show that

$$\min \left\{ \eta_k - s, e - \eta_k \right\} \geq \min \left\{ 1, c_1^2 \right\} \Delta / 16 \tag{73}$$

Suppose that $\eta_k$ is the only change point in $(s, e)$. Then (73) must hold or otherwise it follows from (14) that

$$\sup_{z \in \mathbb{R}^p} \left| \widetilde{f}_{\eta_k}^{s,e}(z) \right| \leq \kappa_k \frac{c_1 \sqrt{\Delta}}{4},$$

which contradicts (72).

Suppose $(s, e)$ contains at least two change points. Then arguing by contradiction, if $\eta_k - s < \min \left\{ 1, c_1^2 \right\} \Delta / 16$, it must be the cast that $\eta_k$ is the left most change point in $(s, e)$. Therefore

$$\sup_{z \in \mathbb{R}^p} \left| \widetilde{f}_{\eta_k}^{s,e}(z) \right| \leq c_1 / 4 \sup_{z \in \mathbb{R}^p} \left| \widetilde{f}_{\eta_{k+1}}^{s,e}(z) \right| + 2 \kappa_k \sqrt{\eta_k - s} \tag{74}$$

$$< c_1 / 4 \max_{s + \rho < t < e - \rho} \sup_{z \in \mathbb{R}^p} \left| \widetilde{f}_t^{s,e}(z) \right| + 2 \sqrt{\Delta} \kappa_k \tag{75}$$

$$\leq c_1 / 4 \max_{s + \rho < t < e - \rho} \sup_{z \in \mathbb{R}^p} \left| \widetilde{F}_t^{s,e}(z) \right| + c_1 / 4 \gamma + 2 \sqrt{\Delta} \kappa_k \tag{76}$$

$$\leq \sup_{z \in \mathbb{R}^p} \left| \widetilde{F}_b^{s,e}(z) \right| - \gamma \tag{77}$$

where the first inequality follows from Lemma 9, the second follows from the assumption of $\eta_k - s$, the third from the definition of the event $\mathcal{A}$ and the last from (70) and Assumption 1. The last display contradicts (72), thus (73) must hold.

**Step 2**. Let

$$z_0 \in \arg\max_{z \in \mathbb{R}^p} \left| \widetilde{f}^{s,e}_{\eta_k}(z) \right|.$$

It follows from Lemma 11 that there exits $d \in \left( \eta_k, \eta_k + c_1 \Delta/32 \right)$ such that

$$\widetilde{f}^{s,e}_{\eta_k}\left(z_0\right) - \widetilde{f}^{s,e}_{d}\left(z_0\right) \geq 2\gamma. \tag{78}$$

We claim that $b \in \left( \eta_k, d \right) \subset \left( \eta_k, \eta_k + c_1 \Delta/16 \right)$. By contradiction, suppose that $b \geq d$. Then

$$\widetilde{f}^{s,e}_{b}\left(z_0\right) \leq \widetilde{f}^{s,e}_{d}\left(z_0\right) \leq \max_{s < t < e} \sup_{z \in \mathbb{R}^p} \left| \widetilde{f}^{s,e}_{t}(z) \right| - 2\gamma \leq \sup_{z \in \mathbb{R}^p} \left| \widetilde{F}^{s,e}_{b}(z) \right| - \gamma, \tag{79}$$

where the first inequality follows from Lemma 10, the second follows from (78) and the third follows from the definition of the event $\mathcal{A}$. Note that (79) is a contradiction to the bound in (72), therefore we have $b \in \left( \eta_k, \eta_k + c_1 \Delta/32 \right)$.

**Step 3**. Let

$$j^* \in \arg\max_{j=1,\ldots,T} \left| \widetilde{F}^{s,e}_{b}(X(j)) \right|, \quad f^{s,e} = \left( f_{s+1}\left(X\left(j^*\right)\right), \ldots, f_e\left(X\left(j^*\right)\right) \right)^\top \in \mathbb{R}^{(e-s)}$$

and

$$F^{s,e} = \left( \frac{1}{h^p} k\left( \frac{X\left(j^*\right) - X(s)}{h} \right), \ldots, \frac{1}{h^p} k\left( \frac{X\left(j^*\right) - X(e)}{h} \right) \right) \in \mathbb{R}^{(e-s)}.$$

By the definition of $b$, it holds that

$$\left\| F^{s,e} - \mathcal{P}^{s,e}_{b}\left(F^{s,e}\right) \right\|^2 \leq \left\| F^{s,e} - \mathcal{P}^{s,e}_{\eta_k}\left(F^{s,e}\right) \right\|^2 \leq \left\| F^{s,e} - \mathcal{P}^{s,e}_{\eta_k}\left(f^{s,e}\right) \right\|^2$$

where the operator $\mathcal{P}^{s,e}(\cdot)$ is defined in Lemma 21 in Wang et al. (2020). For the sake of contradiction, throughout the rest of this argument suppose that, for some sufficiently large constant $C_3 > 0$ to be specified,

$$\eta_k + C_3 \lambda_{\mathcal{A}}^2 \kappa_k^{-2} < b. \tag{80}$$

We will show that this leads to the bound

$$\left\| F^{s,e} - \mathcal{P}^{s,e}_{b}\left(F^{s,e}\right) \right\|^2 > \left\| F^{s,e} - \mathcal{P}^{s,e}_{\eta_k}\left(f^{s,e}\right) \right\|^2, \tag{81}$$

which is a contradiction. If we can show that

$$2\left\langle F^{s,e} - f^{s,e}, \mathcal{P}^{s,e}_{b}\left(F^{s,e}\right) - \mathcal{P}^{s,e}_{\eta_k}\left(f^{s,e}\right) \right\rangle < \left\| f^{s,e} - \mathcal{P}^{s,e}_{b}\left(f^{s,e}\right) \right\|^2 - \left\| f^{s,e} - \mathcal{P}^{s,e}_{\eta_k}\left(f^{s,e}\right) \right\|^2, \tag{82}$$

then (81) holds. To derive (82) from (80), we first note that $\min\left\{ e - \eta_k, \eta_k - s \right\} \geq \min\left\{ 1, c_1^2 \right\} \Delta/16$ and that $\left| b - \eta_k \right| \leq c_1 \Delta/32$ implies that

$$\min\{e - b, b - s\} \geq \min\left\{ 1, c_1^2 \right\} \Delta/16 - c_1 \Delta/32 \geq \min\left\{ 1, c_1^2 \right\} \Delta/32 \tag{83}$$

As for the right-hand side of (82), we have

$$\left\| f^{s,e} - \mathcal{P}^{s,e}_{b}\left(f^{s,e}\right) \right\|^2 - \left\| f^{s,e} - \mathcal{P}^{s,e}_{\eta_k}\left(f^{s,e}\right) \right\|^2 = \left( \widetilde{f}^{s,e}_{\eta_k}\left(X\left(j^*\right)\right) \right)^2 - \left( \widetilde{f}^{s,e}_{b}\left(X\left(j^*\right)\right) \right)^2 \tag{84}$$

$$\geq \left( \widetilde{f}^{s,e}_{\eta_k}\left(X\left(j^*\right)\right) - \widetilde{f}^{s,e}_{b}\left(X\left(j^*\right)\right) \right) \left| \widetilde{f}^{s,e}_{\eta_k}\left(X\left(j^*\right)\right) \right| \tag{85}$$

On the event $\mathcal{A} \cap \mathcal{B}$, we are to use Lemma 11. Note that (57) holds due to the fact that here we have

$$\left|\widetilde{f}_{\eta_k}^{s,e}\left(X\left(j^*\right)\right)\right| \geq \left|\widetilde{f}_b^{s,e}\left(X\left(j^*\right)\right)\right| \geq \left|\widetilde{F}_b^{s,e}\left(X\left(j^*\right)\right)\right| - \gamma \geq c_1 \kappa_k \sqrt{\Delta} - \gamma \geq \left(c_1\right)/2\kappa_k\sqrt{\Delta},$$
(86)

where the first inequality follows from the fact that $\eta_k$ is a true change point, the second inequality holds due to the event $\mathcal{A}$, the third inequality follows from (70), and the final inequality follows from (71). Towards this end, it follows from Lemma 11 that

$$\left| \left|\widetilde{f}_{\eta_k}^{s,e}\left(X\left(j^*\right)\right)\right| - \left|\widetilde{f}_b^{s,e}\left(X\left(j^*\right)\right)\right| \right| > c|b - \eta_k|\Delta|\widetilde{f}_{\eta_k}^{s,e}\left(X\left(j^*\right)\right) | \, (e-s)^{-2}.$$
(87)

Combining (84), (86) and (87), we have

$$\left\| f^{s,e} - \mathcal{P}_b^{s,e}\left(f^{s,e}\right) \right\|^2 - \left\| f^{s,e} - \mathcal{P}_{\eta_k}^{s,e}\left(f^{s,e}\right) \right\|^2 \geq \frac{cc_1^2}{4}\Delta^2\kappa_k\mathcal{A}^2(e-s)^{-2}\left|b - \eta_k\right|.$$
(88)

The left-hand side of (82) can be decomposed as follows.

$$2\left\langle F^{s,e} - f^{s,e}, \mathcal{P}_b^{s,e}\left(F^{s,e}\right) - \mathcal{P}_{\eta_k}^{s,e}\left(f^{s,e}\right) \right\rangle$$
(89)

$$= 2\left\langle F^{s,e} - f^{s,e}, \mathcal{P}_b^{s,e}\left(F^{s,e}\right) - \mathcal{P}_b^{s,e}\left(f^{s,e}\right) \right\rangle + 2\left\langle Y^{s,e} - f^{s,e}, \mathcal{P}_b^{s,e}\left(f^{s,e}\right) - \mathcal{P}_{\eta_k}^{s,e}\left(f^{s,e}\right) \right\rangle$$
(90)

$$= (I) + 2\Big( \sum_{i=1}^{\eta_k - s} + \sum_{i=\eta_k - s+1}^{b-s} + \sum_{i=b-s+1}^{e-s} \Big)\left(F^{s,e} - f^{s,e}\right)_i \left(\mathcal{P}_b^{s,e}\left(f^{s,e}\right) - \mathcal{P}_{\eta_k}^{s,e}\left(f^{s,e}\right)\right)_i$$
(91)

$$= (I) + (II.1) + (II.2) + (II.3).$$
(92)

As for the term (I), we have

$$(I) \leq 2\gamma^2.$$
(93)

As for the term (II.1), we have

$$(II.1) = 2\sqrt{\eta_k - s}\Big\{ \frac{1}{\sqrt{\eta_k - s}} \sum_{i=1}^{\eta_k - s} \left(F^{s,e} - f^{s,e}\right)_i \Big\}\Big\{ \frac{1}{b - s} \sum_{i=1}^{b-s} \left(f^{s,e}\right)_i - \frac{1}{\eta_k - s} \sum_{i=1}^{\eta_k - s} \left(f^{s,e}\right)_i \Big\}.$$

In addition, it holds that

$$\left| \frac{1}{b - s} \sum_{i=1}^{b-s} \left(f^{s,e}\right)_i - \frac{1}{\eta_k - s} \sum_{i=1}^{\eta_k - s} \left(f^{s,e}\right)_i \right| = \frac{b - \eta_k}{b - s}\left| - \frac{1}{\eta_k - s} \sum_{i=1}^{\eta_k - s} f_i\left(X\left(j^*\right)\right) + f_{\eta_{k+1}}\left(X\left(j^*\right)\right)\right|$$

$$\leq \frac{b - \eta_k}{b - s}\left(C_R + 1\right)\kappa_{s_0,e_0}^{\max},$$

where the inequality is followed by Lemma 8. Combining with the good events,

$$(II.1) \leq 2\sqrt{\eta_k - s}\frac{b - \eta_k}{b - s}\left(C_R + 1\right)\kappa_{s_0,e_0}^{\max}\gamma$$
(94)

$$\leq 2\frac{4}{\min\left\{1, c_1^2\right\}}\Delta^{-1/2}\gamma\left|b - \eta_k\right|\left(C_R + 1\right)\kappa_{s_0,e_0}^{\max}$$
(95)

As for the term (II.2), it holds that

$$(II.2) \leq 2\sqrt{\left|b - \eta_k\right|}\gamma\left(2C_R + 3\right)\kappa_{s_0,e_0}^{\max}$$
(96)

As for the term (II.3), it holds that

$$(II.3) \leq 2\frac{4}{\min\left\{1, c_1^2\right\}}\Delta^{-1/2}\gamma\left|b - \eta_k\right|\left(C_R + 1\right)\kappa_{s_0,e_0}^{\max}$$
(97)

Therefore, combining (94), (96), (97), (88), (89) and (93), we have that (82) holds if

$$\Delta^2\kappa_k^2(e-s)^{-2}\left|b - \eta_k\right| \gtrsim \max\left\{\gamma^2, \Delta^{-1/2}\gamma\left|b - \eta_k\right|\kappa_k, \sqrt{\left|b - \eta_k\right|}\gamma\kappa_k\right\}$$
(98)

The second inequality holds due to Assumption 3, the third inequality holds due to (79) and the first inequality is a consequence of the third inequality and Assumption 3. □

