# OpenReview forum: "Change point detection and inference in multivariate non-parametric models under mixing conditions"
_NeurIPS.cc/2023/Conference — NeurIPS 2023 poster_

### Official Review · Reviewer_Ng5S · 2023-06-23

**Soundness:** 3 good
**Presentation:** 3 good
**Contribution:** 3 good
**Rating:** 7
**Confidence:** 3

**Summary:**

This paper studies the problem of localization of multiple change points for offline multivariate time series. A non-parametric kernel-based CUSUM statistics is used together with the SBS algorithm. Moreover, a two-step estimation procedure is proposed where the initial estimate is further modified to the final estimate. The consistency result and the limiting distribution of the change point estimators are proved.

**Strengths:**

The main strength of the paper is its theoretical findings. Although the SBS and the kernel CUSUM statistics are both well-known, the theoretical results on the consistency and especially the limiting distribution under the setting of multivariate time series seem to be novel.

**Weaknesses:**

The presentation can still be improved, and the numerical results is a bit limited.

**Questions:**

In the current theoretical results, the consistency result is proved for the initial estimate, while the limiting distribution is provided for the final estimate. It would be better to also present the consistency result for the final estimate \tilde{\eta}, and to elaborate more on whether or not we still have such limiting distribution for the initial estimate.
And it would be better if the author could illustrate in the numerical results the advantage of the second step (refined estimators), i.e., to show the improvement in estimation accuracy after using the refined estimator.

In the numerical results, the standard deviation of the misestimation rate (fig1) should be provided. The report of std can be very helpful here for comparing different methods since the average misestimation rate of MNSBS (blue) seems to be larger than or only slightly smaller than the other four baseline methods in various scenarios.

In the final estimate, line 126 on page 4, there is a typo in the subscript of F_{s_k,\eta_k}, \eta_k should not appear here since it is the unknown true change-point.

It would be better if the author could provide a pictorial illustration of the SBS algorithm.






**Limitations:**

Yes.

---

> ### Author Rebuttal · Authors · 2023-08-08
>
> We are grateful for your constructive comments. We reply to your comments below, with corresponding edits in the revision.
>
> **Weakness**
>
> Thank you for your suggestions.  We will endeavor to improve our presentation and enhance numerical results in the revision.  For this response, extra simulations were conducted. We examined Scenario 1 with $p = 3$, $n \in \{150, 300\}$, and $X_t$ as i.i.d.~$N(0_p, I_p)$. In Scenario 3, we added Random Forest for Change Points (RFCP) as a competitor. Runtime comparisons for methods with $p=3$, $n\in{150, 300}$ are considered. All results are in Tables 1 and 2 (extra page).
>
> **Initial estimators**
>
> As for the initial estimators, the consistency results are presented in Theorem 2, while the limiting distributions are not obtainable based on the current proof techniques.  The current proof relies on a uniform tightness property and the uniqueness of the minimizer in loss functions' population counterpart.  Neither of these two points holds for the initial estimator.
>
> As for the numerical results, in Scenario 1 (additional simulation), we compared our method with and without refinement. Results indicate MNSBS excels in change point localization, and refinement enhances performance, see Table 2 (extra page).
>
> **Report of standard deviations**
>
> We have included stds for all experiments we performed in our paper. See the extra page for more details about the stds in the extra simulations.
>
> **$F_{s, \eta_k}$**
>
> Thank you for pointing out this typo, which we will edit in the revision.
>
> **Pictorial illustration of SBS**
>
> Thank you for your suggestion.  We will refer to the plot in the original SBS paper in the revision.

---

### Official Review · Reviewer_rZb3 · 2023-07-07

**Soundness:** 3 good
**Presentation:** 4 excellent
**Contribution:** 4 excellent
**Rating:** 5
**Confidence:** 3

**Summary:**

This paper proposes algorithm to localize the multiple change points in nonparametric, short-term dependent time series. Assumptions required on the time series are certain mixing conditions and the smoothness in terms of density. The core idea of the algorithm is based on CUSUM and seeded binary segmentation. A two stage estimator is proposed, both of which are consistent, with the refined one in the second step achieving minimax rate under certain smoothness cases. Limiting distribution of the estimators are also given, which allows inference on the change points. Numerical results demonstrates the potential of the proposed method.

**Strengths:**

This is a solid work with nice theoretical results. The problem tacked is a difficult one: localization of multiple change points in a nonparametric, locally dependent time series. The paper establishes not only consistent results, but also gives limiting distributions which can be beneficial for inference.

**Weaknesses:**

Although this is a solid work with nice theoretical results, I feel the related literature is not sufficiently reviewed which are crucial for determining the significance and novelty of this work. I only found in the last paragraph of page 1 a short summary of the existing literature, which I pasted below and raised some of my questions.

"Firstly, to the best of our knowledge, temporal dependence, which commonly appears in time series, has not been considered."
-- I actually don't think this is true, as far as I know, there are a lot of existing work considering temporal dependence of time series (although I agree most of the exiting change point literature still assumes the iid setting). For example, I randomly searched for "change point dependent process" and there seems a bulk of related works, e.g., [1][2]. I believe the authors probably know better than I about these papers, and I wonder whether they could add some clarification on the difference between their setup and the one in this work.

"Secondly, there is no localization consistency result for data with the underlying densities being Hölder smooth with arbitrary degree of smoothness." -- it seems this is comparing to the work of Padilla et al. (2021) which only focuses on Lipschitz smooth densities, is my understanding correct?

"Lastly and most importantly, the limiting distributions of change point estimators and the asymptotic inference for change points have not been well studied." -- Do the authors specifically here refer to the papers on nonparametric, multiple change point literature? I believe the limiting distribution of change point estimators is one of the core questions in change point problems, and I am a bit surprised that the existing papers do not study them. Do the authors mean that most papers have been focusing on consistency results instead of the limiting distributions of change point estimators?

[1] Ray, Bonnie K., and Ruey S. Tsay. "Bayesian methods for change‐point detection in long‐range dependent processes." Journal of Time Series Analysis 23.6 (2002): 687-705.
[2] Dehling, Herold, Aeneas Rooch, and Murad S. Taqqu. "Non‐parametric change‐point tests for long‐range dependent data." Scandinavian Journal of Statistics 40.1 (2013): 153-173.

**Questions:**

My questions have been raised in the "weakness" section.

**Limitations:**

While this work has nice and solid theoretical results, I wonder if practitioners can truly find the method useful, both due to the high computational complexity, and the somewhat complicated limiting distributions (which might prevent the full deployment of this methods' inference capability).

---

> ### Author Rebuttal · Authors · 2023-08-07
>
> We are grateful for your constructive comments. We reply to your comments below, with corresponding edits in the revision.
>
> **On temporal dependence**
>
> Thank you for bringing these references to our attention.  We indeed overlooked them in the literature review stage.  We will modify our claim correspondingly.  We however believe the problems studied are still substantially different, especially that our focus is in the change point inference.
>
> **On the smoothness**
>
> You are indeed right that we are comparing with Padilla et al.~(2021).
>
> **On the inference literature**
>
> You are indeed right that we mean specifically nonparametric change point inference works.
>
> **More literature review**
>
> We provide some more literature review here.  More will be included in the final version in due course.
>
> There exist works such as [1] and [2], which estimate the location and size of structural breaks in nonparametric time series regression models. However, these studies do not address inference, time dependence or varying degrees of smoothness. Our proposed method is distinct in its adaptability to temporal dependence and its capability to handle a broad range of smoothness levels, providing a limiting distribution under these conditions.
>
> There has been prior research on limiting distributions for nonparametric estimators but these mainly focus on single change point detection within independent data. For instance, [3] proposed a nonparametric change point detection and estimation model under the assumption of independent designs and the presence of a single change point. They also present a limiting distribution of the estimators. In contrast, our proposed method permits dependence with an arbitrary degree of smoothness and does not require prior knowledge of the number of change points. Furthermore, unlike [3], we analyse the optimality of our approach.
>
> [4] considers the estimation of change points in a mean detection problem where errors exhibit long-range dependence. Although they derive a limiting distribution of the estimator like our method, their model requires the knowledge of the number of change points, assuming a single change point exists. In contrast, our model does not make such assumptions.
>
> The recent work by [5] investigates the limiting distributions of multiple change point estimators. However, their analysis differs from ours primarily in two areas. First, they assume the number of change points to be known for deriving their theoretical results and propose a data-driven choice for it. Unfortunately, there are no guarantees provided for the estimation of the number of change points when a sequential test is proposed. Their model only ensures the estimation of the number of change points when this quantity is upper-bounded by a finite integer, and the dependence of the estimators on the estimated break fractions is suppressed. The second significant difference is our method's achievement of an optimal rate for estimating the change points, eliminating the log term, whereas [5] does not present such a rate.
>
> [6] has also a derivation of limiting distribution for the CUSUM estimator based on $\alpha$-mixing sequences. This work is different from our work as their analysis is for univariate non-negative α-mixing non-negative random variables.
>
> Past research has studied the limiting distributions of change point estimates in the parametric setting. A key difference here is our extension of these previous works from finite to infinite dimensions, i.e. in nonparametric problems.
>
> **Practical limitations**
>
> We will make our code public available in due course.  Limiting distributions and their consequence of constructing confidence intervals, obtaining $p$-values are of high demands from practitioners.  In the more challenging case when the jump size vanishes, the limiting distributions are based on standard Brownian motions, echoing the universality.
>
> **References**
>
> [1] Mohr, M., & Selk, L. (2020). Estimating change points in nonparametric time series regression models. Statistical Papers, 61(4), 1437-1463.
>
> [2] Delgado, M. A., & Hidalgo, J. (2000). Nonparametric inference on structural breaks. Journal of Econometrics, 96(1), 113-144.
>
> [3] Dumbgen, L. (1991). The asymptotic behavior of some nonparametric change-point estimators. The Annals of Statistics, 1471-1495.
>
> [4] Horváth, L., & Kokoszka, P. (1997). The effect of long-range dependence on change-point estimators. Journal of Statistical Planning and Inference, 64(1), 57-81.
>
> [5] Fu, Z., Hong, Y., & Wang, X. (2023). On multiple structural breaks in distribution: An empirical characteristic function approach. Econometric Theory, 39(3), 534-581.
>
> [6] Gao, M., Ding, S., Wu, S., & Yang, W. (2022). The asymptotic distribution of CUSUM estimator based on α-mixing sequences. Communications in Statistics-Simulation and Computation, 51(10), 6101-6113.

---

### Official Review · Reviewer_TS3z · 2023-07-09

**Soundness:** 3 good
**Presentation:** 2 fair
**Contribution:** 2 fair
**Rating:** 6
**Confidence:** 3

**Summary:**

The submission studies offline multivariate non-parametric change point detection.
The submission proposes a method for this task by combining 1) CUSUM estimator/statistic with 2) seeded intervals and 3) a refining procedure. The proposed estimator has 1) an improved error bound with weaker assumptions and 2) a limiting distribution for inference.
Both empirical and theoretical results are provided supporting the advantages of the proposed method.

**Strengths:**

The submission extends the existing theoretical results from the setting with Lipschitz smooth densities and temporal independence to the case with Hölder smooth densities with $\alpha$ mixing, and also with a limiting distribution for the non-parametric case.

The writing is clear, with notations well defined.



**Weaknesses:**

The submission is not presented in a very motivating way. For example, SBS appears abruptly without the background or reasoning. The methodology is also presented without explaining the motivations.

The methodological contribution of the proposed method is not clear. Combining CUSUM  and SBS is nothing new and also mentioned in the original SBS paper. As a result, the refining step in the proposed method is important for the methodological novelty of the submission. However, unfortunately, there are not enough discussions to motivate this refining step and emphasizing the novelty of this refining step.

The experiments can also be improved. The figures are hard to read since it is hard to match the competing methods to different colors. The advantages of the prosed method are not clearly demonstrated: some bars are very close to each other without showing significant advantages of the proposed method.

**Questions:**

The theoretical results seem to be a good improvement to the existing theory. However, I did not dig deep enough into the proof to evaluate the technical novelty. It is also unclear how much novelty and significance there is in the methodology.

1. Would it be possible for authors to compare to an ablation method, which just combines CUSUM with SBS without the refining step? Or, is this ablation just the SBS method compared in the experiments? If so,  would it be authors to explain why this method (which is quite close to the proposed one just without the refining step) performs so badly?

2. Would it possible for authors to provide stds of the experiment results to make sure that the proposed method really provides significant performance improvement?

As a result, this submission is really on the borderline to me. I tend to accept the submission for the theoretical contributions.

---

> ### Author Rebuttal · Authors · 2023-08-07
>
> We are grateful for your constructive comments. We reply to your comments below, with corresponding edits in the revision.
>
> **Intuition on algorithms**
>
> Thank you for your valuable comments.  We will include all in the camera ready version where an additional page is allowed.  For this revision, we refrain from the edits and just respond to your comments below.
>
> Our method consists of a two-step procedure that incorporates the use of SBS, which is an approach for efficient change point detection in large-scale data.  SBS determines the best-split point for various search intervals in a greedy manner, forming a deterministic pre-computed set of search intervals. This deterministic nature of SBS allows for computational efficiency, achieving a total length for the search intervals that's linear, up to a logarithmic factor, irrespective of the number of change points.
>
> Algorithm 1 is designed to yield consistent change point estimations with high probability. To further refine the precision of our final estimators, we apply a local refinement step. Notably, the interval
>
> $(\hat \eta_{k-1}, \hat{\eta}_{k+1})$
>
> is anticipated to contain a single true change point, $\eta_k$. To ensure that we can exclude any other change points within
>
> $(\hat \eta_{k-1}, \hat{\eta}_{k+1}),$
>
> we trim this interval to $(s_k, e_k)$ as outlined in Equation (4).
>
> **Novelty**
>
> We do not intend to claim much novelty from the algorithmic aspect.  The contributions of our paper lies in the general framework and the theoretical results.  To reiterate, the nonparametric change point inference with temporal dependence is first time studied in the literature.  On top of this, our refining step leads to optimal localisation errors.
>
> **Numerical results**
>
> We will enhance the clarity of figures in the revision.  We would like to highlight that the strengths of our approach not only lie in the estimation precision, but also in its capability to produce statistically valid confidence intervals for these change points, a feature notably absent in many existing methodologies.
>
> **Methods without refining**
>
> The refining step is necessary in two aspects.
>
> Firstly, the first step leads us to an adaptive bandwidth, tailored to each individual change point $\eta_k$. The optimal bandwidth for each change point $\eta_k$ is of order $O(\kappa_k^{1/r})$, where $\kappa_k$ represents the difference in density functions measured in the $L_2$ norm.  In a multiscale setting where the size of the changes can significantly vary across different change points, a non-adaptive choice of bandwidth parameter could lead to sub-optimal estimation errors for the change points.
>
> Secondly, the proof techniques we derive for the limiting distribution relies on the loss function to contain one and only one change point.  This is unachievable without a two-step method.
>
> **Question 1**
> Using only SBS and CUSUM in the first stage will not adaptively set bandwidths for each change point $\eta_k$, risking errors; hence, a two-step procedure is advised for non-parametric time series change points.
>
> **Question 2**
> We have included stds for all experiments we performed in our paper. See the extra page for more details about the stds in the extra simulations.

---

### Official Review · Reviewer_DHN6 · 2023-07-14

**Soundness:** 3 good
**Presentation:** 2 fair
**Contribution:** 3 good
**Rating:** 5
**Confidence:** 2

**Summary:**

The authors consider offline change point detection for multi-variate data where there could be multiple change points (specifically changes in marginal distributions from one time step to the next).  The marginal densities of the underlying generative model are assumed to be smooth (specifically Hölder continuous, which includes Lipschitz continuous functions as a special case).  Building on a procedure using binary-segmentation search with kernel density estimation and consistency (Padilla et al (2021)), the authors analogously show consistency of change-point estimators for time-dependent data.   Furthermore, the authors derive limiting distributions of the change point estimators, both for the situation where the minimal jump size vanishes and for the situation where it remains constant.

**Strengths:**

- The authors study a challenging problem, non-parametric estimation of multiple changepoints.  Many prior works considered independent data and univariate, as well as stronger assumptions on the generative distributions.

- The authors prove consistency and analyze limiting distributions of the estimated change points.  (also this is with less knowledge (such as of optimal bandwidths) than Padilla et al (2021))

- The authors include experiments and show its performance against baselines (designed for independent data).


**Weaknesses:**

My main concerns regard writing/presentation and discussion of related works.  I list a number of specific points below (many of which, by themselves, are fairly minor; list is not exhaustive).

#### Writing/presentation
- There is no discussion on motivating the time-series model class considered ($\alpha $-mixing sequences of random vectors with unknown marginal distributions), or even mention what types of common parametric classes belong to this class.
    -  The property of $\alpha $-mixing is never explained formally let alone any discussion on what that implies at a high-level for the time-series.  There is a large literature on parametric time-series models – it would be valuable to mention example model classes that satisfy the assumptions here.
    - What types of potential applications could this benefit (the time-series of which could plausibly be modeled with this class of models)?  Lines 20-24 mention application areas with time-series data, which is fine, but there should be more of a connection made between (some of) those applications and the specific problem  considered.

- Algorithm 1 and Section 3
    - Provide discussion in the main text about the steps of Alg. 1 and intuition behind the design.
    - SBS is mentioned but is not described (a brief high-level description should suffice)
    - There is notation used in the algorithm that is not mentioned in Section 3, explain what those are.
    - Include discussion about kernel properties in Section 3. Alg 1 uses kernel estimators implicitly in the line with $\tilde{F}$, but up to this point no $\mathcal{K}$ is defined or taken as input, or described in Section 3.
    - (minor) Def 1 text $\mathcal{K}$ has not been described (or listed in the def. set up)
    - (4) and (5) give some verbal description of what this is doing (i.e. what is the intuition behind how you are improving the estimates, possibly with a line or two about the way(s) in which the MNSBS might give poor(er) estimates that leads to this design).  Presumably it has to do with that the choice of interval locations and sizing in MNSBS was using a simple pattern (oblivious to where the estimated change points would be).  Now you have ‘naturally defined’ intervals between the estimated change points (so size and location of the intervals are adapted to the initial estimates).
    - line 124-125 talk about the choice of weights 9/10 and 1/10.
    - line 125 ‘an estimator of $\kappa_k$’ I would suggest to add verbal descriptors for notation to remind the reader, such as ‘an estimator of the jump size $\kappa_k$ at the $k$th change point’ and similarly for other notation.  Also, several lines later (to explain differences with prior work) there is mention that it corresponds to the ‘optimal bandwidth’ and ‘ we use $\hat{\kappa}_k$ as bandwidth for the kernel density estimator’ though in line 127 the bandwidth used is $h_1$ which depends on, but is not set equal to,  $\hat{\kappa}_k$.  A clarification of that part, and a couple sentences shortly after Definition 1 about the bandwidth $h$ to lead the reader to anticipate why $h_1$ in line 127 is good would help.
    - while bandwidths are mentioned earlier (Def. 1 for instance) there is no discussion on bandwidth optimality.  Padilla et al (2021) have nice discussions regarding the choice of the bandwidth; I would suggest summarizing the key points to give intuition to the reader.
    - line 132 ‘smaller interval’ – provide brief intuition about why the intervals here would be smaller than the


#### Experiments
- Include plots of the realizations of the time-series (to give intuition for how hard/easy the task is for those generative models)
- Report the run-times for the proposed method and baselines.  With that also briefly describe the platform.
- It would also be valuable to run experiments with independent data, to see whether the proposed method suffers compared to baselines designed for that setting specifically.

#### Related works
- “Random Forests for Change Point Detection” by Londschien et al https://arxiv.org/abs/2205.04997  a multivariate nonparametric
change point detection methods for independent data (with an R package available) – in their experiments, their method generally was similar or better than ECP (the best baseline in your experiments) and had subquadratic time complexity

- There are works non-parametrically estimating the location and size of structural breaks in non-parametric time-series regression models.  “Estimating change points in nonparametric time series regression models”  by Mohr and Selk (2020) https://doi.org/10.1007/s00362-020-01162-8,  “  Nonparametric inference on structural breaks “ by Delgado and Hidalgo (2000) https://doi.org/10.1016/S0304-4076(99)00052-4 among others.  Given some (at least superficial) relation in the problem considered and some methodological components collectively in that body of work and the present work, which extends methods and results for non-parametric changepoint detection for independent data (esp. Padilla et al (2021)) to allowing for dependencies, I think some discussion on the how the problems, methods, and results of changepoint detection for non-parametric   time-series regression relate would be helpful.  Xu et al (2022) is mentioned in lines 181-182, though what the paper actually studied and how it related was not described.

- A (brief) discussion of similarities and differences between the problem considered here and that of (non-parametric) online change point detection would be good.
- One of the main contributions is characterizing the limiting distributions for multiple change point estimates in this non-parametric setting.  To my knowledge, that is novel for the specific problem considered.   However, similarities/differences in analyses and results (class of distributions that the limiting distribution belongs to) for past works that have investigated  limiting distributions of change point estimates in different by related settings are not well-discussed.  Unless I overlooked it, the only other work mentioned in the context of identifying limiting distribution of change point estimators is Xu et al (2022) in lines 181-182 and that was brief and implicit.
    - There are prior works that studied limit distributions for non-parametric estimators for single change point detection with independent data (as well as works in the parametric setting), although from what I can tell under simpler changes (such as changes in the mean).  For example, “The Asymptotic Behavior of Some Nonparametric Change-Point Estimators” by Dumbgen (1991)  and Horváth and Kokoszka (1997) "The effect of long-range dependence on change-point estimators."   Also “Optimal change-point estimation in time series” by Chan et al. 2021 for limiting distribution of single time-series change point (Bayes) estimators.   There is a recent work “On multiple structural breaks in distribution: An empirical characteristic function approach” by Fu et al (2022) that analyzes the limiting distributions for estimates of multiple change point. Also potentially relevant is “The asymptotic distribution of CUSUM estimator based on $\alpha$-mixing sequences” by Gao et al (2022), though the analysis is for univariate non-negative $\alpha$-mixing non-negative random variables.
    - Limiting distributions of change point estimates have been studied in the parametric setting.  While those results do not diminish the significance of the results in this paper, there should be some mention and preferably a discussion on similarities/differences between the derived distributions.


**Questions:**

#### Questions
- Assumption 3 and Theorem 1.  The $\gamma_T$ does not explicitly show up in the Theorem 1 statement – which constants depend on $\gamma_T$?  Given the assumption statement is only “arbitrarily slow diverging sequence” that seems to me impressive though almost too mild for anything other than statements about the limit $T \to \infty$

- For the theorems, is $\Delta$ *required* to be growing as a function of $T$?  From the assumptions it looks like as long as the jump size is increasing fast enough, we could fix the location of $K$ changepoints and the assumptions would hold.  $\kappa$ growing quickly should make detecting change points easier, but would there still need to be some minimum value of $\Delta$?

- Contribution 1 (lines 84-87) –  The first major contribution claimed is the development of a novel algorithm and statement “To the best of our knowledge, we are the first to innovatively adapt SBS to a multivariate non-parametric change point model”.  From a (perhaps superficial) understanding, the Algorithm 1 proposed here seems to be an incremental adaptation of the procedure proposed in Padilla et al (2021), with the random binary segmentation used in the latter replaced with the recently proposed deterministic binary segmentation method SBS (Kovács et al. (2020)).  Perhaps the authors can add more discussion on why that change is not straightforward.


#### Spelling, Grammar, etc.
- Line 98 ‘A … estimators’
- Theorem 2 statement – The formal statement could be shortened and I think easier to read if the notation (eg 188-189, 192- (7)) were introduced and discussed (add simple description of $P_k$’s formula when it is introduced) before the formal theorem statement.
- For theorem 3, maybe $\max_{1\leq k \leq K} \dots$
- line 291 ‘additional additional’
- lines 291-293 – that is great you had further experiments; mention that earlier both in introduction and early on in Section 5.


**Limitations:**

It is fine (theoretical contribution)

---

> ### Author Rebuttal · Authors · 2023-08-08
>
> We are grateful for your constructive comments. We reply to your comments below, with corresponding edits in the revision.
>
> **Motivation on $\alpha$-mixing sequences.**
>
> Thank you for pointing this out. We have included the following in the revision.
>
> The $\alpha$-mixing condition with exponential decay as specified in Assumption 1.e  is a commonly held assumption in time series analysis. A broad spectrum of multivariate time series satisfies this condition,  including linear/nonlinear VAR models [e.g. 1],   a comprehensive class of GARCH models [e.g. 2], and various Markov processes [e.g. 3].
>
> **Algorithm 1 and Section 3**
>
> Thank you for all your valuable comments.  We will include all in the camera-ready version where an additional page is allowed.  For this revision, we refrain from the edits and just respond to your comments below.
>
> *Design and intuition of Algorithm 1*
>
> Algorithm 1 has the SBS as its skeleton and a nonparametric version of the CUSUM statistics as its organs.  This is tailored to its nonparametric and potentially multiple change points nature.
>
> *High-level description of SBS*
>
> SBS is a multiscale version of a moving-window scanning method.  To conquer the potentially multiple change points with unknown spacing, instead of using a fixed window width, SBS uses a collection of window width choices, each of which is applied to a moving-window scanning.
>
> *Notation not used in Section 3*
>
> Thanks for pointing it out. These pieces of notation are only local in the algorithm and are irrelevant to the rest of the paper.
>
> *Kernel function*
>
> For Algorithm 1 to execute, there are no further theoretical assumptions needed for the kernel function.  In Definition 1, which is at the beginning of Section 3, the kernel function is already mentioned.  In the revision, we have specified that $\mathcal{K}$ is a kernel function.
>
> *Equations (4) and (5), the choice of the weights and ``smaller interval''*
>
> Given the consistency of the initial change point estimators procured from Algorithm 1, the interval $(\widehat \eta_{k-1},\widehat \eta_{k+1})$ is anticipated to contain merely one undetected change point. By conservatively trimming this interval to $(s_k, e_k) $, we can safely any change points previously detected within $ (\widehat \eta_{k-1},\widehat \eta_{k+1}) $. Consequently, the trimmed interval $ (s_k, e_k)$ is likely to contain only true change point $ \eta_k$ with high probability. Due to the same reason,  our choice of weight in (4),  1/10, is a convenient choice.  Any constant weight between 0 and 1/2 would suffice.
>
> *The choice of the bandwidth and its optimality*
>
> Inspired by Padilla et al. (2021), who proposed to use $O(\kappa_k) $ as an optimal bandwidth in the context of Lipschitz densities, we adopt $h_1 = O(\hat{\kappa}_k^{1/r})$ as the bandwidth for our kernel density estimator.  This choice incorporates the broader scope of our work, which studies a more general degree of smoothness.  Notably, if the underlying density functions strictly adhere to the Lipschitz criterion and $r = 1$, our bandwidth selection aligns with that recommended by Padilla et al. (2021).
>
> **Experiments and Random forests**
>
> See general rebuttal.
>
> **Related works**
>
> *Literature in nonparametric CP, online CP, and CP inference*
>
> Thank you very much for bringing up these valuable suggestions.  Due to the limited time and space, we will defer the detailed comparisons along with corresponding edits to the revision.
>
> In this response, we briefly comment on the comparisons.
>
> In the nonparametric change point literature, different kernel-based methods are adopted for change point localisation and testing.  Compared to the existing work, we follow the suit of using kernel-based CUSUM statistics but incorporate temporal dependence, which is rarely seen in the literature.  Most importantly, we are unaware of existing work on nonparametric change point inference, which is the main selling point of our paper.
>
> In terms of online and offline CP comparisons, the core methodology is largely shared, but with different goals and performance measurements.  It is also unclear how to conduct inference in the online CP context.
>
> Most CP inference work focuses on fixed-dimensional parameters as well as lacks tracking of many model parameters. Xu et al.~(2022), in terms of style, is indeed the most closely related. but tackles high-dimensional linear regression, fundamentally distinct from our nonparametric density estimation.
>
> **$\gamma_T$**
>
> In Assumption 3, we require $\kappa^2 \Delta \log^{-1}(T) T^{-p/(2r+p)} > \gamma_T$, with $\gamma_T$ diverging arbitrarily slow.  Although $\gamma_T$ does not appear in the theorem statements explicitly, its impact can be seen.  For instance, Theorem 1 shows that, with large probability
> $$
>     |\widehat{\eta}_k - \eta_k| \lesssim \kappa_k^{-2}T^{p/(2r+p)} \log(T),
> $$
> which implies that
> $$
>     |\widehat{\eta}_k - \eta_k|/\Delta \lesssim \gamma_T^{-1} \to 0,
> $$
> as $T$ diverges.  This regulates that each CP estimator is close to one and only one true CP.  This is the key to the success of deriving limiting distributions.
>
> **Conditions on $\Delta$**
>
> For $\Delta$, we only require that the signal-to-noise ratio functions of $\kappa$ and $\Delta$ satisfy Assumptions 3 and 4.  In nonparametric literature, the $L_2$-norms of densities are assumed to be bounded, which leads to bounded $\kappa$.
>
> **References**
>
> [1] Eckhard Liebscher. Towards a unified approach for proving geometric ergodicity and mixing properties of nonlinear autoregressive processes. Journal of Time Series Analysis, 26(5):669–689, 2005.
>
> [2] Farid Boussama, Florian Fuchs, and Robert Stelzer. Stationarity and geometric ergodicity of bekk multivariate garch models. Stochastic Processes and their Applications, 121(10):2331–2360, 2011.
>
> [3] Kung-Sik Chan and Howell Tong. Chaos: a statistical perspective. Springer Science & Business Media, 2001.

---

### Official Review · Reviewer_XKms · 2023-07-14

**Soundness:** 3 good
**Presentation:** 3 good
**Contribution:** 3 good
**Rating:** 7
**Confidence:** 3

**Summary:**

This paper studies non-parametric offline change-point detection with the assumption that the probability density functions are Holder continuous on some compact support, and the time series is $\alpha$-mixing. The authors propose a two-stage algorithm, first roughly divide the time series into different segments, then refine the change-point locations. The asymptotic distribution of the estimated change-points are derived, and as a third step, the author also proposes an algorithm to compute a confidence interval for each change-point. Numerical experiments show the effectiveness of the proposed methods.

**Strengths:**

1. The non-parametric model used in this paper is Holder continuous class and is general.
2. The asymptotic distribution of the estimated change-point location is derived for the vanishing and non-vanishing regime respectively.
3. The time series can have temporal dependencies under mixing conditions.
4. The writing and the structure of the paper is clear.

**Weaknesses:**

1. The authors claim their model is under mixing conditions but the theory parts' dependency on the $\alpha$-mixing coefficient $c$ in Assumption 1.e is hardly discussed in the main text.

Minor issue:
1. The equation under line 126: should remove $\arg\min$.
2. The equation under line 126: why is there $\eta_k$? Isn't $\eta_k$ unknown?
3. Line 131: Why is $\widetilde \eta_k$ referred to as the `kernel density estimator'? From previous context it is the change-point estimator.

**Questions:**

1. In Assumption 2.a, is it for any $h>0$?
2. In Assumption 2.b, the end of line 155, what is $v$ on the exponent? Do you mean $\nu$?
3. In Theorem 2.a the non-vanishing regime, do you need $f_{\eta_k},f_{\eta_{k+1}}$ to converge in distribution as well or they can be arbitrary as long as the jump size $\kappa_k$ converges to a constant? How does the $\alpha$-mixing works here? What is $F_{t,h_2}$ for $t<0$? Why the limiting distribution of $\widetilde \eta_k$ depends on $f_0,f_1,f_2$ when $\eta_k$ is far from 0?
4. In equation 7, should it be $\kappa_k^{p/r+2}$ on the numerator or just $\kappa_k^{p/r-2}$?

**Limitations:**

As stated in the submitted paper.

---

> ### Author Rebuttal · Authors · 2023-08-07
>
> We are grateful for your constructive comments.  We reply to your comments below, with corresponding edits in the revision.
>
> **$\alpha$-mixing coefficients**
>
> We appreciate the reviewer's comments regarding the need for a more detailed discussion on $\alpha$-mixing coefficient $c$. We will include the following discussion in our revised manuscript.
>
> Throughout this paper, we focus on multivariate time series that exhibit $\alpha$-mixing behavior with exponential decay coefficients. This condition is denoted as Assumption 1.e. While the constant $2c$ is present in the exponent of the exponential function, it plays a non-essential role in our theoretical framework. We include it solely for the sake of convenience during verification.
>
> The $\alpha$-mixing condition with exponential decay as specified in Assumption 1.e  is a commonly held assumption in time series analysis. A broad spectrum of multivariate time series satisfies this condition,  including linear/nonlinear VAR models [e.g. 1],   a comprehensive class of GARCH models [e.g. 2], and various Markov processes [e.g. 3]. To further elaborate,  consider the
> $p$-dimensional stationary VAR(1) model:
>         $$X_t = A X_{t-1} + \epsilon_t,$$
> where $A$ is the $p \times p$ transition matrix whose spectral norm satisfying $||A|| \in (0,1)$ and the innovations $\epsilon_t$ are i.i.d. Gaussian vectors. Denote $\Sigma = cov(X_1)$, and let  $\lambda_{\max}$ and $\lambda_{\min}$   be  the largest and smallest eigenvalues of $\Sigma$. Then by Theorem 3.1 in [4], we have that for any $k \geq 0$, the $\alpha$-mixing coefficient of the time series $X_t$ satisfying
>         \begin{align*}
>             \alpha_k \leq \sqrt{\frac{\lambda_{\max}}{\lambda_{\min}}}\|A\|^k \leq e^{-C\log(1/\|A\|)k},
>         \end{align*}
>         where $C > 0$ is some constant depending only on $\sqrt{\lambda_{\max}/\lambda_{\min}}$. In this example, the constant $C\log(1/\|A\|)$ corresponds to the constant $2c$ in Assumption 1.e.
>
> Essentially, Assumption 1.e is useful to unlock several technical tools under temporal dependence, which include a Bernstein’s inequality [5], a moment inequality [see Proposition 2.5 in 6], maximal inequalities (see Section G.1) and a central limit theorem (see Section G.2). For instance, we utilize the moment inequality to bound the autocovariances of a dependence process with all lags by $\alpha$-mixing coefficients, thereby demonstrating the existence of the long-run variance, which is the sum of all the autocovariances.
>
> **Minor issues**
>
> Thank you for pointing these out.  The quantity $\eta_k$ should be $\hat{\eta}_k$ and we have changed in our revision.  Regarding Line 131, we have edited to "*we use $\widehat{\kappa}_k$ as bandwidth for the kernel density estimator in deriving $\widetilde \eta_k$*".
>
> **Questions 1 & 2**
>
> Thank you for pointing these out.  We have corrected them correspondingly.
>
> **Question 3**
>
> In Theorem 2.a, you are  right that we do need the density function $f_{\eta_k} $ to remain as a constant or converge asymptotically, we will include this condition in our revision.
>
> The $\alpha$-mixing condition is imposed on the data sequence, while $f_{\eta_k}$ and $f_{\eta_{k+1}}$ are population marginal density functions.
>
> For  $F_{t, h_2}$, you are right that it is a typo.  For each change point $\eta_k$, it should be $ F_{\eta_k+t,h_2}$ and $ t<0$ corresponds to the time series before the change point $\eta_k$.  With this correction, the distribution of $ \widetilde \eta_k$ does not depend on $f_0, f_1, f_2$ when $\eta_k$ is far from 0.  To be more precise, as a byproduct of the limiting distribution in Theorem 2, $|\widetilde \eta_k -\eta_k| =O_p(\kappa_k^{-r/p-2})=o(\Delta) $.  With high probability, the distribution of $ \widetilde \eta_k$, therefore, only depends on data in the interval $(\eta_{k-1},\eta_{k+1})$.
>
> **Question 4**
>
> We appreciate the reviewer's observation.  In equation (7), the correct expression is indeed $\kappa_k^{p/r-2}$.
>
>
> **References**
>
> [1] Eckhard Liebscher. Towards a unified approach for proving geometric ergodicity and mixing properties of nonlinear autoregressive processes. Journal of Time Series Analysis, 26(5):669–689, 2005.
>
> [2] Farid Boussama, Florian Fuchs, and Robert Stelzer. Stationarity and geometric ergodicity of bekk multivariate garch models. Stochastic Processes and their Applications, 121(10):2331–2360, 2011.
>
> [3] Kung-Sik Chan and Howell Tong. Chaos: a statistical perspective. Springer Science & Business Media, 2001.
>
> [4] Fang Han and Wei Biao Wu. Probability inequalities for high-dimensional time series under a triangular array framework. In Springer Handbook of Engineering Statistics, pages 849–863. Springer, 2023.
>
> [5] Florence Merlevède, Magda Peligrad, Emmanuel Rio, et al. Bernstein inequality and moderate deviations under strong mixing conditions. High dimensional probability V: the Luminy volume, 5:273–292, 2009.
>
> [6] Jianqing Fan and Qiwei Yao. Nonlinear time series: nonparametric and parametric methods. Springer Science & Business Media, 2008.

---

### Author Rebuttal · Authors · 2023-08-08

We are grateful for your constructive comments. Taking advantage of the extra page submission we include the following extra simulations to help us to address each of the particular reviews.

**Independent data**

We examined Scenario 1 with $p = 3$, $n \in \{150, 300\}$, and $X_t$ as i.i.d.~$N(0_p, I_p)$. Results indicate MNSBS excels in change point localization, and refinement enhances performance, see Table 2.

**New competitor (Random Forest for change point)**

We compare the random forest change point (RFCP) method in [1] with our proposed method and the rest of the competitors, using Scenario 3 of our simulation studies. The results are summarized in Table 1, which shows that our MNSBS is generally outperformed.


**Runtime comparison**

We compared our method's runtime with others on a machine equipped with an Apple M2 chip (8-core CPU) for $p=3$ and $n\in\{150, 300\}$ in the independent setting.  Our method is comparable at $n=150$ but slower at $n=300$ due to CUSUM's computational demands. See Table 2 for more details.


**Plots**

We include a plot of realizations of the time series under the independent setting, in Figure 1. See the supplementary materials for a plot illustrating our considered real data.

**References**

[1] Malte Londschien, Peter Buhlmann, and Solt Kovacs. Random forests for change point detection. arXiv preprint arXiv:2205.04997, 2022

---

### Decision · Program_Chairs · 2023-09-21

**Decision:**

Accept (poster)

**Comment:**

This paper studies the non-parametric offline change-point detection in multivariate time series, focusing on scenarios where probability density functions exhibit Hölder continuity. The authors suggest a two-stage process: segmenting the time series and then refining the identified change-points. They introduce a method integrating the CUSUM estimator, seeded intervals, and a refinement procedure and present enhanced error bounds. The paper extends prior work by demonstrating consistency in change-point estimators for time-dependent data. Additionally, the authors present the asymptotic distribution of estimated change-points and an algorithm to determine confidence intervals for each identified change-point. Empirical and theoretical evidence underscores the efficiency and advantages of the proposed methods. I believe the paper provides a valuable contribution to the literature.